# Prodrug florfenicol amine is activated by intrinsic resistance to target *Mycobacterium abscessus*

Non-tuberculous mycobacteria are emerging pathogens with high intrinsic drug resistance. Among these, *Mycobacterium abscessus* is particularly refractory owing to its extensive array of resistance mechanisms. Here we introduce florfenicol amine (FF-NH$_2$), a major metabolite of the antibiotic florfenicol, which acts as a prodrug with narrow-spectrum activity against *M. abscessus*–*chelonae* complex species. FF-NH$_2$ leverages intrinsic *M. abscessus* resistance conferred by the transcription factor WhiB7. It avoids WhiB7-dependent resistance mediated by the *O*-acetyltransferase Cat and is activated by the WhiB7-dependent *N*-acetyltransferase Eis2 in a prodrug fashion to generate the active translational inhibitor FF acetyl (FF-ac). FF-NH$_2$ induces Eis2 expression through WhiB7, creating a feed-forward bioactivation loop, which increases FF-ac accumulation and antimicrobial action. FF-NH$_2$ displays antiresistance properties, can synergize with other antibiotics and mitigates toxicity linked to mammalian mitochondrial ribosome inhibition. Importantly, FF-NH$_2$ demonstrated efficacy in a murine model of *M. abscessus* infection. These findings suggest intrinsic resistance can be exploited to develop safer and more effective treatments for this pathogen.

Chronic lung disease due to non-tuberculous mycobacteria (NTM) is a growing and silent global epidemic[1–4]. The incidence and prevalence of NTM pulmonary disease (NTM-PD) now surpasses that of tuberculosis in developed countries[5], where it disproportionately affects high-risk patients, including those with immunocompromised conditions[6] and obstructive lung disease[7,8]. *Mycobacterium avium* complex, *Mycobacterium kansasii* and *Mycobacterium abscessus* are the most common pathogens in NTM-PD[4], with *M. abscessus* being particularly resistant, earning it the label 'antibiotic nightmare'[9,10]. Consequently, therapeutic strategies to combat *M. abscessus* infections are prolonged and involve a combination of antimicrobial agents[11], many of which exhibit suboptimal activity and considerable toxicity[12,13]. Despite this aggressive approach, cure rates for NTM-PD caused by *M. abscessus* remain poor (<50%)[14], contributing to notable long-term mortality, which is estimated to be as high as 50% at 15 years[15]. Moreover, the NTM drug pipeline is limited[16], highlighting the urgent need for safer, long-term therapies against *M. abscessus*[17].

Intrinsic resistance in mycobacteria is often attributed to antimicrobial permeability limitations and the expression of drug-resistance proteins[18,19]. Although these resistance elements can vary across mycobacteria, the WhiB7 'resistome' is conserved and central in limiting the activity of many ribosome-targeting antibiotics[20–25]. WhiB7 is a transcriptional regulator associated with ribosomal stress induced by antibiotics and environmental changes[26]. In *M. abscessus*, WhiB7 (MAB_3508c) controls the expression of over 100 response genes[24,25], including those involved in drug resistance, such as *erm*(41) methylase[27], *eis2 N*-acetyltransferase[28] and the efflux pumps *tap*[25] and *tetV*[29]. The rational design of semi-synthetic analogues offers a promising strategy to overcome *M. abscessus* resistance by these mechanisms. Notable examples include next-generation tetracycline analogues that

✉e-mail: psander@imm.uzh.ch; Richard.Lee@StJude.org

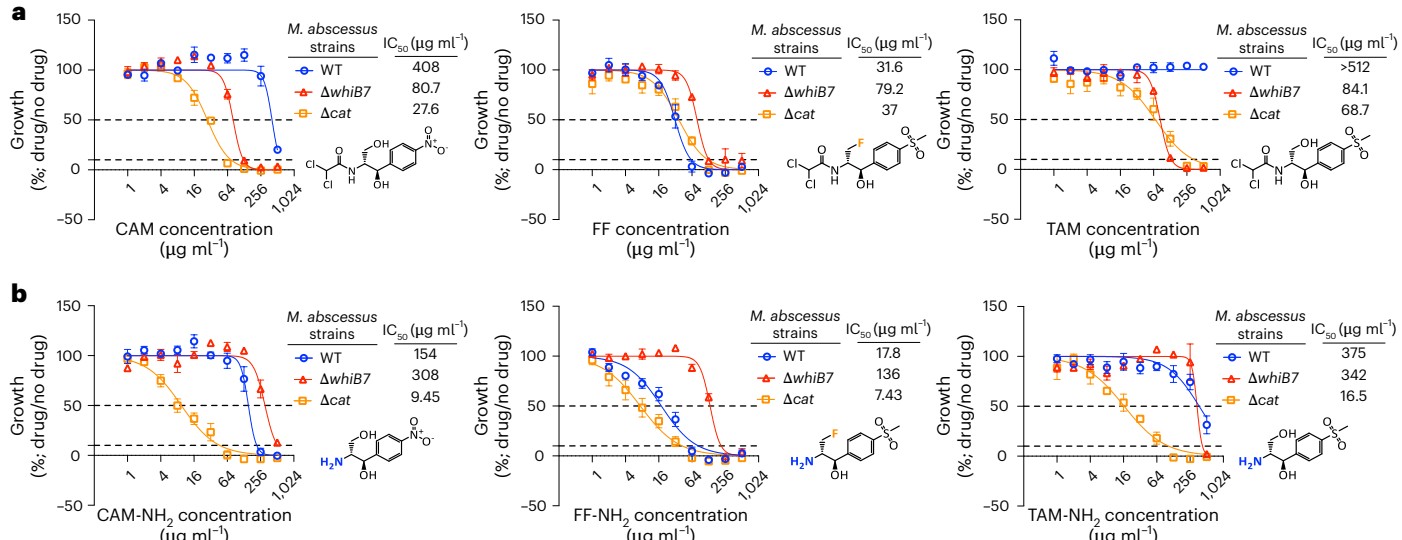

**Fig. 1 | Interplay between WhiB7 and phenicol antibiotics exposes vulnerability. a**, Dose-response curves, chemical structures and $IC_{50}$ values (in µg ml$^{-1}$) for CAM, FF and TAM tested against *M. abscessus* WT, *M. abscessus* Δ*whiB7* and *M. abscessus* Δ*cat*. **b**, Dose-response curves, chemical structures and $IC_{50}$ values (in µg ml$^{-1}$) for CAM-NH$_2$, FF-NH$_2$ and TAM-NH$_2$ tested against the same strains. Florine substitution (orange) and primary amine (blue) are indicated on structures. All curves shown as mean ± s.e.m. from $n = 3$ biological replicates. The WT genotype is *M. abscessus* ATCC19977.

evade MabTetX-mediated inactivation[30,31], rifamycin analogue that circumvent ADP ribosylation[32,33] and spectinomycin (SPC) analogues that overcome TetV-mediated efflux[34,35]. These advancements provide a robust framework for effective strategies for future analogue development campaigns.

Chloramphenicol (CAM) was among the first antibiotics identified and remains on the World Health Organization's List of Essential Medicines[36,37]. CAM functions as a translational inhibitor, binding the bacterial 50S large ribosomal subunit at the interface between the peptidyltransferase centre (PTC) and the growing polypeptide chain[38]. Presently, its clinical use is limited owing to its toxicity, as it can similarly inhibit mammalian mitochondrial ribosomes[39], leading to major side effects such as bone marrow suppression[40]. Additionally, closely related compounds within the phenicol class, such as florfenicol (FF), are widely used in veterinary medicine[41]. Developing phenicols for *M. abscessus* infections requires circumventing WhiB7-mediated resistance as well as maintaining oral bioavailability, compatibility with treatment regimens and tolerability for long-term use. Given the limited research on phenicols and resistance mechanisms in *M. abscessus*, we focused our research on these aspects in our development efforts. Here, we describe the elucidation of CAM resistance in *M. abscessus* and report the discovery of florfenicol amine (FF-NH$_2$), a prodrug with selective activity against the *M. abscessus*–*chelonae* complex. FF-NH$_2$ overcomes and exploits WhiB7-mediated resistance, synergizes with existing *M. abscessus* antibiotics, avoids mammalian toxicity and demonstrates efficacy in a murine infection model, highlighting its potential as a therapeutic strategy against this emerging pathogen.

## Results

### WhiB7 expression results in enhanced FF-NH$_2$ activity against *M. abscessus*

To assess the impact of the WhiB7 regulon on CAM activity in *M. abscessus*, we conducted susceptibility testing using *M. abscessus* ATCC19977 (wild type (WT)) and an isogenic *MAB_3508c* deletion strain (Δ*whiB7*). The potency of CAM increased fivefold against Δ*whiB7*, like that of amikacin (AMK) control (3.7-fold), which is subject to WhiB7-dependent *N*-acetylation encoded by *MAB_4532c* (*eis2*)[28] (Fig. 1 and Extended Data Fig. 1a). This suggested a WhiB7-dependent element impacted CAM susceptibility. To investigate this possibility,

we performed RNA sequencing (RNA-seq) in the WT strain using a sub-minimum inhibitory concentration (MIC) of CAM for 30 min. CAM strongly induced the WhiB7 regulon, notably increasing the expression of a WhiB7-regulated *O*-acetyltransferase *MAB_2989* (log$_2$ fold change (FC) 2.7, false discovery rate (FDR) <0.01; Extended Data Fig. 1b), which has previously been inferred as a probable CAM acetyltransferase (further denoted as *cat*)[42]. Transformation of *Mycobacterium smegmatis* with the multicopy vector pOLYG-*aac(3)IV-cat* decreased CAM activity eightfold compared with pOLYG-*aac(3)IV* vector control (MIC of 128 µg ml$^{-1}$ versus 16 µg ml$^{-1}$) (Supplementary Table 1). An isogenic *MAB_2989* deletion was then constructed in *M. abscessus* (Δ*cat*) showing an increase in CAM activity when compared with the WT strain (Fig. 1a), suggesting that *cat* mediates CAM resistance. Similarly, clinical isolates of the *M. abscessus* complex that possessed *cat* frameshift mutations were more susceptible to CAM (Supplementary Table 1). To determine whether this observation was CAM specific or applies more broadly to other phenicols, we then tested florfenicol (FF) and thiamphenicol (TAM), which differ by a single fluorine replacement, against these strains. While TAM activity was enhanced against Δ*cat*, similar to Δ*whiB7*, FF activity remained unchanged (Fig. 1a), indicating that the fluorine substitution of the C3-hydroxyl in FF prevents inactivation by Cat.

To facilitate future chemistry campaigns, we synthesized amine derivatives of these phenicols to serve as versatile intermediates for subsequent chemical modifications. Routine *M. abscessus* susceptibility testing of these intermediates revealed FF-NH$_2$, a major metabolite of FF in many domesticated animals[43–45], to have similar activity compared with FF in WT and Δ*cat* strains, displaying a MIC of 64 µg ml$^{-1}$ or 225 µM (Fig. 1b). In contrast, chloramphenicol amine (CAM-NH$_2$) and thiamphenicol amine (TAM-NH$_2$) were only active against the Δ*cat* strain (Fig. 1b). Surprisingly, the activity of FF-NH$_2$ was dependent on WhiB7 but in an unexpected manner: FF-NH$_2$ was 7.6-fold more potent against the WT (half-maximum inhibitory concentration ($IC_{50}$) of 17.8 µg ml$^{-1}$ or 62.7 µM) compared with the Δ*whiB7* strain ($IC_{50}$ of 136 µg ml$^{-1}$ or 479 µM). Similar decreases in activity were observed for CAM-NH$_2$ and TAM-NH$_2$ in the Δ*whiB7* strain compared with the Δ*cat* strain, despite WhiB7's established function as a positive regulator of *cat* expression[24,25] (Fig. 1b). These results suggest that WhiB7 expression is a vulnerability in the context of amine phenicol treatment. Given the

role of WhiB7 in antibiotic resistance in *M. abscessus*[23–25,27–29], we sought to explore the mechanism by which WhiB7 enhances FF-NH$_2$ activity.

## The aminoglycoside-modifying enzyme Eis2 activates FF-NH$_2$

We began by selecting FF-NH$_2$-resistant *M. abscessus* mutants on agar plates, which arose at a frequency of $1 \times 10^{-6}$ cells. At a concentration of 64 or 128 µg ml$^{-1}$ FF-NH$_2$, there were noticeable differences in colony size and morphology reflecting two separate populations: (1) a large, rough morphotype and (2) a small, smooth morphotype (Fig. 2a and Extended Data Fig. 2a). At a concentration of 256 µg ml$^{-1}$ FF-NH$_2$, only the small morphotype was observed. Sequencing analysis revealed that the large colonies harboured mutations in the transcription factor *whiB7*, and the small colonies had mutations in the WhiB7-dependent *N*-acetyltransferase *eis2* (Fig. 2a–c and Supplementary Table 2). Notably, these phenotypic changes were transient, as passaging *whiB7* and *eis2* mutants in antibiotic-free media reverted the colonies to the smooth morphotype seen with the WT strain (Extended Data Fig. 2b,c). For the *whiB7* mutants, we identified multiple non-synonymous single nucleotide polymorphisms (SNPs) and insertion–deletions impacting conserved WhiB7 elements. These included mutations near the iron-binding cysteine residues (H16P, L37P and A51T), GVWGG β-turn motif (G59R and G62C/A) and the DNA-binding AT-hook (R76del and P77fs)[46] (Fig. 2b). Among the *eis2* mutants, a substantial proportion (27 out of 45 unique mutations) exhibited substantial structural changes due to frameshift mutations and the introduction of premature stop codons (Supplementary Table 2). Additional SNPs identified in the *eis2* gene conferred amino acid changes near the acetyl-CoA binding site[47] (V83D, R90W, E120K and G128V) (Fig. 2c). Notably, no mutations were found in the 23S rRNA CAM binding site, indicating that FF-NH$_2$ resistance is probably mediated through *whiB7* or *eis2* disruption.

To confirm Eis2 expression is responsible for amine phenicol activity, we conducted susceptibility testing using both isogenic deletion strains (Δ*eis2* and Δ*whiB7*) and spontaneously generated missense mutants (Eis2 T258R and WhiB7 A51T), each complemented with either the multicopy pOLYG-*aac(3)IV-eis2* vector or the pOLYG-*aac(3)IV* control vector. Introduction of pOLYG-*aac(3)IV-eis2* restored FF-NH$_2$ activity in each strain tested (Fig. 2d,e). CAM-NH$_2$ also showed improved activity in the pOLYG-*aac(3)IV-eis2* complemented strains relative to pOLYG-*aac(3)IV* control, whereas the antimicrobial activity of clarithromycin (CLR) and linezolid (LZD) remained unchanged (Fig. 2d,e and Extended Data Fig. 3a,b). In contrast, AMK was less potent in strains complemented with pOLYG-*aac(3)IV-eis2* than in those with the pOLYG-*aac(3)IV* control vector (Fig. 2d,e). These results confirm that Eis2 increases efficacy of FF-NH$_2$ and decreases AMK activity.

To demonstrate that FF-NH$_2$ is enzymatically modified by Eis2, we performed a biochemical kinetic assay using Eis2 purified from *Escherichia coli*[47]. Our results confirm that FF-NH$_2$ is a substrate for Eis2, with a $k_{cat}$ of $11.7 \pm 0.9$ s$^{-1}$ and a specificity constant ($k_{cat}$ $K_M^{-1}$) of $6.7 \times 10^3 \pm 5.1 \times 10^2$ M$^{-1}$ s$^{-1}$ (Fig. 2e, Supplementary Fig. 4 and Supplementary Table 3). Consistent with previous findings, the aminoglycoside controls AMK and hygromycin B were also shown as substrates of Eis2 (ref. 47), as well as CAM-NH$_2$ and TAM-NH$_2$. Notably, these compounds all displayed higher catalytic efficiencies compared with FF-NH$_2$ (Fig. 2f, Extended Data Fig. 4a–c and Supplementary Table 3).

This observation, alongside the enhanced activity of CAM-NH$_2$ and TAM-NH$_2$ against the Δ*cat* strain (Fig. 1b), suggest that these compounds would display antimicrobial activity if they were not also substrates for Cat. In contrast, the dichloroacetyl parent molecules FF, CAM and TAM were not identified as substrates for Eis2 (Fig. 2f, Extended Data Fig. 4a–c and Supplementary Table 3). These findings highlight how specific chemical modifications to antibiotics govern substrate specificity and antimicrobial efficacy.

We hypothesized that Eis2, a GCN5-related *N*-acetyltransferase (GNAT), acetylates FF-NH$_2$ in *M. abscessus* cells, producing the active metabolite FF acetyl (FF-ac). To test whether Eis2-mediated acetylation of FF-NH$_2$ is necessary for ribosomal inhibition, we conducted a cell-free translation inhibition assay using S30 extracts from *M. smegmatis*. The results revealed that FF and CAM display similar potency for inhibiting mycobacterial ribosomes (absolute IC$_{50}$ of 16.9 and 20.2 µM, respectively), whereas FF-ac showed slightly reduced potency (absolute IC$_{50}$ of 39.2 µM) (Fig. 2g and Supplementary Table 4). However, FF-NH$_2$ displayed markedly decreased activity (absolute IC$_{50}$ > 800 µM) (Fig. 2g and Supplementary Table 4), confirming that acetylation of FF-NH$_2$ to FF-ac by Eis2 is crucial for ribosome inhibition and antimicrobial activity.

## FF-NH$_2$ displays a feed-forward mechanism of action

Given that FF-NH$_2$ is a weak ribosomal inhibitor whose activity depends on WhiB7 and Eis2 expression, we hypothesized that its activation is WhiB7 and time dependent. To test this, we took a multifaceted approach utilizing a platform consisting of RNA-seq, quantitative proteomics and an accumulation/conversion assay, all conducted in exponentially growing *M. abscessus* cells over a 3 h period (Fig. 3a).

Our RNA-seq analysis revealed that FF-NH$_2$ treatment at 100 µM (approximately 1/2MIC) upregulates WhiB7 target genes at 0.5 and 3 h compared with vehicle control (Fig. 3b,c). Specifically, at 0.5 h, 104 differentially expressed genes (log$_2$FC >|2|, FDR <0.01) were WhiB7 targets, which increased to 128 differentially expressed genes after 3 h. These results demonstrate that FF-NH$_2$ strongly induces the transcription of genes involved in amine phenicol activity, including *eis2*, *cat* and *whiB7* (Fig. 3b,c). We also compared the transcriptomic profiles for *M. abscessus* cells treated with 25 µM FF-NH$_2$ (approximately 1/8MIC) or FF. While FF treatment induced similar levels of *whiB7* and *eis2* at both 0.5 h and 3 h, FF-NH$_2$ treatment resulted in a more pronounced upregulation of these genes over the same period, highlighting the time-dependent nature of *M. abscessus*'s response to FF-NH$_2$ (Extended Data Fig. 5a,b). The direct comparison of the 3 h FF and FF-NH$_2$ treatment profiles identified only 11 significantly dysregulated genes, suggesting a consistent mechanism of action for both compounds (Extended Data Fig. 5c,d).

To confirm that FF-NH$_2$-induced transcriptomic changes were reflected at the protein level, we analysed the proteomes of *M. abscessus* cells treated with 100 µM FF-NH$_2$ at 0.5 h and 3 h utilizing data-independent acquisition mass spectrometry (MS). The results matched the transcriptomic profiles, showing 18 WhiB7 target proteins were differentially expressed (log$_2$FC >|1|, FDR <0.01) at 0.5 h, increasing to 39 WhiB7 target proteins by 3 h after FF-NH$_2$ treatment (Fig. 3d,e). The abundance of Eis2, Cat and WhiB7 were significantly increased at

**Fig. 2 | The aminoglycoside-modifying enzyme Eis2 activates FF-NH$_2$.**
**a**, A schematic of colony size, morphology and frequency for FF-NH$_2$-resistant colonies selected on agar with 64 µg ml$^{-1}$ compound. Large colonies carry mutations in *whiB7* and small colonies carry mutations in *eis2*. **b**, The structure of WhiB7 from *Mycobacterium tuberculosis* (PDB: 7KUF)[41] with red circles marking homologous mutation sites in FF-NH$_2$-resistant mutants. Conserved regions: green, iron-binding cysteine residues; purple, GVWGG β-turn motif; blue, DNA-binding AT-hook. **c**, The structure of one subunit of Eis2 from *M. abscessus* (PDB: 6RFT)[42] with red circles indicating SNP sites in FF-NH$_2$-resistant mutants. SNPs near the acetyl-CoA (yellow sticks) binding site are shown as red sticks.

**d**, Dose-response curves for CAM-NH$_2$, FF-NH$_2$ and AMK against *M. abscessus* Δ*eis2* or *M. abscessus* Eis2 T258R strains complemented with pOLYG-*aac(3)IV-eis2* overexpression or pOLYG-*aac(3)IV* control vector. **e**, Dose-response curves for CAM-NH$_2$, FF-NH$_2$ and AMK against *M. abscessus* Δ*whiB7* or *M. abscessus* WhiB7 A51T strains, with vector complementation as above. **f**, Michaelis–Menten curves for FF/FF-NH$_2$ and AMK in an Eis2 biochemical assay. **g**, Dose-response curves for FF, FF-ac, FF-NH$_2$, CAM and vehicle control (DMSO) in a cell-free translation inhibition assay. All curves are shown as mean ± s.e.m. (s.d. for enzymatic and translation assays), *n* = 3 biological replicates. All *M. abscessus* strains are derivatives of ATCC19977. Panel **a** created with BioRender.com.

both timepoints (FDR < 0.01; Fig. 3d,e), with Eis2 induction rising from log₂FC of 1.45 at 0.5 h to 3.4 at 3 h. Together, these transcriptomic and proteomic studies demonstrate that FF-NH₂ induces its activating enzyme, Eis2, in a time-dependent manner.

We next aimed to test whether the accumulation of active metabolite FF-ac also occurred over time compared with FF by using a MS-based accumulation/conversion assay. *M. abscessus* WT cells were exposed to 100 μM FF or FF-NH₂ for up to 3 h, and cell lysates

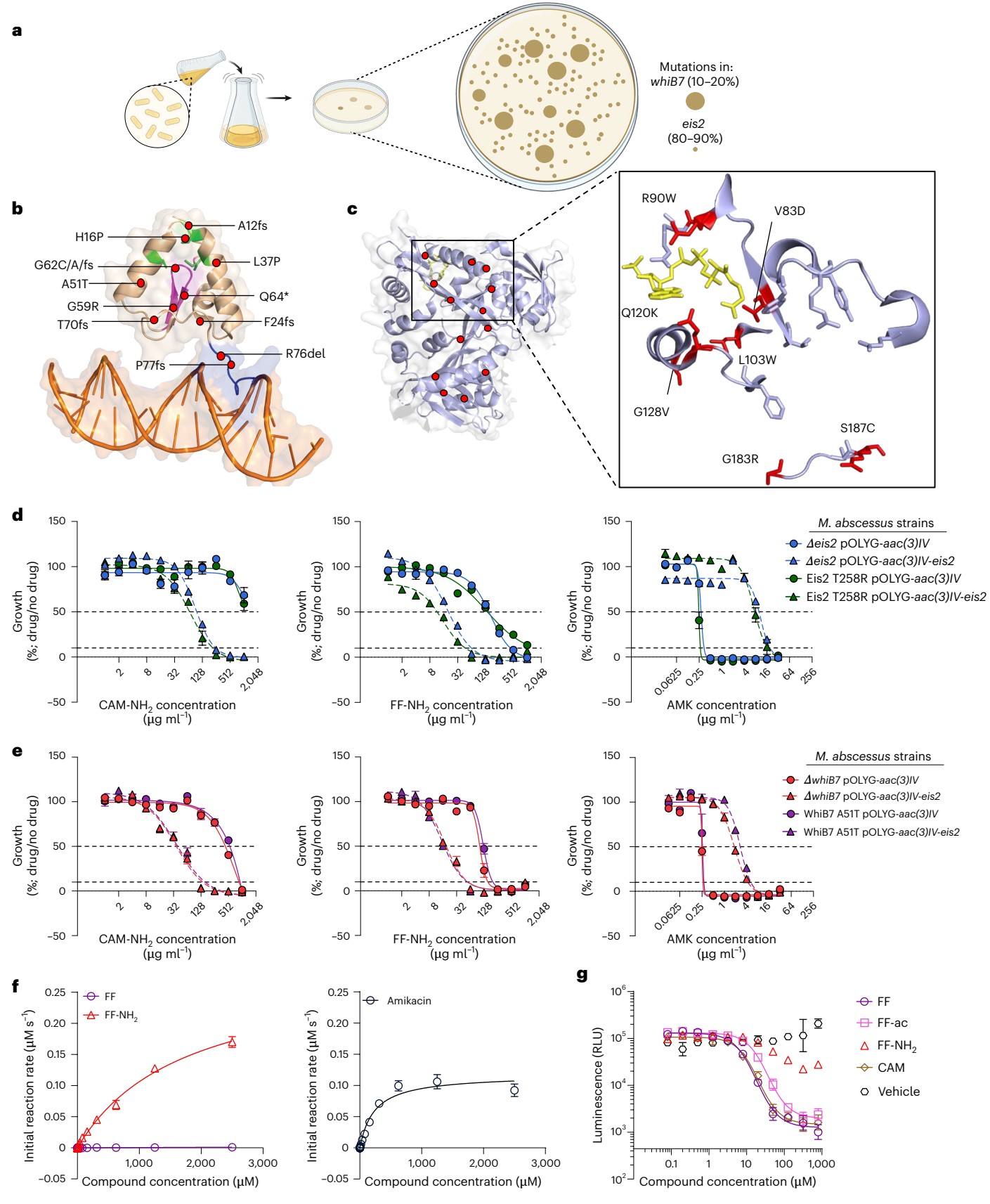

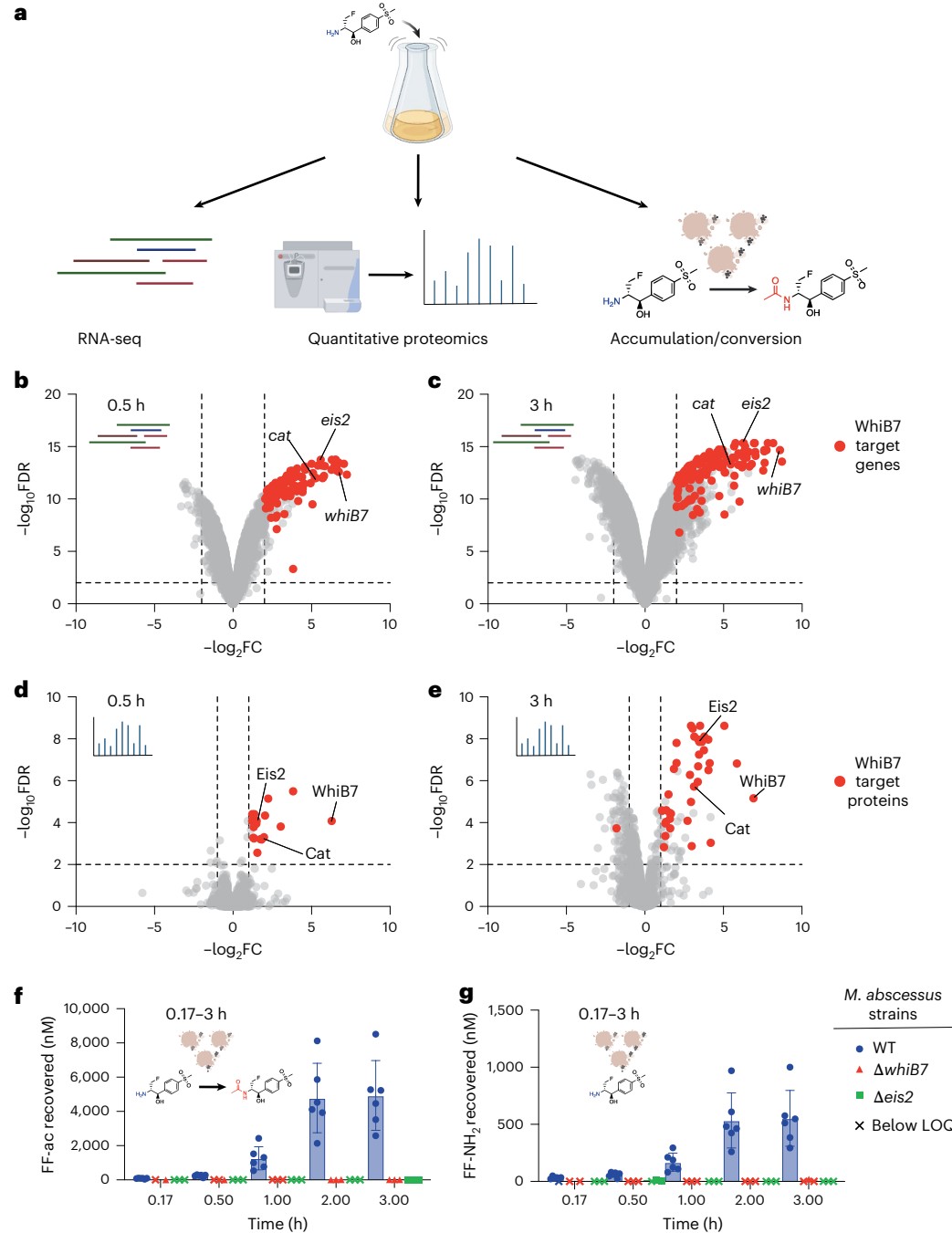

**Fig. 3 | FF-NH₂ induces the expression of Eis2, drives intracellular accumulation of active metabolite FF-ac. a**, A schematic for the mechanism of action platform using 100 µM FF-NH₂. **b,c**, Volcano plots of differentially expressed genes (log₂FC >|2|, −log₁₀FDR <0.01) in *M. abscessus* upon exposure to 100 µM of FF-NH₂ for 0.5 h (**b**) and 3 h (**c**). **d,e**, Volcano plots of differentially expressed proteins (log₂FC >|1|, −log₁₀FDR <0.01) in *M. abscessus* upon exposure 100 µM of FF-NH₂ for 0.5 h (**d**) and 3 h (**e**). **f,g**, The intracellular accumulation of FF-ac (**f**) and FF-NH₂ (**g**) in *M. abscessus* cells up to 3 h after 100 µM FF-NH₂ treatment. In **b**−**e**, data are from three biological replicates. Direct and indirect targets of WhiB7 (ref. 25) are highlighted in red. In **f** and **g**, each point is an independent biological experiment: *M. abscessus* WT (*n* = 6), *M. abscessus* Δ*eis2* (*n* = 3) and Δ*whiB7* (*n* = 3 except Δ*whiB7* at 0.17 h, *n* = 2). Mean ± s.d. shown. LOQ, limit of quantification. All *M. abscessus* strains are derivatives of ATCC19977. Components of this figure created with BioRender.com.

were analysed for FF, FF-NH₂ and FF-ac. We find that accumulation of FF occurs rapidly (within 10 min) and remains consistent (180–320 nM recovered) throughout the 3 h experiment (Extended Data Fig. 6a). In contrast, FF-ac levels from FF-NH₂ treatment increased over time, starting at 83.5 ± 25 nM recovered at 10 min and rising to as high as 4,900 ± 2,000 nM at 3 h (Fig. 3f). FF-NH₂ concentrations also showed a modest increase in accumulation throughout the experiment (Fig. 3g). This could result from drug-induced changes in uptake processes or from the conversion of FF-ac back to FF-NH₂, either through biological

mechanisms (for example, enzymatic deacetylation) or technical factors during MS analysis (for example, in-source fragmentation). Regardless, these observations reinforce the central finding that FF-ac is the primary accumulated species in *M. abscessus* cells. We next assessed the accumulation of FF-ac from FF-NH₂ in the absence of this activation mechanism by testing the Δ*whiB7* and Δ*eis2* strains. We observed minimal accumulation for either species, with most samples falling below the limit of quantification across the duration of the experiment (Fig. 3f,g). To further verify the role of Eis2 in the

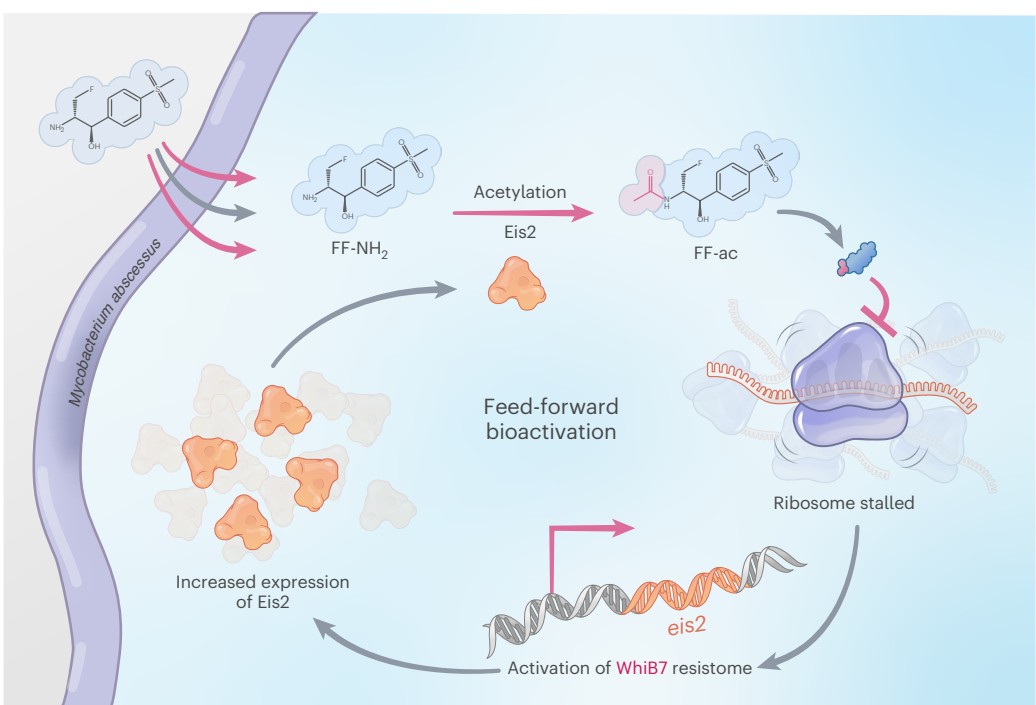

**Fig. 4 | FF-NH₂ displays a unique feed-forward mechanism of action that exploits intrinsic resistance.** A schematic of the mechanism of action of FF-NH₂ in *M. abscessus*. FF-NH₂ is acetylated by Eis2 to form FF-ac (acetyl group shown in pink), a more potent translational inhibitor. Increased ribosomal inhibition induces WhiB7 regulon genes, including *eis2*. Elevated Eis2 expression leads to further FF-NH₂ acetylation, resulting in greater FF-ac accumulation and establishing a self-reinforcing feed-forward loop.

accumulation of FF-ac in *M. abscessus* cells, we repeated the experiment using the Δ*eis2* strain complemented with the pOLYG-*aac(3)IV-eis2* or pOLYG-*aac(3)IV* control vectors. The addition of pOLYG-*aac(3)IV-eis2* vector restored the time-dependent accumulation of FF-ac and FF-NH₂, while the control vector had no effect (Extended Data Fig. 6b).

These results highlight the unique mechanism of action displayed by FF-NH₂, where the Eis2-mediated conversion of FF-NH₂ to FF-ac drives antimicrobial activity by generating a more active ribosome inhibitor, thereby inducing further Eis2 expression through WhiB7. This conversion not only enhances the antimicrobial potency, but also facilitates the accumulation of FF-ac within the cell, creating a perpetual feed-forward loop resulting in the amplified antimicrobial action of FF-NH₂ (Fig. 4).

## FF-NH₂ has narrow-spectrum activity

FF-NH₂ requires acetylation for activity, leading us to hypothesize that only species expressing highly similar versions of Eis2 would be susceptible. We tested FF and FF-NH₂ against various reference and recombinant mycobacterium species, as well as *E. coli* and *Staphylococcus aureus*. We found that FF-NH₂ demonstrated reasonable MIC activity against only *M. chelonae* (MIC of 8 µg ml⁻¹), which possesses a version of Eis2 that is 84.7% similar to Eis2 in *M. abscessus* (Table 1). Of note, FF-NH₂ was 32-fold more potent in *M. chelonae* compared with FF (MIC of 256 µg ml⁻¹) (Table 1). All other strains tested, whose closest orthologues display a maximum of 35% homology to Eis2, showed increased MIC values for FF-NH₂ compared with FF. This includes *M. smegmatis*, which had an FF-NH₂ MIC of 512 µg ml⁻¹ (Table 1). Heterologous expression of *eis2* in *M. smegmatis* using the pOLYG-*aac(3)IV-eis2* vector improved the activity of FF-NH₂ and decreased the activity of AMK compared with the pOLYG-*aac(3)IV* control vector, whereas susceptibility of neither FF nor LZD control changed (Extended Data Fig. 7a–d).

Additionally, we conducted MIC testing on a panel of 34 rapidly growing NTM clinical isolates, including drug-resistant strains

(Supplementary Table 5). This panel comprised members of the *M. abscessus–chelonae* clade (*M. abscessus* subsp. *abscessus*, *M. abscessus* subsp. *massiliense*, *M. abscessus* subsp. *bolletti*, *M. chelonae* and *Mycobacterium saopaulense*), species that are more distantly related but still possess an Eis2 homologue (*Mycobacterium mucogenicum* and *Mycobacterium cosmeticum*) and species having poor (<35%) Eis2 homology (*Mycobacterium fortuitum, Mycobacterium peregrinum* and *Mycobacterium farcinogenes*). The *M. abscessus–chelonae* clinical isolates displayed increased sensitivity for FF-NH₂ when compared with FF, displaying FF/FF-NH₂ MIC ratios of ≥4, with MIC values ranging from 16 µg ml⁻¹ for *M. chelonae* to 64 µg ml⁻¹ for *M. abscessus* subsp. *abscessus* (Supplementary Table 5). Although *M. mucogenicum* and *M. cosmeticum* possess versions of Eis2 that are 64% similar to *M. abscessus* Eis2, they displayed an FF-NH₂ MIC of ≥256 µg ml⁻¹ and an FF/FF-NH₂ MIC ratio of ≤1. This suggests that the function of Eis2 is influenced by its specific sequence, structure and evolutionary context, enabling homologous proteins to recognize different substrates and perform distinct functions in various organisms[48]. Consequently, species lacking Eis2 homologues predictably showed no FF-NH₂ activity and a low FF/FF-NH₂ MIC ratio (≤0.5; Supplementary Table 5). These results further confirm that efficacy of FF-NH₂ depends on Eis2, which acts as a prodrug activator in *M. asbscessus–chelonae* complex bacteria, highlighting FF-NH₂'s potential for targeted treatments against infections caused by these mycobacteria.

## FF-NH₂ synergizes with anti-*M. abscessus* antibiotics

FF-NH₂ resistant mutants were found to possess mutations in either *whiB7* or *eis2* (Fig. 2a–c), suggesting that resistance to FF-NH₂ could lead to collateral sensitivity to anti-*M. abscessus* therapeutics. To investigate this, we conducted a high-throughput susceptibility screen using the *M. abscessus* WT strain, the defined deletion strains (Δ*whiB7* and Δ*eis2*) and ten unique FF-NH₂ resistant strains (*n* = 5 each with *whiB7* and *eis2* mutations). These strains were tested against a panel of antibiotics,

**Table 1 | FF-NH$_2$ is narrow-spectrum for *M. abscessus–chelonae* complex**

| Strain | Closest Eis2* orthologue (%) | MIC (µg ml$^{-1}$) | | FF/FF-NH$_2$ MIC ratio |
|---|---|---|---|---|
| | | FF | FF-NH$_2$ | |
| *M. abscessus* ATCC19977 | 100 | 128–256 | 64 | ≥2 |
| *M. chelonae* ATCC19536[a] | 84.7 | 256 | 8 | 32 |
| *M. fortuitum* ATCC6841 | 31.0 | 64–128 | 128 | 0.5–1 |
| *M. peregrinum* ATCC700686 | 34.8 | 32 | 128 | 0.25 |
| *M. smegmatis* mc$^2$ 155 | 33.8 | 16 | 512 | <0.03 |
| *M. avium* ATCC25291 | 32.6 | 2 | 32 | 0.06 |
| *M. kansasii* ATCC12478 | 34.5 | 8 | 32–64 | 0.13–0.25 |
| *M. tuberculosis* H37Rv | 35.2 | 64 | >512 | <0.13 |
| *E. coli* BW25113 | 13.0 | 16 | 512 | 0.03 |
| *S. aureus* USA300 FPR3757 | 10.7 | 4–8 | >512 | <0.02 |

MIC for FF and FF-NH$_2$ were performed by broth microdilution according to CLSI guidelines. The FF/FF-NH$_2$ ratio was determined by the MIC of FF divided by the MIC of FF-NH$_2$. *Closest orthologue and percentage were collected by NCBI protein BLAST search of the *M. abscessus* Eis2 amino acid sequence against each bacterial organism. [a]MIC was conducted at 30 °C.

including standard-of-care (SOC) agents and other ribosome-targeting antimicrobials. Changes in susceptibility compared with *M. abscessus* WT (ratio for IC$_{50}$ of *M. abscessus* WT/mutant strain) were analysed.

Broadly, we observed that susceptibility patterns for FF-NH$_2$-resistant strains with mutations in either *whiB7* or *eis2* closely mirrored those of their respective isogenic deletion strain (Fig. 5a). This suggests that these genomic alterations result in loss of function, effectively recapitulating the knockout phenotype. As expected, mutations in either *eis2* or *whiB7* increased the potency of all tested aminoglycosides/peptide antibiotics (AMK, kanamycin A/B, hygromycin B and capreomycin), with up to a 16-fold increase in potency observed for capreomycin (Fig. 5a,b and Extended Data Fig. 8). Additionally, strains harbouring *whiB7* mutations showed increased susceptibility to non-fluorinated phenicols (CAM/TAM) and SPC, whereas strains with *eis2* mutations did not (Fig. 5a,b and Extended Data Fig. 8a–s). This differential susceptibility is attributed to the role of WhiB7 in mediating resistance through Cat for non-fluorinated phenicols and the efflux pump TetV for SPC[29]. Increased activity for macrolides—azithromycin (AZITH), CLR and erythromycin—was not observed in the *whiB7* mutants during the 6 day assay, probably because WhiB7-mediated *erm*(41) resistance requires longer durations to manifest. However, under standard 14 day MIC conditions, macrolide susceptibility improved 4–32-fold in the Δ*whiB7* and WhiB7 A51T strains compared with the WT, Δ*eis2* and Eis2 T258R strains (Supplementary Table 6), aligning with previous studies[23,24,49]. As no mutations were found in the ribosomal target site, these results suggest that resistance to FF-NH$_2$ comes at the cost of collateral susceptibility to SOC antimicrobials, highlighting its 'antiresistance' properties.

Given that *M. abscessus* infections are typically treated using a combination of antimicrobials[11], we sought to determine the potential of pairing FF-NH$_2$ with SOC anti-*M. abscessus* therapeutics through an in vitro high-throughput combination screening platform[50]. Using the response surface method, bivariate response to additive interaction doses (BRAID)[51], we found that FF-NH$_2$ strongly synergized ($\kappa = 2.16$) with LZD, as well as displayed moderate synergy with representative macrolides (AZITH and CLR), and the third generation tetracycline analogue, eravacycline (Fig. 5c). Conversely, FF-NH$_2$ was found to be antagonistic with AMK and cefoxitin (Fig. 5c). The observed synergy between

different classes of protein synthesis inhibitors, excluding aminoglycosides that typically show antagonism with bacteriostatic ribosomal agents, aligns with existing literature[52,53]. However, it is unclear how FF-NH$_2$ and LZD, both presumed to act at the ribosomal PTC[54], display such a high degree of synergy. Nonetheless, at various sub-IC$_{50}$ concentrations, FF-NH$_2$ displayed improved potency and significant synergy with LZD, as well as CLR, as also determined by Bliss independence model[55] (Fig. 5d,e). These findings highlight the potential of FF-NH$_2$ to be effectively combined with existing anti-*M. abscessus* therapies.

### FF-NH$_2$ displays reduced mitochondrial toxicity

The requirement for FF-NH$_2$ to be activated by Eis2 led us to hypothesize that it would lack the mitochondrial protein synthesis (MPS) inhibition liability of other PTC-targeting agents. We therefore profiled FF-NH$_2$, FF-ac and other analogues for MPS inhibition activity. FF-ac exhibited potent MPS inhibition (IC$_{50}$ 4.7 ± 1.7 µM), similar to FF (IC$_{50}$ 1.2 ± 0.2 µM), CAM (IC$_{50}$ 5.5 ± 2.5 µM), TAM (IC$_{50}$ 1.4 ± 0.7 µM) and LZD (IC$_{50}$ 6.2 ± 1.6 µM) (Supplementary Table 7). However, FF-NH$_2$, as well CAM-NH$_2$ and TAM-NH$_2$, displayed markedly decreased activity in this assay (IC$_{50}$ >100 µM; Supplementary Table 7). Further, an in vitro cytotoxicity assay showed FF-NH$_2$ to be non-cytotoxic at the top concentrations tested (IC$_{50}$ >200 µM) (Supplementary Table 7). These results indicate that FF-NH$_2$'s prodrug mechanism may mitigate the toxicity typically associated with PTC-targeting antimicrobials.

This mechanism parallels recent work on 5-aminomethyl oxazolidinone prodrugs, which are also activated by a mycobacterial-specific *N*-acetyltransferase and show limited MPS inhibition[56]. However, those compounds were found to undergo oxidative deamination in vivo, producing toxic 5-alcohol metabolites with potent MPS activity. To assess whether FF-NH$_2$ shares this vulnerability, we evaluated its metabolic stability and metabolite profile in hepatocytes, the primary site of drug metabolism. FF-NH$_2$ demonstrated high stability in human hepatocytes ($t_{1/2}$ > 440 min), with lower stability in rat ($t_{1/2}$ = 131.7 ± 9.0 min) and mouse hepatocytes ($t_{1/2}$ = 92.3 ± 4.4 min) (Supplementary Table 8). Metabolite analysis identified only a reduction product (mouse) and glucuronide conjugate (mouse and rat), with no evidence oxidative deamination or *N*-acetylation across all three species (Extended Data Fig. 9 and Supplementary Table 9), suggesting a reduced risk of toxic metabolite formation associated with FF-NH$_2$.

### FF-NH$_2$ shows efficacy in a *M. abscessus* mouse infection model

To further investigate the drug-like properties of FF-NH$_2$, we determined it's in vivo pharmacokinetic properties across different doses and routes of administration. The results indicate the absolute bioavailability of FF-NH$_2$ was relatively high, ranging between 44% and 66% for oral dosing, and reached 86% for subcutaneous administration, with a linear increase in dose-dependent exposure for $C_{max}$ and higher than linear increase for the area under the concentration–time profile curve (AUC$_{inf}$; Supplementary Table 10). For all routes of administration, FF-NH$_2$ exhibited limited duration of exposure after administration with terminal half-lives ranging between 0.28 and 1.02 h. The rapid clearance of FF-NH$_2$ aligns with the natural metabolic pathway of FF, where conversion to FF-NH$_2$ facilitates elimination in many animal species[43–45].

The promising mechanistic results and lack of toxicity associated with FF-NH$_2$ motivated us to conduct proof-of-concept testing using an in vivo granulocyte macrophage colony stimulating factor (GM-CSF) knock out murine model of acute *M. abscessus* infection[57]. For optimal exposure, FF-NH$_2$ was administered subcutaneously at a dosage of 400 mg kg$^{-1}$ twice daily, which was well tolerated. SOC antibiotics, CLR and LZD, were also included as reference. Following nine consecutive days of therapy, FF-NH$_2$ treatment led to a statistically significant reduction in *M. abscessus* bacterial burden in the lung (1.3 log c.f.u. reduction), liver (1.6 log c.f.u. reduction) and spleen (1.2 log c.f.u. reduction) compared with vehicle control (Fig. 5f), achieving reductions comparable to those of CLR and LZD. This demonstration of in vivo

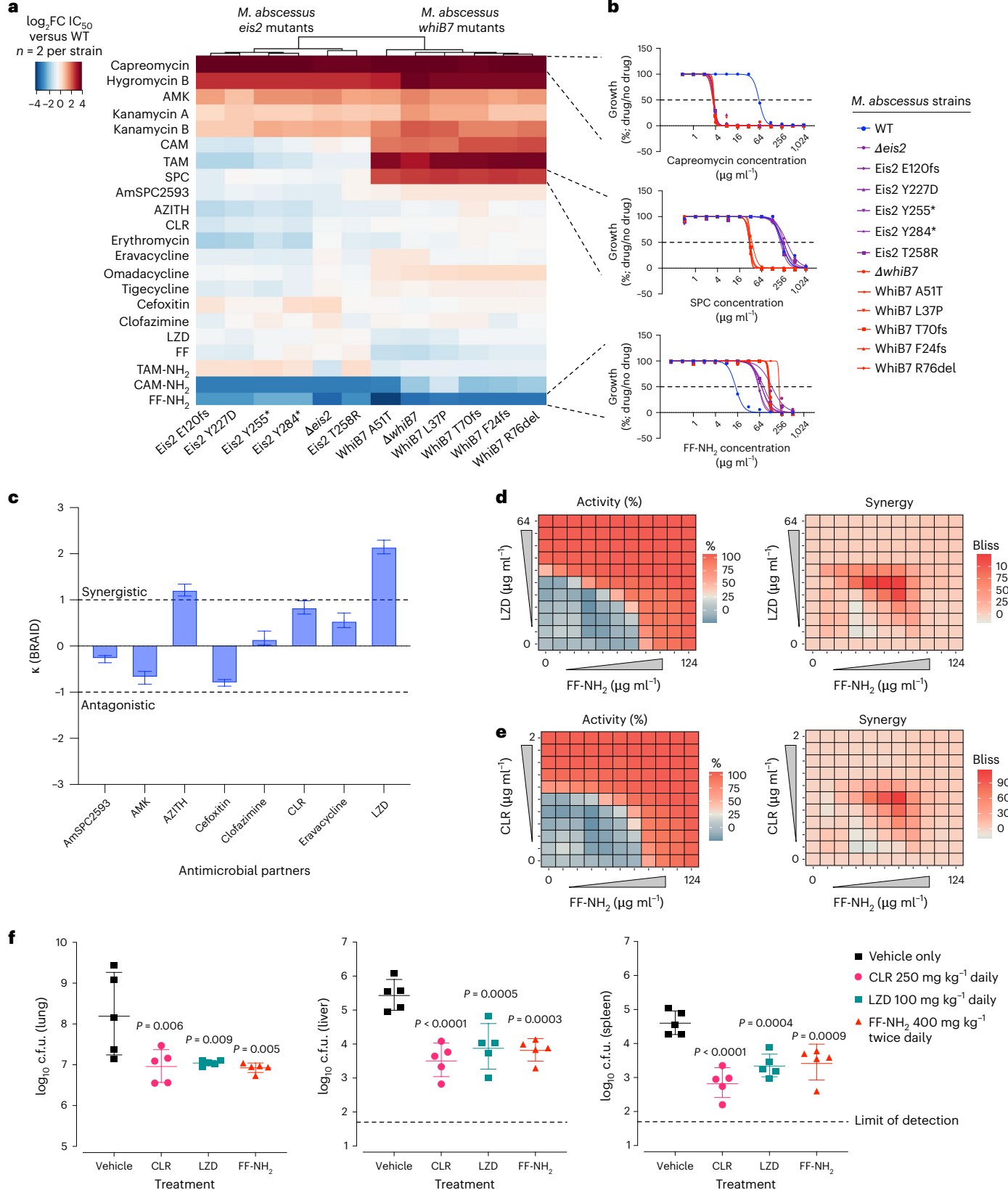

efficacy highlights the translational potential of leveraging intrinsic resistance to develop safe and effective therapies against this pathogen.

## Discussion

Infections caused by *M. abscessus* are notoriously challenging to treat owing to its extensive array of intrinsic resistance mechanisms, many of which are regulated by a single transcription factor WhiB7. In this study, we introduce the prodrug FF-NH₂, a natural metabolite of FF. FF-NH₂ evades inactivation by the WhiB7-depedent *O*-acetyltransferase Cat, while also leveraging the WhiB7-dependent *N*-acetyltransferase Eis2 for activation. This conversion creates a perpetual feed-forward loop that results in an increase in FF-NH₂ antimicrobial activity driven

**Fig. 5 | FF-NH$_2$ demonstrates therapeutic compatibility and in vivo efficacy.**
**a**, A heat map displaying relative differences in antimicrobial susceptibility for *M. abscessus* deletion strains (Δ*whiB7* and Δ*eis2*) and selected FF-NH$_2$-resistant mutants, compared with the WT. Each square is shown as log$_2$FC in IC$_{50}$ (WT/strain). **b**, Representative dose-response curves of *M. abscessus* WT, *whiB7* mutants, and *eis2* mutants against capreomycin, SPC and FF-NH$_2$; shown as mean of *n* = 2 biological replicates per strain. **c**, Summary of drug combination testing of FF-NH$_2$ with antimicrobials against *M. abscessus* WT, with synergy/antagonism displayed as BRAID kappa ($\kappa$) values with confidence intervals. The BRAID model was generated by pooling all biological replicates (*n* = 3) to fit the response surface; $\kappa$ > 0 indicates synergy, $\kappa$ < 0 indicates antagonism, values exceeding ±1 indicate strong interactions and confidence intervals crossing zero indicate non-significant interactions. **d,e**, Heat maps of *M. abscessus* WT cell viability (resazurin assay, left) and bliss score (right) for the FF-NH$_2$ in combination with LZD (**d**) and CLR (**e**). Checkerboard assays are displayed as mean, *n* = 3 biological replicates. **f**, Efficacy of CLR, LZD and FF-NH$_2$ versus saline vehicle in an acute mouse infection model (*n* = 5 mice per group). GM-CSF KO mice were infected with *M. abscessus* WT, treatment began on day 2 and continued for 9 days (CLR/LZD oral daily, FF-NH$_2$ subcutaneous twice daily). Bacterial burden (colony forming units (c.f.u.)) was measured in lung, liver and spleen at day 12 post-infection. Statistical significance was determined by one-way ANOVA with Dunnett post-test where exact *P* values are indicated on the plots. The WT genotype is *M. abscessus* ATCC19977.

by *M. abscessus*'s intrinsic resistance machinery (Fig. 4, mechanism of action summary).

Resistance to FF-NH$_2$ was found to occur through mutations in either *eis2* or *whiB7*, which could be differentiated based on colony morphology (Fig. 2a–c). Although we did not observe it in our study, modifications of the ribosomal target site (*rrl*) probably occur, albeit at a much lower frequency compared with mutations affecting *eis2* and *whiB7*. Importantly, our results suggest that resistance to FF-NH$_2$ could even be advantageous by paving the way for other, established antibiotic treatment regimens, as it results in collateral sensitivity to several classes of antimicrobials, such as macrolides and aminoglycosides, which are considered the cornerstone of current *M. abscessus* treatment (Fig. 4a,b and Supplementary Table 6). Lorè et al. demonstrated this concept in vivo, showing that mice infected with *M. abscessus* Δ*eis2* strain exhibited a 3–4 log c.f.u. reduction in bacterial burden in the lungs when treated with AMK, compared with mice infected with the *M. abscessus* WT strain[58]. This suggests that an alternating treatment regimen of FF-NH$_2$ followed by AMK could be a viable therapeutic strategy, as bacteria that develop resistance to FF-NH$_2$ through *eis2* loss-of-function mutations become more susceptible to bactericidal agents such as AMK. Similarly, a study by Li et al. further demonstrated the in vivo potential of collateral sensitivity by highlighting the treatability of a *whiB7*-deficient lineage (L1.2.1) of *M. tuberculosis* found in Southeast Asia using CLR[49]. These studies bridge the gap between our in vitro data and the in vivo potential, implying that resistance to FF-NH$_2$ or future analogues could be beneficial from a therapeutic standpoint by making other antimicrobials substantially more effective.

Regarding drug modification, much attention has been focused on the detrimental impact of drug-modifying enzymes; however, biotransformation can also be beneficial. Several examples of prodrug activation in *M. tuberculosis* have been described, including the oxidation of isoniazid by KatG[59] and ethionamide by EthA[60], the hydrolysis of pyrazinamide by PncA[61] and the reduction of pretomanid by Ddn[62]. Recently, Boshoff and colleagues also demonstrated acetylation-based activation mediated by Rv0133 in *M. tuberculosis*, which activates the PTC-targeting 5-aminomethyl oxazolidinones[56]. Unique to FF-NH$_2$, however, is its ability to induce the expression of its activating enzyme, Eis2 (Fig. 3b–e). Traditionally, the biotransformation function of specific enzymes towards antibiotics has been viewed as beneficial or detrimental, but not both[63]. This study illustrates that a resistance mechanism for one antibiotic can be harnessed to enhance the efficacy of another.

A limitation of this study is the potency of FF-NH$_2$, which leaves room for improvement (*M. abscessus*–*chelonae* complex MIC of 8–64 µg ml$^{-1}$). However, FF-NH$_2$ is non-cytotoxic and lacks MPS inhibition (Supplementary Table 7), a toxicity limitation that has previously restricted the use of PTC-targeting agents. This suggests that higher doses of FF-NH$_2$ could be administered without associated liabilities, as evidenced by its tolerability at 400 mg kg$^{-1}$ twice daily in mice. The development of next-generation phenicol prodrugs with improved potency and pharmacokinetic properties is currently ongoing, which holds promise for enhancing therapeutic outcomes against *M. abscessus* infections.

As antimicrobial resistance (AMR) continues to rise globally, uncovering innovative strategies to address this challenge is essential. This study provides proof-of-concept for a promising approach to combat AMR, not merely by overcoming drug resistance but by leveraging resistance mechanisms to target drug-resistant pathogens. Eis2 is one of over 272,000 bacterial GNAT proteins, a class directly implicated in AMR[64]. While acetylation is a common mechanism of modification-based resistance, other processes—including phosphorylation, adenylation and hydroxylation—also play important roles in AMR[65]. We hypothesize that, in the right circumstances, some of these enzymes could be harnessed for drug activation and antibiotic discovery. Given the escalating threat of AMR, future research should focus on identifying further opportunities to exploit microbial resistance mechanisms for therapeutic innovation.

## Methods

### Bacterial strains
*M. abscessus* strains are derivates of *M. abscessus* ATCC19977 unless otherwise noted. All other reference strains were purchased from commercial sources (ATCC) and described below.

**Clinical NTM isolates.** Clinical isolates from primary patient samples or obtained as cultures from local laboratories were cultivated and differentiated by the Institute of Medical Microbiology, University of Zurich/National Reference Laboratory for Mycobacteria by target PCR amplification and subsequent sequencing of 16S rRNA gene (*rrs*), RNA polymerase β subunit gene (*rpoB*), ribosomal methyltransferase (*erm*(41)) and heat shock protein 65 promotor (*hsp65*) when needed. Targeted sequencing of 23S rRNA gene (*rrl*) and *rrs* was applied for genotypic confirmation of constitutive macrolide and pan-aminoglycoside resistance, respectively.

### Bacterial cell culture
*E. coli* (BW25113), *S. aureus* subsp. *aureus* (USA300_FPR3757), *M. fortuitum* (ATCC6841), *M. peregrinum* (ATCC700686), *M. smegmatis* (mc$^2$155) and *M. abscessus* strains were maintained at 37 °C on cation-adjusted Mueller Hinton (CAMH) II agar (BBL) or broth (BBL). *M. abscessus* Δ*eis2* and *M. smegmatis* strains expressing vector pOLYG-*aac(3)IV* were supplemented with 50 µg ml$^{-1}$ apramycin (Apr) sulfate (MCE). *M. chelonae* (ATCC19536) and *M. chelonae* clinical isolate strains were grown at 30 °C in MH agar or broth. *M. avium* (ATCC25291) and *M. kansasii* (ATCC12478) were grown at 37 °C on MH agar or broth supplemented with 10% Middlebrook OADC (HIMEDIA). *M. tuberculosis* (H37Rv) was cultured at 37 °C with 5% CO$_2$ in Middlebrook 7H9 broth (Difco Laboratories) supplemented with 10% albumin–dextrose complex and 0.05% (v/v) Tween 80 (Millipore), pH adjusted to 7.4. All work with *M. tuberculosis* H37Rv was conducted in a certified BSL-3 facility in accordance with institutional and national biosafety regulations.

### Antibacterial activity measurements
**Standard antimicrobial testing.** Compounds and antibiotics were dissolved in DMSO, other than aminoglycosides (AMK, kanamycin A/B

and hygromycin B) and the peptide antibiotic capreomycin, which were dissolved in water. Compounds were serially diluted 1:2 in a 96-well polypropylene plate (Greiner, 651201) and 4 µl was transferred to 96 µl of CAMHB in standard 96-well round-bottom plates using a Biomek FXP liquid-handing robot (Beckman Coulter). Four growth control wells and cell death control (AMK 128 µg ml$^{-1}$) wells were included for each plate. Bacterial cultures were grown to late-log phase and back-diluted to an $OD_{600}$ of 0.001 (0.01 for *M. tuberculosis*) in CLSI recommended media and temperatures described above. Subsequently, 100 µl of each culture was plated onto the pre-drugged plates. Non-mycobacterial species were incubated for 20 h. Rapidly growing mycobacteria were incubated for 5 days, whereas *M. avium* and *M. kansassii* were incubated for 14 days. *M. tuberculosis* was incubated for 7 days. MICs were read by visual inspection and recorded (range of MIC values) as the lowest concentration of compound that prevented visible growth of at least two biologically independent replicates. Strain identifiers are provided within each table and figure.

**IC$_{50}$ testing.** For dose-response curves, assay plates were prepared as described above and *M. abscessus* strains were incubated for 5 days at 37 °C after which plates were removed and 20 µl of 0.77 mM resazurin (final concentration 70 µM) was added to each well and incubated for an additional 24 h. For *M. smegmatis* strains, plates were incubated for 48 h at 37 °C before the addition of resazurin for the final 24 h incubation. After incubation, plates were sealed with light-absorbing film (AbsorbMax, 202537) and fluorescent intensity (540/590) was measured using a PHERAstar FS Multilabel reader (BMG). Raw fluorescent intensity values were normalized to growth and cell death controls to estimate the percent growth. Susceptibility testing was performed in triplicate and IC$_{50}$ values (in µg ml$^{-1}$) for each compound were calculated in GraphPad Prism version 10.2.0 by using the equation (inhibitor) versus normalized response – variable slope.

**High-throughput susceptibility assays.** High-throughput susceptibility assays were performed as described above with the following changes. Compounds were dissolved and serially diluted 1:2 in a 384-well Echo PP source plate (Labcyte, PP-0200) to generate a 12-point concentration range. Using an Echo 655 Acoustic Liquid Handler (Beckman), 400 nl from each well of this source plate were then transferred to a 384-well black, clear-bottom plate (ThermoFisher Scientific, 142761). *M. abscessus* strains were diluted to an $OD_{600}$ of 0.0005 in CAMH and 40 µl was delivered to each well of 384-well assay plate containing compound using a Multidrop Combi (Thermo Fisher Scientific) and incubated for 5 days at 37 °C. Then 4 µl of 0.77 mM resazurin (final concentration 70 µM) was added to each well and incubated for an additional 24 h. After incubation, plates were sealed and fluorescent intensity was read as described above. Raw fluorescent intensity values were normalized to growth (maximum set to 100% growth) and cell death controls to calculate the percent growth. Susceptibility testing was performed in duplicate and IC$_{50}$ values (µg ml$^{-1}$) for each compound were calculated in GraphPad Prism version 10.2.0 by using the equation (inhibitor) versus normalized response – variable slope. A ratio of IC$_{50}$ values was calculated by dividing the *M. abscessus* WT IC$_{50}$ values by that of the different mutant strains and the log$_2$ transformation of this ratio was displayed as a heat map using the gplots function heatmap.2 in the R statistical computing environment.

**Drug combination testing.** Drug combinations were assessed utilizing a previously described high-throughput combination platform (combocat.stjude.org)[50]. Inoculum, DMSO concentration, timing and assay readout were consistent as in 'High-throughput susceptibility assays' section. Synergy and antagonism were determined using both the BLISS independence model[50,55] and BRAID[51]. For BRAID, kappa ($\kappa$) values and confidence intervals were recorded and interpreted as $\kappa > 0$ are synergistic and $\kappa < 0$ are antagonistic with values further from 0 representing

stronger interaction profiles (increase for synergy, decrease for antagonism). Error bars crossing $\kappa = 0$ indicate a non-statistically significant interaction. All checkerboard experiments were performed in triplicate and heat maps of concentration matrices of the mean of normalized cell activity and bliss scores are displayed. BRAID models for each antibiotic combination are provided as Supplementary Information.

**Agar spotting and passaging assays.** *M. abscessus* strains were grown from single colonies in MHB(II) + 0.05% tyloxapol without antibiotics to an $OD_{600}$ of 0.4–0.8 and tenfold serial dilutions were spotted on antibiotic-free MHA(II) or MHA(II) containing 64 µg ml$^{-1}$ of FF-NH$_2$ and incubated at 37 °C. *M. abscessus* Δ*whiB7* and *M. abscessus* WhiB7 A51T mutant were grown from single colonies in MHB(II) + 0.05% tyloxapol to an $OD_{600}$ of 0.4–0.8, serial diluted 10$^{-6}$ and 50 µl were plated on antibiotic-free MHA(II) or MHA(II) containing 64 µg ml$^{-1}$ of FF-NH$_2$ and incubated at 37 °C. A large, rough colony from MHA(II) + 64 µg ml$^{-1}$ was selected for each strain and passaged in antibiotic-free media and plated as before.

### Selection and sequencing of FF-NH$_2$-resistant *M. abscessus* mutants
Resistant mutants were raised against FF-NH$_2$ by plating 100 µl of a saturated *M. abscessus* ATCC19977 culture (OD >0.8) on CAMHA plates containing 64, 128 or 256 µg ml$^{-1}$ of FF-NH$_2$. Plates were incubated for 10 days at 37 °C, at which 22 colonies were selected and grown in 5 ml of CAMHB + 0.05% tyloxapol with identical concentrations of FF-NH$_2$ for 3–4 days at 37 °C. Genomic DNA (gDNA) from the culture was extracted using the DNeasy Blood and Tissue Kit (Qiagen, 69504) according to the manufacturer's directions on gDNA extraction of Gram-positive bacteria. Purified gDNA were then sequenced on an Illumina NovaSeq X Plus sequencer at SeqCenter. Variants from the reference *M. abscessus* ATCC 19977 genome (genome accession: NC_010397) were detected using the CLC genomics workbench. Additional experiments were subsequently performed, where Sanger sequencing was performed on purified PCR amplicons of *whiB7* and *eis2* using the primers listed in Supplementary Table 11. All whole-genome sequencing reads associated with this study have been deposited to National Center for Biotechnology Information (NCBI) Sequence Read Archive (SRA) under accession number PRJNA1141985.

### Genetic disruption and complementation in Mycobacterial species
A 1.5 kbp PCR fragment from position 3047952 to 3049451 (5′ *cat* flanking sequence) and a 1.5 kbp PCR fragment from position 3050112 to 3051611 (3′ *cat* flanking sequence) were amplified from *M. abscessus* gDNA using primers P1/P2 and P3/P4 (Supplementary Table 11) and then ligated via Gibson assembly[66] into the allelic replacement vector pKH_aac(3)IVDsRed2_KatG[67] previously digested with MunI and PvuI. The resulting suicide vector pKH_aac(3)IVDsRed2_KatG_MAB2989 was introduced into *M. abscessus* via electroporation, and the expression of DsRed2 and aprR indicated plasmid integration. Subsequently, for counter selection, colonies that were both AprR and DsRed2 positive were plated on isoniazid-containing medium (32 µg ml$^{-1}$). Only clones that had undergone intramolecular homologous recombination can survive on isoniazid plates since KatG activates the prodrug isoniazid[68]. Using this counter-selection technique, either deletion mutant strains or WT (revertant) strains were selected. The putative deletion mutants were screened via PCR and confirmed using Southern blot and whole-genome sequencing analysis. Southern blot analysis with the 5′ cat probe amplified from *M. abscessus* gDNA using primers P7 and P8 was used to verify recombination events (Supplementary Fig. 1).

For complementation of the Δ*cat* strain, the 1,221-bp PCR fragment of the *MAB_2989* region including its native promotor region was amplified from *M. abscessus* gDNA with primers P5 and P6 and inserted into the KpnI linearized pMV361-*aac(3)IV* vector, thereby generating

pMV361-*aac(3)IV*-MAB_2989. Integration of the complementation vector in the *M. abscessus* Δ*cat* genome was confirmed by PCR screening and whole-genome sequencing (Supplementary Fig. 2).

For overexpression of *eis2* (*MAB_4532c*) and *cat* (MAB_2989), genes were amplified from *M. abscessus* gDNA using primers P11/P12 and P13/P14 respectively (Supplementary Table 11). Amplified fragments were inserted into the HindIII-digested multicopy vector pOLYG-aac(3)IV using Gibson assembly[66]. Constructed pOLYG-*aac(3)IV-eis2* was transformed into *M. abscessus* Δ*eis2*, *M. abscessus* Δ*whiB7*, *M. abscessus* Eis2 T258R, *M. abscessus* WhiB7A51T and *M. smegmatis* mc² 155 by electroporation while pOLYG-aac(3)IV-cat transformed only into *M. smegmatis* mc² 155 (see Supplementary Table 11 for the full list of strains used in this study). Clones were selected on LB agar plates containing Apr (50 µg ml⁻¹), and the putative *eis2* and *cat* overexpressed recombinant strains were confirmed by PCR screening using pOLYG-aac(3)IV backbone primers P19/20 (Supplementary Figs. 3 and 4).

### Structural mapping of mutation sites
Structures of WhiB7 from *M. tuberculosis* (PDB: 7KUF) and Eis2 from *M. abscessus* (PDB: 6RFT) were obtained from the Protein Data Bank (PDB). For Fig. 2b, residues in *M. tuberculosis* WhiB7 corresponding to mutation sites identified in our *M. abscessus* isolates were mapped onto the structure based on structural alignment using Foldseek (root mean squared deviation of 0.93 Å)[69]. For Fig. 2c, mutation sites were mapped directly onto one subunit of the *M. abscessus* Eis2 structure. All structural figures were prepared using PyMOL (Schrödinger LLC).

### Eis2 purification and enzymatic assay
**Eis2 purification.** The Eis2 protein was purified as previously described with minor modifications[47]. In brief, the expression construct (Eis2–pET30a), which encodes for Eis2 containing a TEV cleavable N-terminal 6 histidine tag, was transformed into *E. coli* BL21 (DE3) cells. The large-scale cultures were grown at 37 °C in LB media containing kanamycin 50 µg ml⁻¹. Cells were induced by adding 1 mM isopropyl-β-ᴅ-1-thiogalactopyranoside, and the cell pellet was collected after 16 h of induction at 18 °C. The resulting cell pellet was resuspended in the lysis buffer with 50 mᴍ Tris–HCl (pH 8), 500 mM NaCl, 0.5 mM imidazole, 5 mM β-mercaptoethanol (β-Me) and 1 mM benzamidine. The cell suspension was incubated with DNase I and lysozyme on ice for 30 min and lysed using an emulsifier. The clarified supernatant was subjected to a 5 ml metal affinity cobalt column equilibrated with lysis buffer. The unbound protein was washed by passing 25 column volumes of lysis buffer. Eis2 was eluted in a gradient manner by using an elution buffer consisting with 50 mᴍ Tris–HCl (pH 8), 500 mM NaCl, 150 mM imidazole, 5 mM β-Me and 1 mM benzamidine. Eluted Eis2 was dialysed overnight in a buffer of 25 mᴍ Tris–HCl (pH 8), 300 mM NaCl, 5% glycerol, 5 mM β-Me and 5 mM EDTA, as well as TEV protease (1 mg:40 mg ratio) to facilitate N-terminal polyhistidine tag cleavage. The resulting protein solution was subjected to a 5 ml metal affinity cobalt column equilibrated with a size-exclusion buffer (20 mᴍ Tris–HCl pH 7.4, 200 mM NaCl and 0.5 mM TCEP) to collect the removed his tag. Eis2 was further purified by using a Sephacryl S-300 HR size-exclusion column and purity of the protein was estimated by using SDS–PAGE gel.

**Eis2 enzymatic assay.** The rate of Eis2 acetylation of experimental compounds and controls was determined through the measurement of free thiols in solution based on previously established methods but optimized for a 384-well format[47]. Assay buffer (50 mM Tris and 200 mM NaCl, pH 8.0) alongside a 5 ml solution of 0.1 µM Eis2 and 1 mM acetyl-CoA was incubated for at least 5 min at 8 °C. Additionally, a 50 ml solution of 4 mM 5,5′-dithiobis-(2-nitrobenzoic acid) (DTNB) in the assay buffer was prepared. Compounds were brought into a 10 mM solution in the DTNB buffer, followed by serial dilution and transfer to an LDV Echo Plate (Beckman, LP-0200) with subsequent 1 µl transfer to

the assay plate using an Echo 650 acoustic liquid handler (Beckman). Active wells received 1 µl of 4 mM DTNB buffer, followed by 2 µl of the protein acetyl-CoA solution, and were shaken on a combi shaker before centrifugation for 30 s at 500*g*. Absorbance readings at 412 nm were then measured using a PHERAstar FS Multilabel reader (BMG) for 30 min at 25 °C. Subsequent data analysis involved converting absorbance readings to micromolar substrate concentration using a pre-established linear range equation and calculating the reaction rate in each well in micromoles of sulfhydryl (SH) group per second. GraphPad Prism software (v10.2.0) was utilized for data plotting and determination of Michaelis constant ($K_M$) and maximum reaction rate ($V_{max}$) for each ligand. All experiments were conducted in at least triplicate and results are presented as mean ± s.d.

### Cell-free Mycobacterial translation inhibition assay
The bacterial *M. smegmatis* S30 extract was prepared as described previously[70]. The test compounds FF-NH₂, FF and CAM were dissolved in DMSO and AMK was dissolved in deionized water. Using an I-DOT dispenser (Dispendix), compounds or equivalent amount of control solvent were added to a white 96-well plate (Eppendorf) in concentrations ranging between 800 and 0.8 µM. On ice, 13.5 µl translation master mix (4 µl S30 extract, 0.2 mM amino acid mix, 6 µg *E. coli* tRNA (Sigma), 0.4 µg firefly luciferase mRNA, 0.3 µl protease inhibitor (cOmplete, EDTA-free, Roche), 12 U RNAse inhibitor (Ribolock, Thermo Scientific) and 6 µl S30 premix without amino acids (Promega)) was dispensed to each well. The plates were sealed with transparent foil and incubated for 60 min at 37 °C. The reaction was stopped by placing the plates on ice for 5 min. Afterwards, 75 µl of luciferase assay reagent (Promega) was dispensed to each well and luminescence was recorded using a BioTek Synergy H1 plate reader. Regression analysis and absolute IC₅₀ determination on relative light units (RLU) were performed in GraphPad Prism version 10.2.3, using preset equations for absolute IC₅₀ calculation. The mean log₁₀(RLU) of the fully inhibiting AMK concentrations were used as the baseline constraint. Experiments were performed in triplicate and plotted in GraphPad Prism (version 10.2.3).

### Total RNA extraction, RNA-seq and analysis
**Treatment and RNA extraction.** Five millilitres of mid-log *M. abscessus* cultures (OD₆₀₀ 0.4–0.6) were treated with ½MIC of CAM (256 µg ml⁻¹) for 30 min, 25 µM of florfenicol (FF) for 30 min and 3 h, 25 or 100 µM of FF-NH₂ for 30 min and 3 h, or equivalent vehicle control (DMSO) for 30 min or 3 h shaking at 225 rpm at 37 °C. After treatment, 5 ml of Bacterial RNAprotect (Qiagen) was added for 5 min at room temperature. The culture was then centrifuged at 3,214*g* for 10 min at 4 °C and the supernatant discarded. An RNAeasy Plus Mini kit (Qiagen, 74134) was used to extract the RNA following the manufacturer's instructions utilizing mechanical lysis for Gram-positive bacteria.

**RNA-seq.** Libraries were prepared from ribosomal RNA-depleted total RNA using the Illumina Stranded Total RNA Prep ligation with Ribo-Zero Plus mRNA Library Prep kit according to the manufacturer's instructions (Illumina, PN 20037135). Paired-end 150 cycle sequencing was performed on a NovaSeq X Plus (Illumina) at SeqCenter. The resulting reads were saved in FASTQ format and data were processed through the High-Performance Computing Facility at St. Jude. Adaptor content and quality trimming was performed using Cutadapt[71]. The quality of the raw and trimmed reads was assessed using MultiQC[72]. The resulting processed reads were aligned to the reference *M. abscessus* ATCC19977 (Genome accession NC_010397) using STAR2.7.1a[73] and aligned reads were counted using featureCounts[74]. Statistical testing was conducted using *t* statistics modified to incorporate gene strength into variance estimation, leveraging Limma[75] and Voom[76] packages alongside edgeR[77] in the R statistical environment. The FDR values (*q* values) are derived from Limma. All RNA-seq data shown are representative of three biological replicates and the raw reads have been deposited to NCBI Gene

Expression Omnibus (GEO) under accession number GSE273574 and processed data are provided as Supplementary Table 12.

### Data-independent acquisition MS

**Cell treatment and extraction.** Cells were grown and treated with 100 μM of FF-NH$_2$ under the same conditions as described under the RNA-seq experiment with the exception that 15 ml was used. After treatment, cells were centrifuged at 3,214$g$ for 10 min at 4 °C and the supernatant discarded. Pellets were washed three times with chilled PBS and resuspended in 500 μl of lysis buffer (8 M urea, 10 mM Tris–HCl (pH 8.0), 30 mM NaCl and 10 mM iodoacetamide) and 100 μl of 0.1 mm silicone spheres (BioSpec, 11079101Z) were added. Each sample was then subjected to 12 cycles of bead beating using the Bead Mill four homogenizer (Fisher Scientific) at maximum speed for 25 s followed by 5 min on ice. Lysates were pelleted at 14,500$g$ for 2 min at 4 °C and 400 μl of the lysate supernatant was collected.

**Sample processing protocol.** For analysis, 10 μg of lysate in 50 mM HEPES (pH 8.5), 8 M urea buffer was used. The lysates were digested with Lys-C (Lys-C:protein ratio of 1:10) for 30 min at room temperature. Following Lys-C digestion, the samples were diluted to 2 M urea with 50 mM HEPES and further digested with trypsin (Trypsin:protein ratio of 1:10) for 3 h at room temperature. The digests were acidified with 5% formic acid and 10% of the digests were loaded on Evotips following the manufacturer's recommendations. The samples were analysed on a Bruker timsTOF HT instrument coupled to EvosepOne (Evosep Biosystems) using the pre-programmed 30 samples per day method, which has a 44 min gradient and a cycle time of 48 min. Evosep endurance column (8 cm × 100 μm, C18, 3 μm particle size) was used for peptide separation. Data were acquired using the dia-PASEF mode with an MS1 scan range of 100–1,700 $m/z$. Twenty-one isocratic $m/z$ and ion mobility windows were selected for serial MS2 fragmentation ranging from 475 to 1,000 $m/z$ and 0.85 to 1.27 $1/K_0$ respectively.

**Data processing protocol.** The raw timsTOF data (.d folders) were imported into DIA-NN (version 1.8.2beta8) and analysed with the library-free mode. *M. abscessus* ATCC19977 sequences were downloaded from the UniProt database (taxonomy id: 561007; version 2024/12/17). The protein FASTA file was then used for in silico library generation with the following settings: Trypsin/P with maximum 2 missed cleavage; oxidation on methionine as variable modification; maximum number of variable modifications set to 2; peptide length from 7 to 30; precursor charge 2–4; precursor $m/z$ from 400 to 1,200; fragment $m/z$ from 200 to 1,800. The search parameters of DIA-NN were set as follows: precursor FDR 1%; mass accuracy at MS1 and MS2 were set to 0 (automatic inference); scan window set to 0 (automatic inference); isotopologues and MBR (match-between-run) turned on; protein inference at gene level; heuristic protein inference enabled; quantification strategy set to any LC; cross-run normalization turned off; speed and RAM usage set to optimal results. The search results were further filtered with precursor $q$ value <0.01 and protein group $q$ value <0.01 at the library level. For quantification, DIA-NN first obtained precursor quantities by summing the intensities of the top six fragments (ranked by their library intensities) for each precursor. Then protein-level intensities are set as the intensities of the most abundant precursors identified with $q$ value <0.01. Cross-run normalization was then performed based on the median protein intensity of each sample. Raw mass spectra from proteomics experiment have been deposited to proteomeXchange and MassIVE repositories with identifiers PXD059834 and MSV000096854.

**Differential expression analysis.** Protein-level intensities of all identified proteins were summarized in a table and input into our in-house quantification platform, JUMPshiny. Two pairwise comparisons were performed in this step: FFA-30 (FF-NH$_2$) versus DMSO-30 and FFA-180 versus DMSO-180. JUMPshiny first transformed the raw protein intensities into a log$_2$ scale, then Limma was used for differential expression analysis[75]. The output table from JUMPshiny contains the original expression data, log$_2$FC, $P$ values and FDR. The processed data are provided as Supplementary Table 13.

### *M. abscessus* accumulation/conversion assay

**Treatment.** *M. abscessus* strains were grown from single colonies to late-log phase and back-diluted to an OD$_{600}$ of 0.0025 in 150 ml of CAMHB + 0.05% tyloxapol and incubated for 48 h at 37 °C with shaking at 225 rpm. Once cultures reach a mid-log density (OD$_{600}$ 0.4–0.6), 150 μl of 100 mM FF or FF-NH$_2$ solubilized in DMSO were added directly to the cultures. At set intervals (10 min, 30 min, 1 h, 2 h and 3 h), 25 ml of culture was removed and centrifuged for 8 min at 4,000$g$. The pellet was then washed three times with 10 ml ice-cold PBS containing 0.05% tyloxapol. Cell pellets were lysed by resuspending in 300 μl of 100% acetonitrile and 100 μl of 0.1 mm silicone spheres (BioSpec, 11079101Z) were added. Each sample was then subjected to three cycles of bead beating using the Bead Mill four homogenizer (Fisher Scientific) at maximum speed for 60 s followed by 3 min on ice. Lysates were pelleted at 14,500$g$ for 2 min at 4 °C and 200 μl of the lysate supernatant was collected and added to 200 μl of deionized water. The mixtures were then filtered using 1.5 ml centrifugal 0.22 μm filter tubes (Millipore, UFC30GVNB) before being analysed by Echo MS.

**Analysis.** To prepared samples for analysis, 25 nl of 1 mM internal standards were added to each well of a 384-well Echo PP source plate (Labcyte, PP-0200) using an Echo 655 Acoustic Liquid Handler (Beckman), resulting in a final internal standard concentration of 500 nM after sample or standard curve (diluted in mock-treated *M. abscessus* lysate) was added to each well. Warfarin (CAS 81-81-2) was used as the internal standard for samples measure in the positive ion mode and propylparaben (CAS 94-13-3) for the negative ion mode. The source microplates containing the sample cosolvent were centrifuged (537$g$ for 5 min) to remove gas bubbles and ensure a consistent fluid meniscus.

For sample collection, the Sciex OS-MQ Analytics Software (version 3.3.10, Sciex) controlled the Echo MS system operating in multiple reaction monitoring mode. The Echo MS set-up included an external transducer assembly from an Echo MS autosampler, an open port interface (OPI) linked to a carrier solvent pump and a transfer capillary leading to the IonDrive Turbo V electrospray ionization (ESI) source of an AB Sciex Triple Quad 6500+ system. The carrier liquid, consisting of methanol with 1 mM ammonium fluoride, flowed at 400 μl min$^{-1}$ to create a stable vortex at the OPI inlet, optimizing signal performance. The contactless sampling method involved ejecting 20 nl directly from the microtitre plate wells containing sample into the carrier liquid vortex of the OPI, with sampling occurring every 2 s. The ESI source of the triple quadrupole MS instrument operated in positive ionization mode for warfarin, FF-NH$_2$ and FF-ac, and negative ionization mode for propylparaben and FF. The nebulizer gas (GS1) was set to 90 psi, the heater gas (GS2) to 70 psi, the curtain gas to 35 psi for positive mode and 20 psi for negative mode, and the collision-activated dissociation gas to 9 units. Mass transitions for analysis were 248.18 → 130.13 for FF-NH$_2$, 290.12 → 131.14 for FF-ac, 309.1 → 163 for warfarin, 355.84 → 335.9 for FF and 178.99 → 91.11 for propylparaben. MS parameters included an ion source temperature of 500 °C, a dwell time of 20 ms, a pause time of 5 ms and unit resolution for Q1 and Q3. Analyte-specific settings for the Q1 to Q3 transitions were FF-NH$_2$ (DP 80, EP 10, CE 34, CXP 8), FF-ac (DP 91, EP 10, CE 35, CXP 8), warfarin (DP 57, EP 10, CE 44, CXP 20), FF (DP −115, EP −10, CE −14, CXP −15) and propylparaben (DP −80, EP −10, CE −32, CXP −11). Data processing for acquired sample batches used signal integration parameters with a minimum signal-to-noise threshold of 2. The expected retention time was set to a minimum of 0.02 min. Accumulation concentrations of unknown compounds were determined using calibration curve based on the area ratio (compound/

internal standard) of known concentrations dissolved in lysate from mock-treated samples. Accumulation experiments were performed in biological triplicate for those using *M. abscessus* Δ*eis2* and *M. abscessus* Δ*whiB7* strains, except for 0.167 h timepoint for *M. abscessus* Δ*whiB7* ($n$ = 2) and the *M. abscessus* WT strain ($n$ = 6). All values were plotted as mean and s.d. in GraphPad Prism v10.2.0 (GraphPad Software). All raw and processed data, including chromatograms for total ion and exact masses for FF, FF-NH$_2$, FF-ac, warfarin and propylparaben, are provided as Supplementary Table 14 and Source data.

## MPS inhibition and cytotoxicity assays

MPS and cytotoxicity of experimental compounds and PTC-targeting controls against HepG2 (liver hepatocellular carcinoma cells ATCC HB-8065) were performed as previously described without any modifications (see ref. 56 for MPS assay and ref. 78 for cytotoxicity assay). The concentration of test compounds that inhibited MPS or HepG2 growth by 50% (IC$_{50}$ value) was computed using nonlinear regression-based fitting of inhibition curves using [inhibitor] versus response-variable slope model in GraphPad Prism version 10.2.0 (GraphPad Software). Results reported as mean ± s.d., $n$ = 3 biological replicate.

## Metabolic stability and metabolite identification for FF-NH$_2$ in cryopreserved hepatocytes

**Metabolic stability.** The metabolic stability assay was performed in human, mouse and rat hepatocytes as described by Cai and Shalom, with minor modifications[79]. Briefly, the assay incubation system consisted of $2 \times 10^6$ cryopreserved hepatocytes (BIOIVT) suspended in incubation media (BIOIVT) and 1 μg ml$^{-1}$ drug concentration (FF-NH$_2$). Two standard positive controls were tested in each run: Verapamil and 7-hydroxycoumarin (Sigma-Aldrich). The incubation was carried out in a Cytometry 2 incubator (Thermo Scientific), maintained at 37 °C with 5% CO$_2$. Samples of 50 μl were collected at 0, 30, 60, 120, 180 and 240 min after initiation of the reaction. After collection, the samples were quenched by adding 150 μl of acetonitrile containing 100 ng ml$^{-1}$ SPC as internal standard. Samples were centrifuged at 4,000 rpm at 4 °C using a filtration plate and filtrate was collected into an analysis plate. Sample quantification and analysis were performed using an AB Sciex 5500 triple quadrupole mass spectrometer (Sciex). Species (human, rat and mouse) matrix cell count, timepoint and incubation volume were used for the calculation of metabolic stability half-life and hepatocyte intrinsic clearance parameters. In the following equations, $k$ is the slope of the log-transformed regression of the percent of compound remaining over time, $t_{1/2}$ is the corresponding stability half-life and CL$_{int}$ is the hepatic intrinsic clearance

$$t_{1/2} \, (\text{min}) = \frac{0.693}{k} \tag{1}$$

$$\text{CL}_{int} \, (\mu l/min/10^6 \, \text{cells}) = \frac{V \times k}{N} \tag{2}$$

where $V$ represents the incubation volume (μl) and $N$ represents the number of hepatocytes ($10^6$ cells) in the incubation.

The metabolic stability half-lives and hepatocyte intrinsic clearance values for FF-NH$_2$ and the positive controls, verapamil and 7-hydroxycoumarin, in cryopreserved hepatocytes of the investigated three species, human, mouse and rat, are provided in Supplementary Table 8.

**Metabolite identification.** For the metabolite identification assay, 50 μl of supernatant from the previously described experiment was collected and quenched with an equal volume of acetonitrile and analysed using a Xevo G2-S quadrupole time-of-flight (Q-TOF) mass spectrometer (Waters Corporation) to elucidate the structure of metabolites based on scan differences between samples at 0 min to later timepoints.

The fragmentation profiles from the Q-TOF MS analysis, detailing the detected metabolites of FF-NH$_2$, are provided in Supplementary Table 9. After hepatocyte incubation, the major metabolites identified were a reduction product and a glucuronide conjugate for mouse hepatocytes, and a glucuronide conjugate for rat hepatocytes (Extended Data Fig. 9).

## In vivo pharmacokinetics of FF-NH$_2$ in mice

Mouse pharmacokinetic studies were performed in BALB/C mice (~20 g body weight; Charles River) at four different doses: 10 mg kg$^{-1}$ administered intravenously by tail vein injection, 30 mg kg$^{-1}$ administered by oral gavage, 300 mg kg$^{-1}$ administered by oral gavage and 400 mg kg$^{-1}$ administered by subcutaneous injection. The drug was formulated in PlasmaLyte A injection (Multiple Electrolytes injection type I USP, Baxter Healthcare) and sterile water (9:1 for low doses and 1:1 for high doses). The prespecified sampling timepoints for intravenous and subcutaneous dosing were at 0.08, 0.25, 0.5, 1, 2, 4, 8 and 24 h post dose and those for oral gavage dosing were at 0.25, 0.5, 1, 2, 4, 6, 8 and 24 h post dose. Groups of three mice were used for each timepoint with destructive sampling. Blood was collected at each sampling timepoint by cardiac puncture in euthanized animals and was subsequently centrifuged in a lithium–heparin plasma separation tube (Microtainer, Becton Dickinson) to collect plasma. Plasma samples were stored at −70 °C until bioanalysis. Plasma samples were extracted using 8 volumes of methanol with internal standard (SPC, 250 ng ml$^{-1}$). Drug concentrations in the extracts were quantified using an AB Sciex 5500 triple quadrupole mass spectrometer (Sciex).

The pharmacokinetic analysis of the obtained plasma concentration–time courses was performed by non-compartmental methodology using Phoenix WinNonlin 6.4 (Certara) and the obtained parameters are presented in Supplementary Table 10. The AUC$_{inf}$ was calculated by the log-linear trapezoidal rule and was extrapolated to infinity by addition of the value $C_{last}/\lambda_Z$, where $C_{last}$ is the concentration at the last sampling timepoint and $\lambda_Z$ is the terminal slope of the log concentration–time curve determined by linear regression. The terminal half-life $t_{1/2}$ is 0.693 divided by $\lambda_Z$. The terms $C_{max}$ and $t_{max}$ represent the maximum concentration achieved and the time to maximum concentration, respectively, and were determined from the concentration–time profiles. Absolute bioavailability $F$ for the oral gavage and subcutaneous administration were determined by calculating the dose normalized ratios of AUC$_{inf}$ after oral gavage or subcutaneous administration relative to intravenous administration.

All pharmacokinetic animal studies were conducted in accordance with the Animal Welfare Act and the Public Health Service Policy on Humane Care and Use of Laboratory Animals. Before initiation, the respective animal protocol was approved by the Institutional Animal Care and Use Committee (IACUC no. 24-0590) of the University of Tennessee Health Science Center.

## *M. abscessus* GM-CSF KO mouse infection model

**Inoculation.** Female 6–9-week-old GM-CSF KO mice (strain 026812, B6.129S-Csf2tm1Mlg/J) were purchased from the Jackson Laboratory and housed in a BSL-3 facility at CSU, with free access to food and water. Mice were infected via the aerosol route using a Glas-Col apparatus, with a high dose of *M. abscessus* ATCC19977 smooth morphotype. Lung seeding was confirmed in three mice by plating lung homogenates on 7H11 agar plates and enumerating CFUs 4–5 days after incubating at 37 °C. Two days post-infection and before initiating antibiotic treatment, baseline bacterial burden in lungs, liver and spleen was determined in three mice. Thereafter, antibiotic treatment was initiated as described below and continued for nine consecutive days. Finally, bacterial burden in lungs, liver and spleen was enumerated at day 12 post-infection.

**Compound treatment.** FF-NH$_2$ was aliquoted as dry powder for each day of treatment. Vials were resuspended in sterile saline solution

every morning and applied by subcutaneous injection twice daily at 400 mg kg$^{-1}$. CLR (Sigma-Aldrich) was resuspended in sterile water plus 0.5% carboxymethylcellulose and 0.5% Tween80 (both from Sigma-Aldrich). LZD (MedChemExpress) was resuspended in sterile water plus 5% PEG-200 (Sigma-Aldrich) and 0.5% carboxymethylcellulose. LZD and CLR were prepared weekly and dosed once daily via oral gavage at 100 mg kg$^{-1}$ and 250 mg kg$^{-1}$, respectively. Efficacy experimentation in mice was approved by CSU's IACUC committee, protocol no. 5157.

### Reporting summary

Further information on research design is available in the Nature Portfolio Reporting Summary linked to this article.

## Data availability

Whole-genome sequencing reads associated with this study have been submitted to the NCBI SRA under accession number PRJNA1141985. Raw RNA-seq data reads have been deposited in the NCBI GEO under accession number GSE273574. Raw mass spectra from proteomics experiment have been deposited to proteomeXchange and MassIVE repositories with identifiers PXD059834 and MSV000096854. All data supporting the findings of this study are provided within the Article and its Supplementary Information, extended datasets and source data files. Source data are provided with this paper.

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

## Acknowledgements

This research was supported by funding from the National institutes of Health (grant no. R01AI157312 to R.E.L.), NIH Ruth L. Kirschstein National Research Service Award (grant no. F31AI169961 to G.A.P.), and the American Lebanese Syrian Associated Charities (ALSAC), St. Jude Children's Research Hospital. Work in the laboratory of P.S. is supported by Swiss National Science Foundation (grant no. 310030_197699/1), Joint Program Initiative Antimicrobial Resistance (JPIAMR) ACOMa (grant no. 2022-050), the Federal Office of Public Health (FOPH) (grant no. 3632001500), Cystic Fibrosis Switzerland (CFS) and the Institute of Medical Microbiology. L.B. was supported by the Swiss National Science Foundation (grant no. 320030 215557). S.K. was supported by Student Abroad Program of the Republic of Türkiye Ministry of National Education. We thank P. Selchow for expert technical assistance. We thank M. Rice of SJCRH for scientific illustration assistance in generating Fig. 4.

## Author contributions

G.A.P., R.E.L. and P.S. conceived the research. G.A.P., S.K., R.E.L., P.S., J.O., A.A.H., B.S., J.W.R., P.G., S.N.H., L.B., B.M. and A.O.-H. contributed to the design of the study. G.A.P., S.K., C.W.T., W.C.W., R.B.L., A.K.W., A.I. and O.G.-C. performed antimicrobial susceptibility and drug combination testing against reference strains. S.K. conducted susceptibility testing against clinical isolates, designed plasmid constructs and generated isogenic *M. abscessus* strains. A.R.J. and S.M.A. performed the chemistry. G.A.P. and C.W.T. generated spontaneous-resistant *M. abscessus* mutants. B.W. and F.K.B. performed in vitro translation assays. T.D.J. and E.C.G. carried out enzymatic biochemical assays. G.A.P. conducted transcriptomic analyses and analysed all sequencing data. G.A.P., V.R.P., L.W., S.B., Y.F. and Z.-F.Y. performed and analysed proteomics experiments. G.A.P., C.W.T. and L.Y. conducted accumulation assays. G.A.P. and V.L. performed mitotoxicity and cytotoxicity studies. A. Srivastava, A. Singh, B.T. and H.P. conducted pharmacokinetic and metabolite identification studies. D.K.C. and B.T.T. performed in vivo murine experiments. G.A.P., S.K., R.E.L. and P.S. wrote the manuscript. All authors contributed to data analysis and review of the article.

## Competing interests

R.E.L., G.A.P., A.R.J. and S.M.A. disclose intellectual property associated with this compound series (US Patent application serial no. 63/635,720). The other authors declare no competing interests.

## Additional information

**Extended data** is available for this paper at https://doi.org/10.1038/s41564-025-02147-9.

**Correspondence and requests for materials** should be addressed to Peter Sander or Richard E. Lee.

Gregory A. Phelps [1,2,13], Sinem Kurt [3,13], Alexander R. Jenner[1,4], Shelby M. Anderson[1], Thalina D. Jayasinghe[1], Elizabeth C. Griffith[1], Carl W. Thompson[1], Lei Yang[1], Basil Wicki [5], Frederick K. Bright [5], Victoria Loudon[1], William C. Wright [6], Ashish Srivastava [4], Amarinder Singh[4], Bhargavi Thalluri[4], Hyunseo Park[4], Robin B. Lee[1], Anna K. Wright [1], Oliver Grant-Chapman [2], Daryl K. Conner [7], Brennen T. Troyer [7], Amy Iverson[8], Jason Ochoado[1], Vishwajeeth R. Pagala[9], Long Wu[9], Stephanie Byrum[9], Yingxue Fu[9], Zu-Fei Yuan[9], Anthony A. High [9], Bettina Schulthess[3,10], Jason W. Rosch [8], Paul Geeleher[6], Sven N. Hobbie [11], Lucas Boeck [5,12], Bernd Meibohm [4], Andres Obregon-Henao[7], Peter Sander [3,10] ✉ & Richard E. Lee [1] ✉

[1]Department of Chemical Biology and Therapeutics, St. Jude Children's Research Hospital, Memphis, TN, USA. [2]Graduate School of Biomedical Sciences, St. Jude Children's Research Hospital, Memphis, TN, USA. [3]Institute of Medical Microbiology, University of Zurich, Zurich, Switzerland. [4]Department of Pharmaceutical Sciences, University of Tennessee Health Science Center, Memphis, TN, USA. [5]Department of Biomedicine, University of Basel, Basel, Switzerland. [6]Department of Computational Biology, St. Jude Children's Research Hospital, Memphis, TN, USA. [7]NTM Center, Mycobacteria Research Laboratory, Department of Microbiology, Immunology, and Pathology, Colorado State University, Fort Collins, CO, USA. [8]Department of Host-Microbe

Interactions, St. Jude Children's Research Hospital, Memphis, TN, USA. [9]Center for Proteomics and Metabolomics, St. Jude Children's Research Hospital, Memphis, TN, USA. [10]National Reference Center for Mycobacteria, Zurich, Switzerland. [11]Division of Clinical Bacteriology and Mycology, University Hospital Basel, Basel, Switzerland. [12]Pulmonary Medicine, University Hospital Basel, Basel, Switzerland. [13]These authors contributed equally: Gregory A. Phelps, Sinem Kurt. ✉e-mail: psander@imm.uzh.ch; Richard.Lee@StJude.org

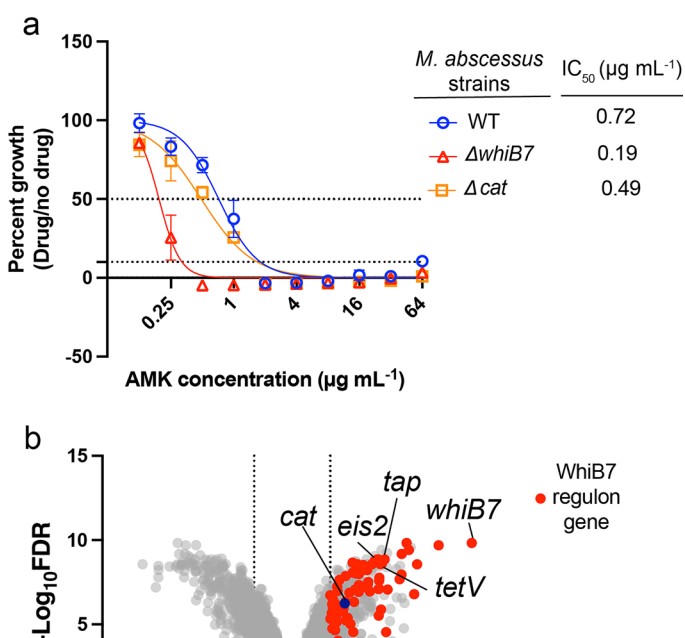

**Extended Data Fig. 1 | Chloramphenicol susceptibility is mediated by a WhiB7-responsive element. a**, Dose-response curves and IC$_{50}$ values (µg mL$^{-1}$) for amikacin (AMK) against *M. abscessus* WT, *M. abscessus* Δ*whiB7*, and *M. abscessus* Δ*cat*. **b**, Volcano plot of differentially expressed genes (L2FC > |2|, -log10 FDR < 0.01) in *M. abscessus* ATCC19977 upon exposure to 256 µg mL-1 of chloramphenicol (CAM) for 30 min. All dose-response curves in **a** shown mean ± s.e.m. from *n* = 3 biological replicates. Volcano plot in **b** is data from three biological replicates; direct and indirect targets of WhiB7 (ref. 25) highlighted in red. *MAB_2989* (*cat*), a WhiB7 regulon gene, is highlighted in blue.

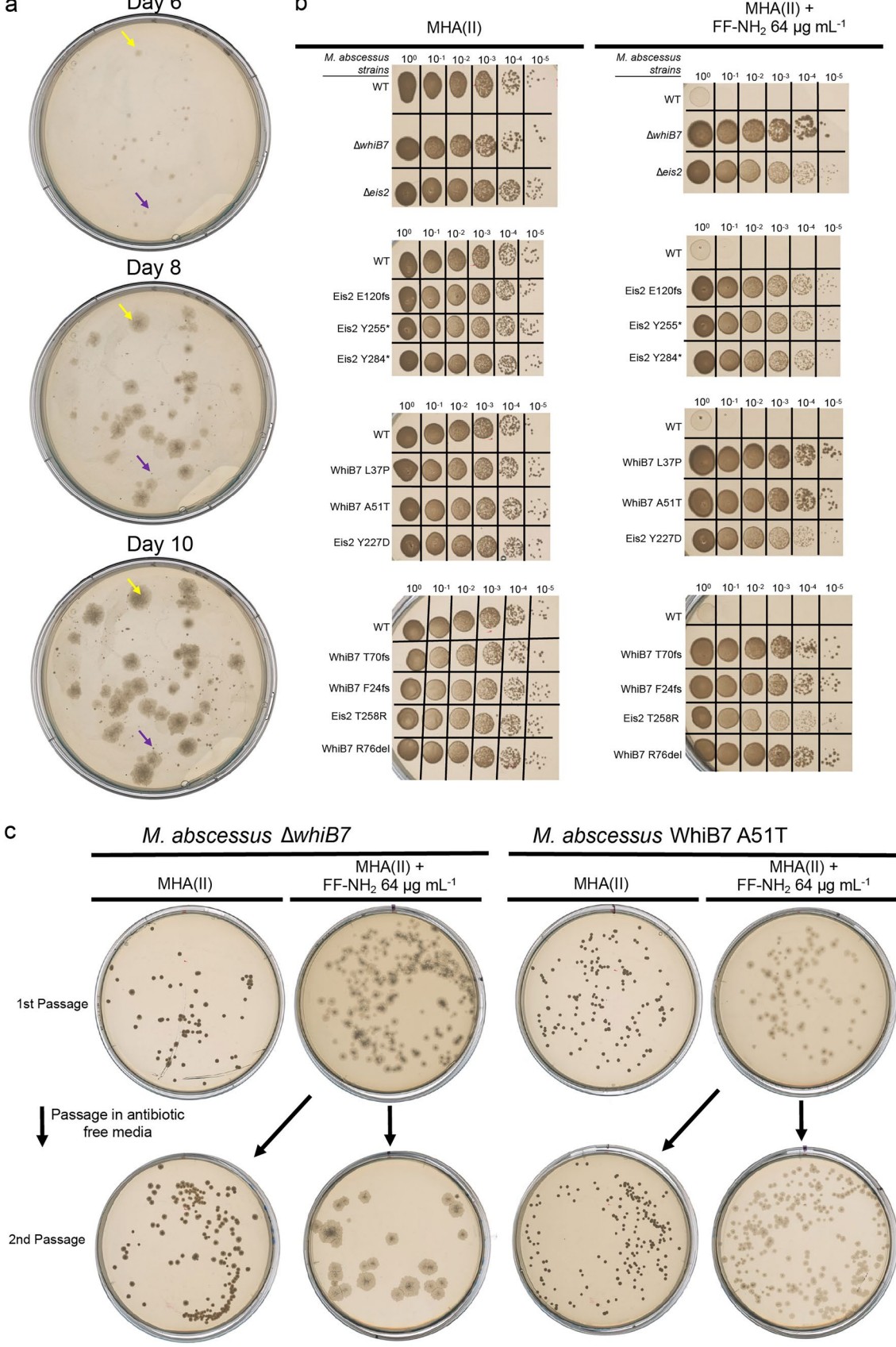

**Extended Data Fig. 2 | See next page for caption.**

**Extended Data Fig. 2 | Colony morphologies of florfenicol amine resistant *M. abscessus* strains. a**, Raw images for a single selection plate containing 64 µg mL$^{-1}$ of florfenicol amine (FF-NH$_2$) taken on day 6, 8, and 10 after plating. Yellow arrows track the position of a single *M. abscessus* colony containing a mutation in *whiB7*. Purple arrows track the position of single *M. abscessus* colony containing a mutation in *eis2*. **b**, Differences in FF-NH$_2$ sensitivities and morphologies of *M. abscessus* WT, *M. abscessus* Δ*whiB7*, *M. abscessus* Δ*eis2*, and ten spontaneously generated FF-NH$_2$ mutants. Strains were grown from single colonies in MHB(II) + 0.05% tyloxapol to an OD$_{600}$ of 0.4-0.8 and ten-fold serial dilutions were spotted on antibiotic-free MHA(II) (left) or MHA(II) containing 64 µg mL$^{-1}$ of FF-NH$_2$ (right). Colonies with mutations in *eis2* or *whiB7* could be differentiated based on colony size in plates containing FF-NH$_2$. Colonies grew

similarly in the antibiotic-free condition indicating no changes in growth rate/ colony morphology were observed in the absence of FF-NH$_2$. **c**, *M. abscessus* Δ*whiB7* (bottom left) and *M. abscessus* WhiB7 A51T mutant (bottom right) were grown in MHB(II) + 0.05% tyloxapol to an OD$_{600}$ of 0.4-0.8, serial diluted 10$^{-6}$, and 50 µL were plated on antibiotic-free MHA(II) (left side of each panel) or MHA(II) containing 64 µg mL$^{-1}$ of FF-NH$_2$ (right side of each panel). A large, rough colony from the MHA(II) + 64 µg mL$^{-1}$ of FF-NH$_2$ was selected for each strain and passaged in antibiotic-free media and plated as before. The absence of FF-NH$_2$ reverted the morphology changes to that of the antibiotic free condition further demonstrating the changes associated with *whiB7* mutations were transient upon FF-NH$_2$ exposure. The wild-type (WT) genotype is *M. abscessus* ATCC19977; other strains are derivatives of ATCC19977.

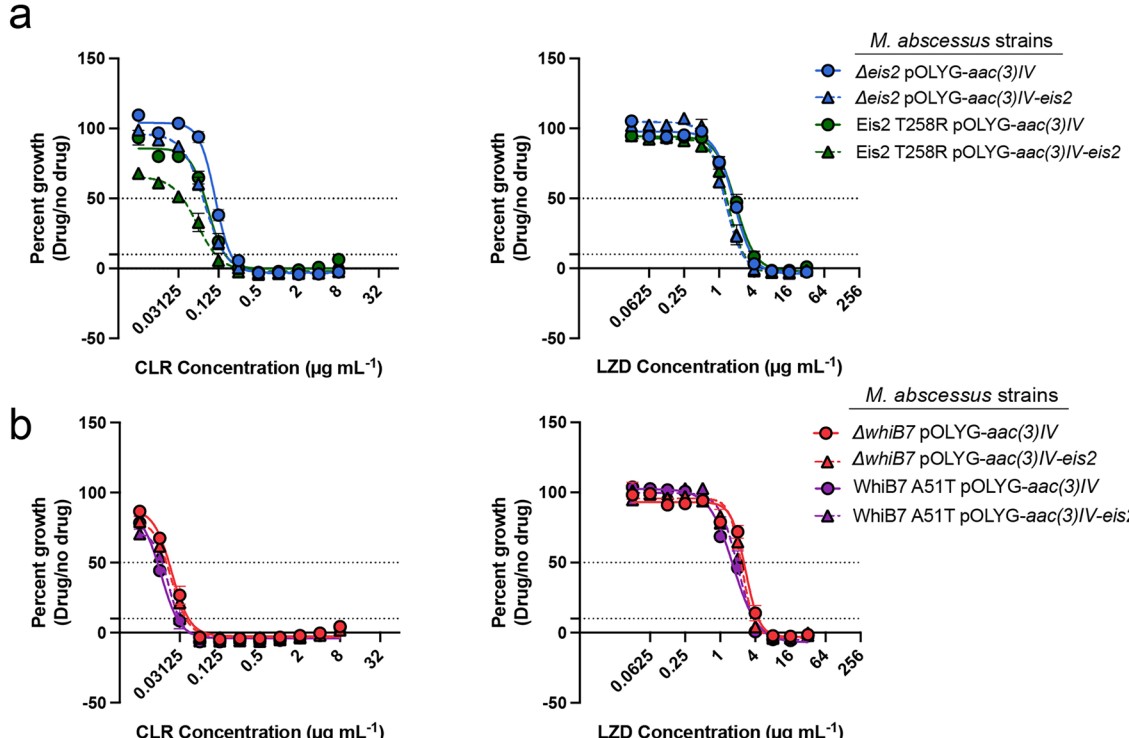

**Extended Data Fig. 3 | Eis2 complementation in *eis2* and *whiB7* deficient *M. abscessus* does not affect clarithromycin or linezolid susceptibility.**
**a**, Dose-response curves for clarithromycin (CLR) and linezolid (LZD) against *M. abscessus* Δ*eis2* or *M. abscessus* Eis2 T258R strains complemented with pOLYG-*aac(3)IV-eis2* overexpression or pOLYG-*aac(3)IV* control vector. **b**, Dose-response

curves for CLR and LZD against *M. abscessus* Δ*whiB7* or *M. abscessus* WhiB7 A51T strains complemented with pOLYG-*aac(3)IV-eis2* overexpression or pOLYG-*aac(3) IV* control vector. Dose-response curves are displayed as mean ± s.e.m., *n* = 3 biological replicates. All *M. abscessus* strains are derivatives of ATCC19977.

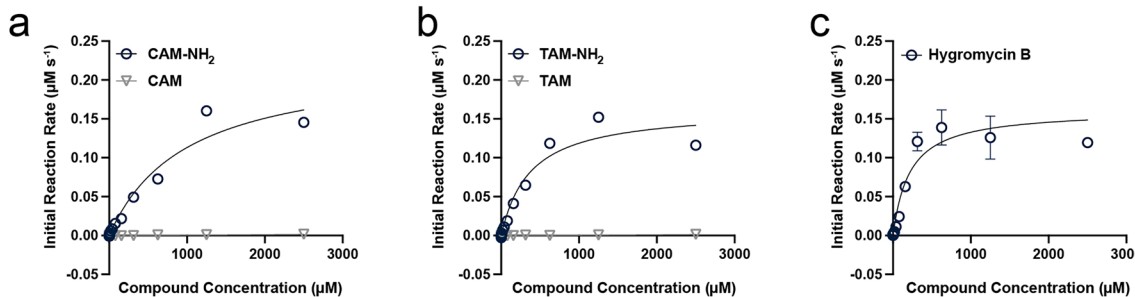

**Extended Data Fig. 4 | Amine phenicols are substrates of Eis2. a**, Michaelis-Menten curves for chloramphenicol (CAM)/CAM amine (CAM-NH$_2$), **b**, thiamphenicol (TAM)/TAM amine (TAM-NH$_2$), and **c**, hygromycin B in Eis2 biochemical assay. All dose-response curves are displayed as mean ± s.d., $n$ = 3 biological replicates.

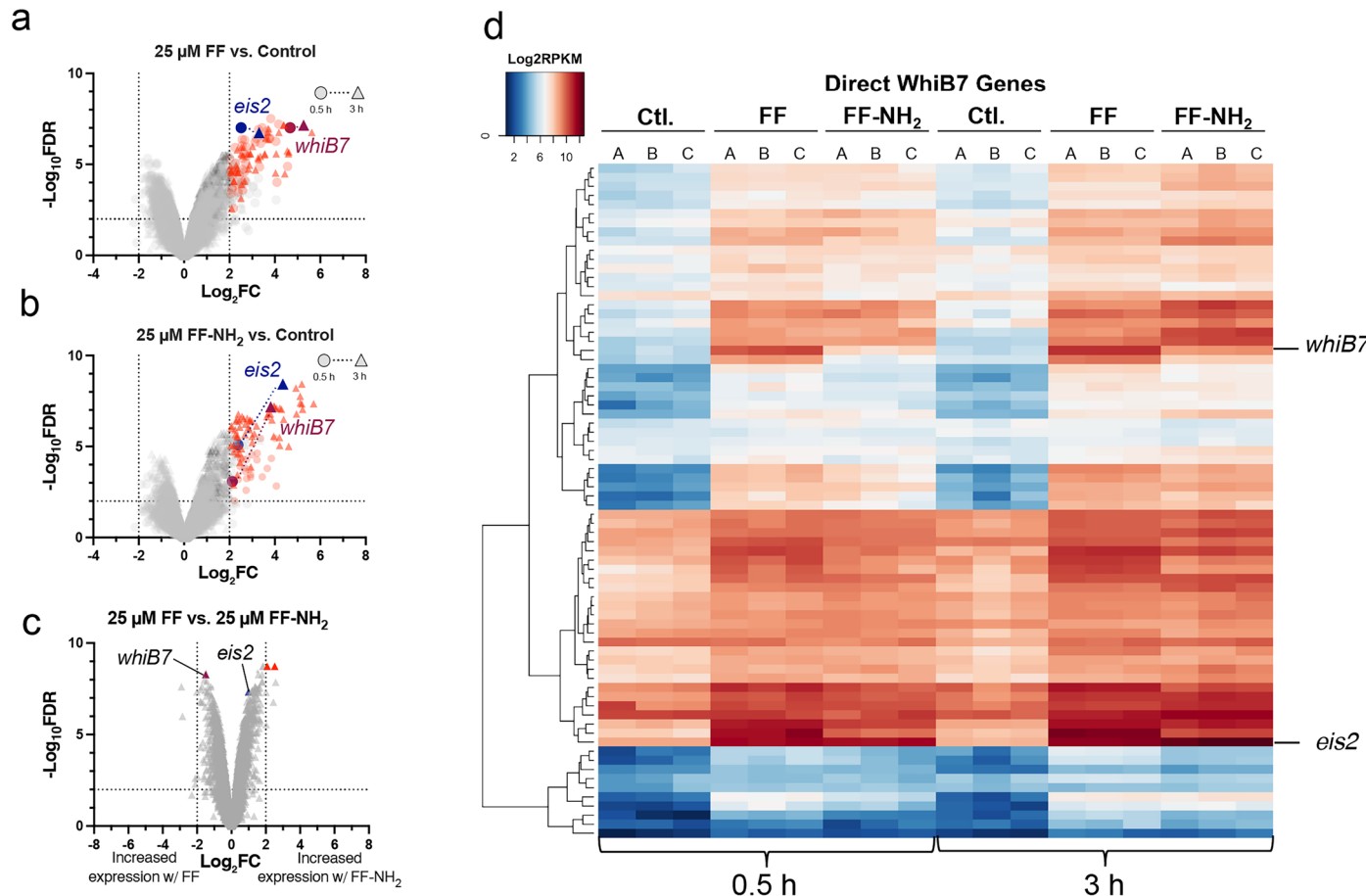

**Extended Data Fig. 5 | Florfenicol and florfenicol amine induce the expression of *whiB7* and *eis2*. a**, **b**, Overlapping volcano plot of transcriptomic profiles for *M. abscessus* cells treated for 0.5-h (circles) and 3-h (triangles) with 25 µM of (**a**) florfenicol (FF) and (**b**) florfenicol amine (FF-NH₂). **c**, Volcano plot comparing transcriptomic profile of FF and FF-NH₂ treated *M. abscessus* cells at the 3-h time point. **d**, Heatmap displaying Log2RPKM (reads per kilobase per million mapped reads) of 74 direct WhiB7 target and operonic genes identified in ref. 25 upon treatment with 25 µM FF, FF-NH2, or vehicle control (DMSO) at 0.5 and 3 h. Transcriptomic data are representative of three biological replicates. **a**–**c**, Differentially expressed (L2FC > |2|, -Log10 FDR < 0.01) WhiB7 target genes are identified as red circles/triangles, where *whiB7* (purple circle/triangle) and *eis2* (blue circle/triangle) are specifically highlighted.

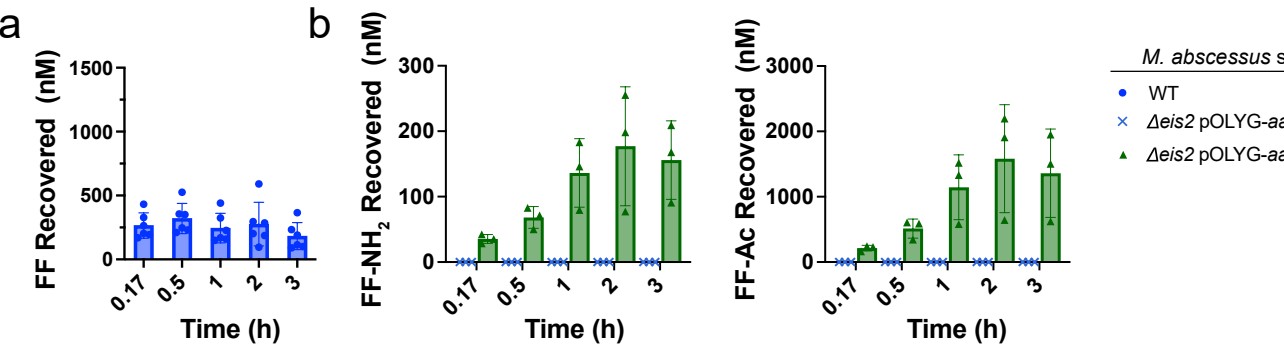

**Extended Data Fig. 6 | Florfenicol and florfenicol acetyl display differential accumulation patterns. a**, Accumulation of florfenicol (FF) in the *M. abscessus* WT strain up to 3 h after treatment with 100 µM FF (*n* = 6). **b**, Accumulation of FF amine (FF-NH₂, left) and FF acetyl (FF-ac, right) in *M. abscessus* Δ*eis2* strain

complemented with pOLYG-*aac(3)IV-eis2* or pOLYG-*aac(3)IV* control vector for to 3 h after 100 µM florfenicol amine (FF-NH₂) treatment (*n* = 3 per group). Each point represents an independent biological experiment. Mean ± s.d. shown. x, below the limit of quantification (LOQ).

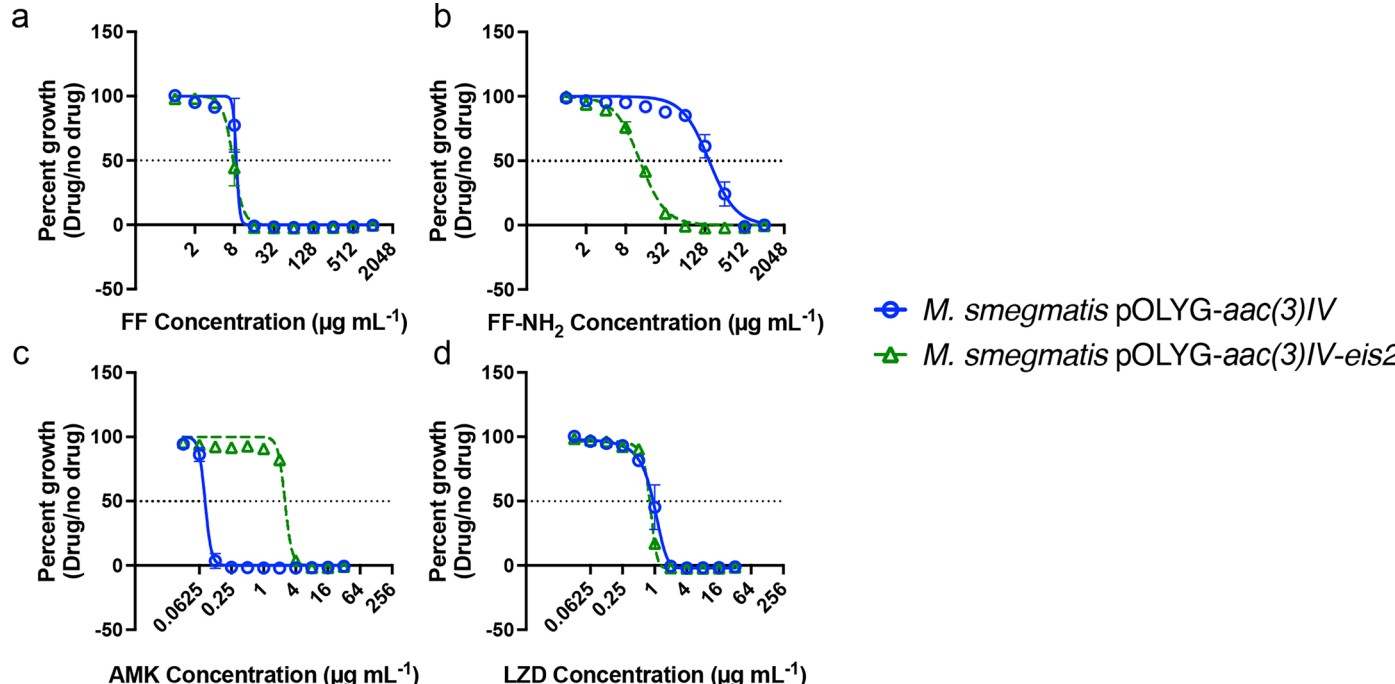

M. smegmatis pOLYG-*aac(3)IV*
M. smegmatis pOLYG-*aac(3)IV-eis2*

**Extended Data Fig. 7 | Heterologous expression of *eis2* in *M. smegmatis* confers florfenicol amine activity. a**, Dose-response curves for **a**, florfenicol (FF) **b**, florfenicol amine (FF-NH$_2$), **c**, amikacin (AMK) and **d**, linezolid (LZD) tested against *M. smegmatis* mc$^2$ 155 strain with pOLYG-*aac(3)IV-eis2* overexpression or pOLYG-*aac(3)IV* control vectors. All dose-response curves are displayed as mean ± s.e.m., *n* = 3 biological replicates.

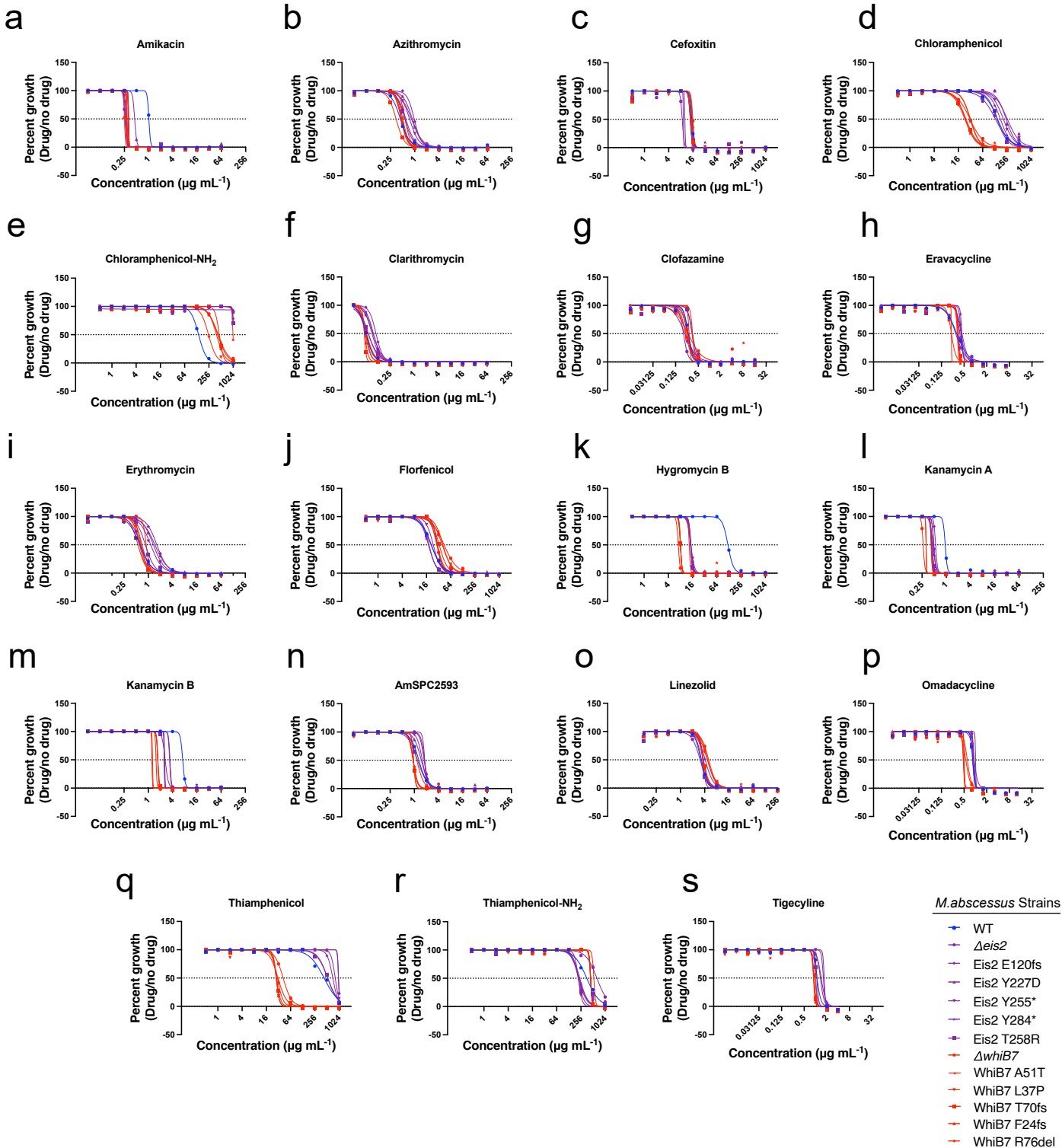

**Extended Data Fig. 8 | Activity of antimicrobials against florfenicol amine resistant *M. abscessus* strains.** Dose-response curves of *M. abscessus* WT, *whiB7* mutants, and *eis2* mutants against **a**, amikacin, **b**, azithromycin, **c**, cefoxitin, **d**, chloramphenicol, **e**, chloramphenicol amine (CAM-NH$_2$), **f**, clarithromycin, **g**, clofazimine, **h**, eravacycline, **i**, erythromycin, **j**, florfenicol, **k**, hygromycin B, **l**, kanamycin A, **m**, kanamycin B, **n**, AmSPC2593, **o**, linezolid, **p**, omadacycline, **q**, thiamphenicol, **r**, thiamphenicol amine (TAM-NH$_2$), and **s**, tigecycline. All dose-response curves are displayed as mean, *n* = 2 biological replicates.

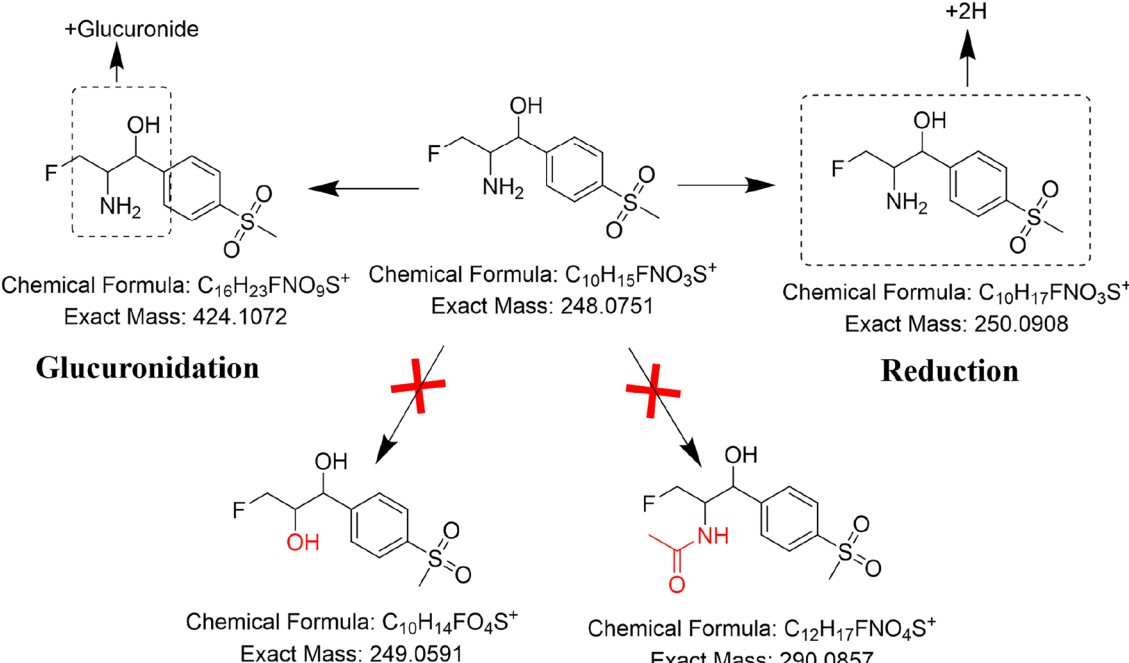

**Extended Data Fig. 9 | Proposed structure of metabolites of florfenicol amine.** The chemical structure, formula, and exact mass of parent molecule florfenicol amine (FF-NH₂), along with its proposed metabolites from hepatocyte stability assay. Glucuronidation and reduction were identified as the predominant metabolites in rodent hepatocytes, while metabolites resulting from *N*-acetylation or oxidative deamination (chemical modifications highlighted in red) were not detected for any of the tested species (mouse, rat, or human). This absence is denoted with a red 'X' to indicate their lack of formation.

# Reporting Summary

## Statistics

For all statistical analyses, confirm that the following items are present in the figure legend, table legend, main text, or Methods section.

| n/a | Confirmed | |
|---|---|---|
| ☐ | ☒ | The exact sample size (*n*) for each experimental group/condition, given as a discrete number and unit of measurement |
| ☐ | ☒ | A statement on whether measurements were taken from distinct samples or whether the same sample was measured repeatedly |
| ☐ | ☒ | The statistical test(s) used AND whether they are one- or two-sided  *Only common tests should be described solely by name; describe more complex techniques in the Methods section.* |
| ☒ | ☐ | A description of all covariates tested |
| ☐ | ☒ | A description of any assumptions or corrections, such as tests of normality and adjustment for multiple comparisons |
| ☐ | ☒ | A full description of the statistical parameters including central tendency (e.g. means) or other basic estimates (e.g. regression coefficient) AND variation (e.g. standard deviation) or associated estimates of uncertainty (e.g. confidence intervals) |
| ☐ | ☒ | For null hypothesis testing, the test statistic (e.g. *F*, *t*, *r*) with confidence intervals, effect sizes, degrees of freedom and *P* value noted  *Give P values as exact values whenever suitable.* |
| ☒ | ☐ | For Bayesian analysis, information on the choice of priors and Markov chain Monte Carlo settings |
| ☒ | ☐ | For hierarchical and complex designs, identification of the appropriate level for tests and full reporting of outcomes |
| ☒ | ☐ | Estimates of effect sizes (e.g. Cohen's *d*, Pearson's *r*), indicating how they were calculated |

*Our web collection on statistics for biologists contains articles on many of the points above.*

## Software and code

Policy information about availability of computer code

| Data collection | Dose-response activity data were obtained using a PHERAstar FS Multilabel Reader (BMG) for antibacterial activity measurements and enzymatic assays, and a Synergy H1 Plate Reader (BioTek) for the translation inhibition assay. Drug accumulation data were collected with the Echo MS system, managed and analyzed using Sciex OS-MQ Analytics Software (AB Sciex). Next-generation sequencing was performed on a NovaSeq X Plus system (Illumina). |
|---|---|
| Data analysis | Detailed methodologies are provided in the Methods section and are available upon request from the corresponding author. The following programs and versions were utilized in this study:  GraphPad Prism (version 10.2.0)  Microsoft Excel (365)  CLC Genomics Workbench (version 23.0.2)  R (version 4.1.0)  Sciex OS-MQ Analytics Software (version 3.3.10; version 2.1.6)  bcl-convert (version 4.2.4)  subread (version 2.0.1)  STAR (version 2.7.1a)  FastQC (version 0.11.5)  MultiQC (version 1.15)  tidyverse (version 2.0.0)  ggplot2 (version 3.4.4)  gplots (version 3.1.3 and 3.2.0)  edgeR (version 4.4.2) |

limma (version 3.62.2)
magrittr (version 2.0.3)
NMF (version 0.23)
RColorBrewer (version 1.1-3)
JUMP shiny (jumpshiny.genenetwork.org)
braidrm (version 1.0.3)

For manuscripts utilizing custom algorithms or software that are central to the research but not yet described in published literature, software must be made available to editors and reviewers. We strongly encourage code deposition in a community repository (e.g. GitHub). See the Nature Portfolio guidelines for submitting code & software for further information.

# Data

Policy information about availability of data

All manuscripts must include a data availability statement. This statement should provide the following information, where applicable:
- Accession codes, unique identifiers, or web links for publicly available datasets
- A description of any restrictions on data availability
- For clinical datasets or third party data, please ensure that the statement adheres to our policy

Whole-genome sequencing reads associated with this study have been submitted to the NCBI Sequence Read Archive (SRA) under accession number PRJNA1141985. Raw RNA-seq data reads have been deposited in the NCBI Gene Expression Omnibus (GEO) under accession number GSE273574. Raw mass spectra from proteomics experiment have been deposited to proteomeXchange and MassIVE repositories with identifiers PXD059834 and MSV000096854. All data supporting the findings of this study are provided within the Article, Supplementary Information (SI), Extended Datasets, and Source Data files.

# Research involving human participants, their data, or biological material

Policy information about studies with human participants or human data. See also policy information about sex, gender (identity/presentation), and sexual orientation and race, ethnicity and racism.

| Reporting on sex and gender | NA |
| --- | --- |
| Reporting on race, ethnicity, or other socially relevant groupings | NA |
| Population characteristics | NA |
| Recruitment | NA |
| Ethics oversight | NA |

Note that full information on the approval of the study protocol must also be provided in the manuscript.

# Field-specific reporting

Please select the one below that is the best fit for your research. If you are not sure, read the appropriate sections before making your selection.

☒ Life sciences    ☐ Behavioural & social sciences    ☐ Ecological, evolutionary & environmental sciences

For a reference copy of the document with all sections, see nature.com/documents/nr-reporting-summary-flat.pdf

# Life sciences study design

All studies must disclose on these points even when the disclosure is negative.

| Sample size | The drug dose-response assays presented in this manuscript are derived from a minimum of three independent experiments, except for the high-throughput susceptibility testing of the Mab mutants, which was conducted in duplicate per strain (13 strains tested total). Checkerboard assays were performed in triplicate, and MIC assays were conducted in at least biological duplicate. Accumulation experiments were performed in biological triplicate for those using Mab Δeis2 and Mab ΔwhiB7 strains, except for 0.167 h time point for Mab ΔwhiB7 (two replicates), and Mab WT strain was tested with a n=6. Transcriptomic and proteomic studies were conducted in biological triplicate. Hepatocyte stability studies were performed in triplicate. For the animal studies, pharmacokinetic studies were performed in triplicate and efficacy studies contained five animals per treatment group. |
| --- | --- |
| Data exclusions | No data were excluded |
| Replication | The frequency of each experiment is detailed in the figure legends and/or the corresponding Methods sections. |
| Randomization | Randomization was not considered for the in vitro experiments conducted in this study. GM-CSF mice (6-9 weeks old) were randomly assigned to treatment groups upon arrival. |

| Blinding | Blinding was not considered necessary for this study. |
|---|---|

# Reporting for specific materials, systems and methods

We require information from authors about some types of materials, experimental systems and methods used in many studies. Here, indicate whether each material, system or method listed is relevant to your study. If you are not sure if a list item applies to your research, read the appropriate section before selecting a response.

## Materials & experimental systems

| n/a | Involved in the study |
|---|---|
| ☐ | ☒ Antibodies |
| ☐ | ☒ Eukaryotic cell lines |
| ☒ | ☐ Palaeontology and archaeology |
| ☐ | ☒ Animals and other organisms |
| ☒ | ☐ Clinical data |
| ☒ | ☐ Dual use research of concern |
| ☒ | ☐ Plants |

## Methods

| n/a | Involved in the study |
|---|---|
| ☒ | ☐ ChIP-seq |
| ☒ | ☐ Flow cytometry |
| ☒ | ☐ MRI-based neuroimaging |

## Antibodies

| Antibodies used | The Ab110217 MitoBiogenesis™ In-Cell ELISA Kit (Abcam) was utilized to assess the drug-induced effects on mitochondrial biogenesis of the experimental and control compounds. Primary antibodies were supplied at a 200X concentration and secondary antibodies at a 2500X concentration. The experiment was conducted according to the manufacturer's instructions, with minor modifications pertaining to cell density as detailed in the Methods section. |
|---|---|
| Validation | Validation was conducted in-house by testing controls, including chloramphenicol (CAM). Our results demonstrated a similar propensity for CAM to inhibit mitochondrial protein synthesis (MPS), with an MPS IC50 of 5.5 +/- 2.5μM (Table S7), closely aligning with the manufacturer's reported value of 8.1 μM. |

## Eukaryotic cell lines

Policy information about cell lines and Sex and Gender in Research

| Cell line source(s) | HepG2 cells were obtained from ATCC (ATCC HB-8065) |
|---|---|
| Authentication | Cell authentication was performed using STR-PCR. |
| Mycoplasma contamination | The cells tested negative for mycoplasma contamination. |
| Commonly misidentified lines (See ICLAC register) | No misidentified cell lines were used in this study. |

## Animals and other research organisms

Policy information about studies involving animals; ARRIVE guidelines recommended for reporting animal research, and Sex and Gender in Research

| Laboratory animals | PK (UTHSC): Healthy male BALB/c mice (8-12 weeks old) weighing around 20 g were procured from Charles River (Wilmington, MA). All pharmacokinetic animal studies were conducted in accordance with the Animal Welfare Act and the Public Health Service Policy on Humane Care and Use of Laboratory Animals. Prior to initiation, the respective animal protocol was approved by the Institutional Animal Care and Use Committees of the University of Tennessee Health Science Center.

Efficacy (CSU): Female GM-CSF knockout mice (Strain #: 026812, B6.129S-Csf2tm1Mlg/J), aged 6–9 weeks, were purchased from the Jackson Laboratory. The mice were infected with M. abscessus via the intrapulmonary aerosol route in the Biosafety Level-3 (BSL-3) laboratory. |
|---|---|
| Wild animals | No wild animals were used in this study. |
| Reporting on sex | All animals were female. |
| Field-collected samples | No field-collected samples were used. |
| Ethics oversight | PK (UTHSC): All pharmacokinetic animal studies were conducted in accordance with the Animal Welfare Act and the Public Health Service Policy on Humane Care and Use of Laboratory Animals. Prior to initiation, the respective animal protocol (ACUC #24-0590) was approved by the Institutional Animal Care and Use Committees of the University of Tennessee Health Science Center. |

Efficacy (CSU): All mouse experiments were approved by Colorado State University's Institutional Animal Care and Use Committee (IACUC), protocol #5157.

Note that full information on the approval of the study protocol must also be provided in the manuscript.

## Plants

Seed stocks

NA

Novel plant genotypes

NA

Authentication

NA

