## [Peer Review File · Nature Microbiology]

Prodrug florfenicol amine is activated by intrinsic resistance to target *Mycobacterium abscessus*

Corresponding Author: Dr Richard Lee

Version 0:

Reviewer comments:

Reviewer #1

(Remarks to the Author)

The authors report an intriguing discovery of collateral sensitivity by exploiting an N-acetyltransferase for bioactivation of an antibiotic precursor, converting it to an active drug, whereas that same N-acetyl transferase inactivates other classes of antibiotics. The opposing use of the same enzyme for bioactivation of one antibiotic and resistance to a different antibiotic provides a unique opportunity to exploit collateral sensitivity through antibiotic cycling. This approach also simultaneously eliminates the toxicity of the parent antibiotic, caused by inhibition of the corresponding host target. While antibiotics are generally regarded as highly selective and safe, several antibiotics targeting protein synthesis do inhibit the human mitochondrial ribosomes leading to host toxicity. Indeed, the antibiotic investigated, florfenicol (FF), is a member of phenicol class of antibiotics used worldwide for almost two decades before toxicity was broadly recognized that greatly curtailed the clinical use of this highly effective antibiotic.

Antibiotics targeting the peptidyl-transferase center of the ribosome, like the phenicols, exhibit unavoidable mechanism-based toxicity since the binding site is highly conserved between humans and bacteria. Thus, the findings reported herein exploiting a bacterial N-acetyltransferase for selective bioactivation of des-acetyl FF (or FF-NH₂ in the manuscript) to FF mitigates the major liability of the phenicol class of antibiotics. Moreover, the described mechanism is selective for members of the *Mycobacterium abscessus*-*chelonae* complex, responsible for a growing number of incurable infection worldwide. The translation potential is significant since the resistance to des-acetyl FF would lead to collateral sensitivity to other protein synthesis inhibitors routinely used for therapy of these nontuberculous mycobacterial infections. The approach may be generalizable to other classes of antibiotics and indeed a recent paper by Boshoff and co-workers disclosed a related finding with the oxazolidinones (ACS Infect. Dis. 2024, 10, 5, 1679–1695, doi: 10.1021/acsinfectdis.4c00025). The study by Boshoff has many parallels and is certainly impressive in scope, but was focused on the medicinal chemistry and translational discovery, whereas the present manuscript is focused on mechanism, selectivity, antibiotic potentiation, and collateral sensitivity using different methodology. These studies are therefore highly complementary. Overall, the manuscript demonstrates a fascinating strategy that hijacks a pathogen's resistance mechanism against itself further exploiting positive regulation for amplification of the response. Antimicrobial resistance is inevitable, but the proposed findings offer a potential solution using antibiotic cycling to induce collateral sensitivity forcing bacteria to oscillate between two states, both of which are antibiotic susceptible. Moreover, the antibiotic described is effective in a murine *Mycobacterium abscessus* infection model showing translational potential. The work is deemed of high significance and impact and appropriate for Nature Microbiology once the following concerns have been addressed.

1. Can the authors cite some relevant clinical studies of Mab in addition to the cited 2020 review. What are the mortality rates of *M. abscessus* infections if the cure rates are so low?
2. Line 58. Change "*Mycobacterium*" to "mycobacteria"
3. The authors may want to rephrase "However, closely related compounds within the phenicol class, such as florfenicol (FF)...." as "however" is used twice within three sentences. Chloramphenicol is on the WHO' list of essential medicines and anecdotally appears to have high use in many developing countries, so the negative connotation of "however" may be unnecessary.
4. Line 99. Why is IC₅₀ used instead of a more conventional minimum inhibitory concentration (MIC)? If it simply selected because the fold-difference at 50% inhibition between WT and whiB7 mutant strains for FF is maximal, then this the value is not justified. The growth inhibition curves (Figure 1, panel B, middle curve for FF-NH₂) show the MIC against the Mab WT is remarkably flat such that the actual MIC value is closer to 128 ug/mL. The use of GR₅₀ value or the concentration that inhibits 50% of the normalized growth rate during the early exponential phase is a superior metric for measuring drug responsiveness. (see: Hafner, M. et al. Nature Methods, 13 (6), 2016; Sasseti, et al. PNAS 2022, 19(15):e2201632119:521-7. doi: 10.1038/nmeth.3853)
5. The drug concentrations should also be reported in units of molarity (μM) at least somewhere.
6. It would be nice to validate an *eis2* mutant genetically and/or biochemically. The validation provided was in isogenic deletion strain and complementation should ideally use a mutant background.
7. Line 153. Please do not report IC₅₀ values to 4 significant figures as the precision in this measurement is likely only to two significant figures. Use scientific notation. This applies to all fitted values throughout. In order to report to this level of precision, the drug must be accurately to 5 significant figures and all subsequent operations must be performed with such precision. Propagation of error in a two-fold series will lead to higher error at lower concentrations.

8. a) Line 161. Adjust the significant figures of K_m . The symbols K_m and V_{max} should have the 'K' and 'V' italicized by standard convention. However, suggest reporting the k_{cat} value rather than the V_{max} by dividing by the enzyme concentration giving units of per second. The specificity constant (k_{cat}/K_m) is the best parameter for characterizing substrate specificity (expressed in units of $M^{-1}s^{-1}$) and this should be reported too. It would be useful to have one known substrate with reported kinetic parameters included in the Supporting Information Table (if one of the drugs has reported data already, then this can be included with a reference)

9. FF-NH₂ has a k_{cat}/K_m of 1.7×10^3 indicating it is a poor substrate and hence will be converted slowly. In the accumulation studies, the concentration of FF-NH₂ is far below K_m , thus the Michaelis-Menten equation simplifies to $k_{cat}/K_m \times [E][S]$ (where k_{cat}/K_m is effectively a second-order rate constant). This can be used to calculate the rate of bioactivation. Initially, Eis2 is uninduced and the constitutive expression is probably low. Using an estimate of 1 μM for Eis2, one can calculate the initial rate of bioactivation of FF-NH₂ (initial concentration ~ 10 nM) to FF-Ac, which equates to $1.7 \times 10^3 M^{-1}s^{-1} \times [10 \times 10^{-9} M] \times [10^{-6} M] = 1.7 \times 10^{-11} M s^{-1} = 1.7 \times 10^{-5} \mu M s^{-1}$. At 10 minutes (600 seconds), the total conversion would be: $1.7 \times 10^{-5} \mu M s^{-1} \times 600 s = 10^{-2} \mu M = 10$ nM, which is within an order of magnitude of the measured concentration, but the bacteria remained viable until lysed, which by the protocol would have easily taken another 20 minutes giving an estimate of ≥ 30 nM within 2-fold of the observed value. At 3 hours, this predicts about 180 nM of FF-Ac, but as Eis2 is induced the rate of bioactivation will increase. Indeed, the observed value of 1800 nM suggests significant induction. Thus, the kinetic parameters can be quite useful.

10. The rate of bioactivation should also be discussed briefly as this becomes an important parameter in the context of animal studies where the drug concentration is not constant, but decreasing with time. Thus, if the rate of drug elimination in vivo ($t_{1/2} = 0.8$ h, Table S6) is faster than the rate of bioactivation ($t_{1/2} = 6.8$ h, estimated initial rate before induction of Eis2, $k_{obs} = k_{cat}/K_m \times [E] = (1.7 \times 10^3 M^{-1}s^{-1}) \times [10^{-6} M] = 1.7 \times 10^{-3} M$); $t_{1/2} = \ln 2/k_{obs} = 6.8$ h), then efficacy in vivo cannot be obtained. This likely explains the need to dose subcutaneously, when the drug has 50% oral bioavailability.

11. TAM-NH₂ and CAM-NH₂ are bioactivated with similar or better efficiencies (TAM-NH₂ is 3x better) yet TAM-NH₂ shows no enhancement in activity against the whiB7 mutant, despite being a significantly better substrate (three-fold better than FF-NH₂) while CAM-NH₂ shows a modest 2-fold increase in potency against the whiB7 mutant. The authors should provide a rationale for the observed differences.

12. Was the concentration of FF-Ac in the supernatant measured. The discussion invokes Fick's law to explain accumulation, but this would presume the FF-Ac is trapped within the bacteria. Presumably, FF-Ac can equilibrate with the extracellular environment. The compound could be sequestered by the abundant ribosomes explaining their accumulation.

13. Does O-acetyltransferase Mab2989 (Cat) confer resistance to the amphenicols? Once bioactivated they may be susceptible to inactivation by Cat. What is the MIC when Cat is deleted? This would be valuable information and may explain the overall modest activity of the amphenicols.

14. The time-course of the RNA-Seq experiments is very nice. The experimental states 25 μM of FF-NH₂ and FF were used that corresponds to about 1/8 of the MIC values (MIC is ~ 64 $\mu g/mL = 200$ μM). It would also be useful to give the concentration relative to the MIC values. Expressing the concentration in units of molarity throughout would be preferred. Was the concentration optimized?

15. The reported analytical method for compound accumulation does not report an internal standard that is required for quantitation. No information is provided on how the intracellular concentration was determined. No information is provided for the chromatography (column, gradient).

16. The reported clearance and volume of distribution for the pharmacokinetic experiments in Table S6 are incorrect. These are based on data from one mouse with a reported half-life of 2.4 hours following intravenous administration, whereas the oral half-life is 0.8 hours. These values are incongruent.

17. Line 300. The data is reported to 6 significant figures. Based on the method, the data should not be reported beyond 2 significant figures.

18. The reported Pharmacokinetic Studies in the Supplementary methods lack information on the bioanalysis (internal standard, LC-MS methods).

19. Since mice were dosed subcutaneously, the PK for this route of administration would be important, to demonstrate that sufficient exposure was obtained.

20. Source data is provided for the RNA-Seq experiments, but is lacking for everything else. All data reported in the manuscript and the Supplementary methods should have all data for every data point provided to reproduce the values given in the Figures and Tables.

21. Information on metabolism of FF-NH₂ should be provided given Boshoff observed the linezolid amine metabolite exhibited pronounced mitochondrial cytotoxicity.

Reviewer #2

(Remarks to the Author)

M. abscessus infections are extremely difficult to treat, requiring lengthy treatments often associated with therapeutic failure and toxicity. New therapeutic approaches are, therefore, needed to bypass these toxicity issues. In this study, Phelps et al. introduced an amine derivative in florfenicol (FF-NH₂), a drug used in veterinary medicine. Through the use of MedChem, genetic and biochemical approaches, the authors demonstrated that FF-NH₂ avoids neutralization by O-acetylation (catalyzed by Cat) and, at the same time, exploits N-acetylation (catalyzed by Eis2) in *M. abscessus*. Both Cat and Eis2 are part of the WhiB7 regulon, associated to resistance to different antibiotic classes, such as aminoglycosides and macrolides. Herein, the authors show that FF-NH₂ acts as a prodrug that requires activation by Eis2, leading to the accumulation of the active acetyl metabolite, FF-Ac, which in turn inhibits ribosome translation. Importantly, this prodrug displays no toxicity linked to mammalian mitochondrial ribosome inhibition, as compared to the parental FF molecule. In addition, it is orally bioavailable and synergizes with other anti-*M. abscessus* drugs.

In general, the manuscript is very well-written and the results are supported by the data with appropriate controls. The mode of action/activation of FF-NH₂ described here is particularly original as it demonstrates the possibility to exploit the presence of an intrinsic drug resistance element known to be detrimental to other antibiotics used to treat the same pathogen. However, while the mechanisms described here sound very interesting and novel, the global impact of the work is somehow mitigated by the

fact that FF-NH2 is only active to a very restricted number of strains (essentially the *M. abscessus* complex). This is due to the absence or divergence of the amino acid sequence of Eis2 in other pathogens, which seriously limits the applicability of the concept to other (mycobacterial) pathogens. In addition, FF-NH2 does not outperform the standard drug clarithromycin in the mouse model.

There are major points that need to be clarified to further support and validate the conclusions raised.

Major points:

1. FF-NH2 resistant mutants were selected on plates (Figure S2), leading to large and rough colonies for *whiB7* colonies and small and smooth for *eis2* colonies. Both *whiB7* and *eis2* null mutants have been reported previously in the literature with no mention on changes in the morphology and/or growth rate. These differences in the morphology need to be clarified. The presence of the rough colonies can be simply explained by the presence of the antibiotic in the agar. What happens if these colonies are streaked on antibiotic-free plates, do they remain rough or become smooth? In addition, roughness is often linked to the absence of glycopeptidolipids (GPL). What are the GPL profiles of these *whiB7* and *eis2* mutants? Does complementation with *whiB7* restore a smooth background in the *whiB7* mutants?
2. Given the very small size of the *eis2* mutants (again not reported in previous studies) and growth impairment, how did the authors grow these strains to properly determine their drug susceptibility profiles (Table S2)?
3. The efficacy of FF-NH2 has been tested against a very small collection of clinical strains of the *M. abscessus* complex, including only two isolated of *M. abscessus* subsp. *abscessus* (Table S3). What does "CL I" mean. To support the usefulness of this new compounds for the treatment of *M. abscessus* diseases, this table should be expanded by testing more clinical strains, including multidrug resistant strains of the *M. abscessus* complex.
4. An important finding is that FF-NH2 increases the expression of the *WhiB7* regulon, that includes *eis2* (Figure 3). However, this is only based on RNAseq and transcriptional data and needs to be proven at a protein level. I suggest the authors to confirm that *Eis2* expression is increased upon treatment of cultures with FF-NH2 by Western blotting.
5. I understand that FF-NH2 synergizes with different antibiotics, including macrolides, which are often used in clinical settings. However, the data in Fig. 4c also demonstrates that FF-NH2 antagonizes amikacin and cefoxitin, which are both used in the standard treatments. On the one hand, FF-NH2 increases the potency of macrolides but, on the other hand, it decreases the potency of aminoglycosides and β -lactams. Therefore, I suggest the authors to dampen this in the Results and Discussion sections. So, what would be the net result of FF-NH2 when combined with all these different drugs?
6. Related to the previous point, and in order to highlight the usefulness and applicability of FF-NH2 in treatment, drug combinations need to be evaluated *in vivo*. Fig. 4f, shows that FF-NH2 is not better than CLR or LNZ (given at lower concentrations than FF-NH2). Therefore, what would be the advantage of FF-NH2 from a therapeutic perspective, unless the authors can demonstrate that, in mice, the a FF-NH2/CLR combination is better than the drugs administered alone? This data would make the study much more appealing.

Minor points:

-use the correct designation of gene throughout the whole manuscript and figures (for instance MAB_2989 rather than Mab2989).

-line 191: "indicate"

-line 717: "liver and spleen was determined in three mice". However, the CFU shown in Figure 4f suggest 5 mice/group. This needs to be better clarified (as well as the mode of drug administration) in the figure legend.

-Figure S4: I cannot distinguish the colors in the different graphs.

Reviewer #3

(Remarks to the Author)

Phelps and colleagues discovered that florfenicol amine (FF-NH2) acts as a narrow spectrum antibiotic against several non-tuberculous mycobacteria (NTM) including *M. abscessus*. Interestingly, FF-NH2 is a prodrug that requires activation by a modifying enzyme (*Eis2*) regulated by the transcription factor *WhiB7*. As is expected for prodrugs that require activation by a non-essential enzyme, the frequency of resistance is relatively high with loss of activation being the most frequent mechanism of resistance. Intriguingly, in the case of FF-NH2 resistance due to mutations inactivating *Eis2/WhiB7* leads to a collateral sensitivity to several other antibiotics, which could help develop resistance avoiding multi-drug regimens. Overall, this is a very convincing study that describes an attractive starting point to develop a new drug for the treatment of NTM infections (which are among the most difficult to cure infections).

My comments are all minor:

1. The abstract describes FF-NH2 as orally bioavailable, which is technically correct. But oral bioavailability is clearly limited. Otherwise subcutaneous administration would not have been necessary to demonstrate *in vivo* efficacy. Personally, I would tone down the claims regarding oral bioavailability.
2. Perhaps the authors could comment on why they used 6 day experiments for the synergy studies when 14 days seem to have been more informative (lines 263-265).
3. Consider rewording the sentence starting in line 322. Modification of the ribosomal target site likely occurs but with a frequency much lower than those affecting *Eis2/WhiB7*.
4. The authors describe leveraging drug resistance as a generalizable strategy for antibacterial drug development. I agree that this would be attractive, but I wonder how generalizable this strategy actually is. Perhaps the authors could add a few sentences to discuss at least one other resistance mechanism to which they think this strategy could be applied.

Decision Letter:

11th November 2024

Dear Professor Lee,

Thank you for your patience while your manuscript "Hijacking intrinsic resistance in *Mycobacterium abscessus*" was under peer-review at Nature Microbiology. It has now been seen by 3 referees, whose expertise and comments you will find at the end of this email. Although they find your work of some potential interest, they have raised a number of concerns that will need to be addressed before we can consider publication of the work in Nature Microbiology.

In particular, referee #1 asks for further validation of the *eis2* mutant, and for further investigation of whether O-acetyltransferase Mab2989 (Cat) confers resistance to the aminoglycosides. Furthermore, referee #1 states that an internal standard is required for the analytical method for compound accumulation, and that further PK characterization of the subcutaneous route of mouse infection will be required. Importantly, referee #1 is also concerned that the reported IC50 may overestimate the observed effects, and that the potency of the compounds is very weak. Referee #2 is concerned that the global impact of the work may be "mitigated by the fact that FF-NH2 is only active to a very restricted number of strains (essentially the *M. abscessus* complex)." This referee is also concerned that FF-NH2 does not outperform the standard drug clarithromycin in the mouse model. Further, referee #2 feels that more more clinical strains should be tested, including multidrug resistant strains of the *M. abscessus* complex. Also, that FF-NH2 increases the expression of the *WhiB7* regulon will need to be proven at a protein level. Furthermore, the referee say that in order to highlight the usefulness and applicability of FF-NH2 in treatment, drug combinations need to be evaluated *in vivo*. Referee #3 has several minor points to address. Editorially, we will require the referee concerns to be addressed in full.

Should further experimental data allow you to address these criticisms, we would be happy to look at a revised manuscript.

Please include a data availability statement as a separate section after Methods but before references, under the heading "Data Availability". This section should inform readers about the availability of the data used to support the conclusions of your study. This information includes accession codes to public repositories (data banks for protein, DNA or RNA sequences, microarray, proteomics data etc...), references to source data published alongside the paper, unique identifiers such as URLs to data repository entries, or data set DOIs, and any other statement about data availability. At a minimum, you should include the following statement: "The data that support the findings of this study are available from the corresponding author upon request", mentioning any restrictions on availability. If DOIs are provided, we also strongly encourage including these in the Reference list (authors, title, publisher (repository name), identifier, year). For more guidance on how to write this section please see: <http://www.nature.com/authors/policies/data/data-availability-statements-data-citations.pdf>

* If you have not done so already we suggest that you begin to revise your manuscript so that it conforms to our Article format instructions at <http://www.nature.com/nmicrobiol/info/final-submission>. Refer also to any guidelines provided in this letter.

When submitting the revised version of your manuscript, please pay close attention to our [href="https://www.nature.com/nature-portfolio/editorial-policies/image-integrity">Digital Image Integrity Guidelines](https://www.nature.com/nature-portfolio/editorial-policies/image-integrity) and to the following points below:

Link Redacted

Note: This url links to your confidential homepage and associated information about manuscripts you may have submitted or be reviewing for us. If you wish to forward this e-mail to co-authors, please delete this link to your homepage first.

Nature Microbiology is committed to improving transparency in authorship. As part of our efforts in this direction, we are now requesting that all authors identified as 'corresponding author' on published papers create and link their Open Researcher and Contributor Identifier (ORCID) with their account on the Manuscript Tracking System (MTS), prior to acceptance. This applies to primary research papers only. ORCID helps the scientific community achieve unambiguous attribution of all scholarly contributions. You can create and link your ORCID from the home page of the MTS by clicking on 'Modify my Springer Nature account'. For more information please visit www.springernature.com/orcid.

If you wish to submit a suitably revised manuscript we would hope to receive it within 6 months. If you cannot send it within this time, please let us know.

Yours sincerely,

Reviewer Expertise:

Referee #1: Mycobacteria, drug discovery

Referee #2: Mycobacterial pathogenesis, therapeutic targets

Referee #3: Mycobacterial pathogenesis, therapeutic targets

Reviewer Comments:

Reviewer #1 (Remarks to the Author):

The authors report an intriguing discovery of collateral sensitivity by exploiting an N-acetyltransferase for bioactivation of an antibiotic precursor, converting it to an active drug, whereas that same N-acetyl transferase inactivates other classes of antibiotics. The opposing use of the same enzyme for bioactivation of one antibiotic and resistance to a different antibiotic provides a unique opportunity to exploit collateral sensitivity through antibiotic cycling. This approach also simultaneously eliminates the toxicity of the parent antibiotic, caused by inhibition of the corresponding host target. While antibiotics are generally regarded as highly selective and safe, several antibiotics targeting protein synthesis do inhibit the human mitochondrial ribosomes leading to host toxicity. Indeed, the antibiotic investigated, florfenicol (FF), is a member of phenicol class of antibiotics used worldwide for almost two decades before toxicity was broadly recognized that greatly curtailed the clinical use of this highly effective antibiotic. Antibiotics targeting the peptidyl-transferase center of the ribosome, like the phenicols, exhibit unavoidable mechanism-based toxicity since the binding site is highly conserved between humans and bacteria. Thus, the findings reported herein exploiting a bacterial N-acetyltransferase for selective bioactivation of des-acetyl FF (or FF-NH2 in the manuscript) to FF mitigates the major liability of the phenicol class of antibiotics. Moreover, the described mechanism is selective for members of the Mycobacterium abscessus-chelonae complex, responsible for a growing number of incurable infection worldwide. The translation potential is significant since the resistance to des-acetyl FF would lead to collateral sensitivity to other protein synthesis inhibitors routinely used for therapy of these nontuberculous mycobacterial infections. The approach may be generalizable to other classes of antibiotics and indeed a recent paper by Boshoff and co-workers disclosed a related finding with the oxazolidinones (ACS Infect. Dis. 2024, 10, 5, 1679–1695, doi: 10.1021/acscinfeddis.4c00025). The study by Boshoff has many parallels and is certainly impressive in scope, but was focused on the medicinal chemistry and translational discovery, whereas the present manuscript is focused on mechanism, selectivity, antibiotic potentiation, and collateral sensitivity using different methodology. These studies are therefore highly complementary. Overall, the manuscript demonstrates a fascinating strategy that hijacks a pathogen's resistance mechanism against itself further exploiting positive regulation for amplification of the response. Antimicrobial resistance is inevitable, but the proposed findings offer a potential solution using antibiotic cycling to induce collateral sensitivity forcing bacteria to oscillate between two states, both of which are antibiotic susceptible. Moreover, the antibiotic described is effective in a murine Mycobacterium abscessus infection model showing translational potential. The work is deemed of high significance and impact and appropriate for Nature Microbiology once the following concerns have been addressed.

1. Can the authors cite some relevant clinical studies of Mab in addition to the cited 2020 review. What are the mortality rates of M. abscessus infections if the cure rates are so low?
2. Line 58. Change "Mycobacterium" to "mycobacteria"
3. The authors may want to rephrase "However, closely related compounds within the phenicol class, such as florfenicol (FF)..." as "however" is used twice within three sentences. Chloramphenicol is on the WHO' list of essential medicines and anecdotally appears to have high use in many developing countries, so the negative connotation of "however" may be unnecessary.
4. Line 99. Why is IC50 used instead of a more conventional minimum inhibitory concentration (MIC)? If it simply selected because the fold-difference at 50% inhibition between WT and whiB7 mutant strains for FF is maximal, then this the value is not justified. The growth inhibition curves (Figure 1, panel B, middle curve for FF-NH2) show the MIC against the Mab WT is remarkably flat such that the actual MIC value is closer to 128 ug/mL. The use of GR50 value or the concentration that inhibits 50% of the normalized growth rate during the early exponential phase is a superior metric for measuring drug responsiveness. (see: Hafner, M. et al. Nature Methods, 13 (6), 2016; Sasseti, et al. PNAS 2022, 19(15):e2201632119:521-7. doi:

10.1038/nmeth.3853)

5. The drug concentrations should also be reported in units of molarity (μM) at least somewhere.
6. It would be nice to validate an *eis2* mutant genetically and/or biochemically. The validation provided was in isogenic deletion strain and complementation should ideally use a mutant background.
7. Line 153. Please do not report IC_{50} values to 4 significant figures as the precision in this measurement is likely only to two significant figures. Use scientific notation. This applies to all fitted values throughout. In order to report to this level of precision, the drug must be accurately to 5 significant figures and all subsequent operations must be performed with such precision. Propagation of error in a two-fold series will lead to higher error at lower concentrations.
8. a) Line 161. Adjust the significant figures of K_m . The symbols K_m and V_{max} should have the 'K' and 'V' italicized by standard convention. However, suggest reporting the k_{cat} value rather than the V_{max} by dividing by the enzyme concentration giving units of per second. The specificity constant (k_{cat}/K_m) is the best parameter for characterizing substrate specificity (expressed in units of $\text{M}^{-1}\text{s}^{-1}$) and this should be reported too. It would be useful to have one known substrate with reported kinetic parameters included in the Supporting Information Table (if one of the drugs has reported data already, then this can be included with a reference)
9. FF-NH2 has a k_{cat}/K_m of 1.7×10^3 indicating it is a poor substrate and hence will be converted slowly. In the accumulation studies, the concentration of FF-NH2 is far below K_m , thus the Michaelis-Menten equation simplifies to $k_{\text{cat}}/K_m \times [E][S]$ (where k_{cat}/K_m is effectively a second-order rate constant). This can be used to calculate the rate of bioactivation. Initially, *Eis2* is uninduced and the constitutive expression is probably low. Using an estimate of $1 \mu\text{M}$ for *Eis2*, one can calculate the initial rate of bioactivation of FF-NH2 (initial concentration $\sim 10 \text{ nM}$) to FF-Ac, which equates to $1.7 \times 10^3 \text{ M}^{-1}\text{s}^{-1} \times [10 \times 10^{-9} \text{ M}] \times [10^{-6} \text{ M}] = 1.7 \times 10^{-11} \text{ M s}^{-1} = 1.7 \times 10^{-5} \mu\text{M s}^{-1}$. At 10 minutes (600 seconds), the total conversion would be: $1.7 \times 10^{-5} \mu\text{M s}^{-1} \times 600 \text{ s} = 10^{-2} \mu\text{M} = 10 \text{ nM}$, which is within an order of magnitude of the measured concentration, but the bacteria remained viable until lysed, which by the protocol would have easily taken another 20 minutes giving an estimate of $\geq 30 \text{ nM}$ within 2-fold of the observed value. At 3 hours, this predicts about 180 nM of FF-Ac, but as *Eis2* is induced the rate of bioactivation will increase. Indeed, the observed value of 1800 nM suggests significant induction. Thus, the kinetic parameters can be quite useful.
10. The rate of bioactivation should also be discussed briefly as this becomes an important parameter in the context of animal studies where the drug concentration is not constant, but decreasing with time. Thus, if the rate of drug elimination *in vivo* ($t_{1/2} = 0.8 \text{ h}$, Table S6) is faster than the rate of bioactivation ($t_{1/2} = 6.8 \text{ h}$, estimated initial rate before induction of *Eis2*, $k_{\text{obs}} = k_{\text{cat}}/K_m \times [E] = (1.7 \times 10^3 \text{ M}^{-1}\text{s}^{-1}) \times [10^{-6} \text{ M}] = 1.7 \times 10^{-3} \text{ M s}^{-1}$; $t_{1/2} = \ln 2 / k_{\text{obs}} = 6.8 \text{ h}$), then efficacy *in vivo* cannot be obtained. This likely explains the need to dose subcutaneously, when the drug has 50% oral bioavailability.
11. TAM-NH2 and CAM-NH2 are bioactivated with similar or better efficiencies (TAM-NH2 is 3x better) yet TAM-NH2 shows no enhancement in activity against the *whiB7* mutant, despite being a significantly better substrate (three-fold better than FF-NH2) while CAM-NH2 shows a modest 2-fold increase in potency against the *whiB7* mutant. The authors should provide a rationale for the observed differences.
12. Was the concentration of FF-Ac in the supernatant measured. The discussion invokes Fick's law to explain accumulation, but this would presume the FF-Ac is trapped within the bacteria. Presumably, FF-Ac can equilibrate with the extracellular environment. The compound could be sequestered by the abundant ribosomes explaining their accumulation.
13. Does O-acetyltransferase Mab2989 (*Cat*) confer resistance to the amphenicols? Once bioactivated they may be susceptible to inactivation by *Cat*. What is the MIC when *Cat* is deleted? This would be valuable information and may explain the overall modest activity of the amphenicols.
14. The time-course of the RNA-Seq experiments is very nice. The experimental states $25 \mu\text{M}$ of FF-NH2 and FF were used that corresponds to about 1/8 of the MIC values (MIC is $\sim 64 \mu\text{g/mL} = 200 \mu\text{M}$). It would also be useful to give the concentration relative to the MIC values. Expressing the concentration in units of molarity throughout would be preferred. Was the concentration optimized?
15. The reported analytical method for compound accumulation does not report an internal standard that is required for quantitation. No information is provided on how the intracellular concentration was determined. No information is provided for the chromatography (column, gradient).
16. The reported clearance and volume of distribution for the pharmacokinetic experiments in Table S6 are incorrect. These are based on data from one mouse with a reported half-life of 2.4 hours following intravenous administration, whereas the oral half-life is 0.8 hours. These values are incongruent.
17. Line 300. The data is reported to 6 significant figures. Based on the method, the data should not be reported beyond 2 significant figures.
18. The reported Pharmacokinetic Studies in the Supplementary methods lack information on the bioanalysis (internal standard, LC-MS methods).
19. Since mice were dosed subcutaneously, the PK for this route of administration would be important, to demonstrate that sufficient exposure was obtained.
20. Source data is provided for the RNA-Seq experiments, but is lacking for everything else. All data reported in the manuscript and the Supplementary methods should have all data for every data point provided to reproduce the values given in the Figures and Tables.
21. Information on metabolism of FF-NH2 should be provided given Boshoff observed the linezolid amine metabolite exhibited pronounced mitochondrial cytotoxicity.

Reviewer #2 (Remarks to the Author):

M. abscessus infections are extremely difficult to treat, requiring lengthy treatments often associated with therapeutic failure and toxicity. New therapeutic approaches are, therefore, needed to bypass these toxicity issues. In this study, Phelps et al. introduced an amine derivative in florfenicol (FF-NH2), a drug used in veterinary medicine. Through the use of MedChem, genetic and biochemical approaches, the authors demonstrated that FF-NH2 avoids neutralization by O-acetylation (catalyzed by *Cat*) and, at the same time, exploits N-acetylation (catalyzed by *Eis2*) in *M. abscessus*. Both *Cat* and *Eis2* are part of the *WhiB7* regulon, associated to resistance to different antibiotic classes, such as aminoglycosides and macrolides. Herein, the

authors show that FF-NH2 acts as a prodrug that requires activation by Eis2, leading to the accumulation of the active acetyl metabolite, FF-Ac, which in turn inhibits ribosome translation. Importantly, this prodrug displays no toxicity linked to mammalian mitochondrial ribosome inhibition, as compared to the parental FF molecule. In addition, it is orally bioavailable and synergizes with other anti-*M. abscessus* drugs.

In general, the manuscript is very well-written and the results are supported by the data with appropriate controls. The mode of action/activation of FF-NH2 described here is particularly original as it demonstrates the possibility to exploit the presence of an intrinsic drug resistance element known to be detrimental to other antibiotics used to treat the same pathogen. However, while the mechanisms described here sound very interesting and novel, the global impact of the work is somehow mitigated by the fact that FF-NH2 is only active to a very restricted number of strains (essentially the *M. abscessus* complex). This is due to the absence or divergence of the amino acid sequence of Eis2 in other pathogens, which seriously limits the applicability of the concept to other (mycobacterial) pathogens. In addition, FF-NH2 does not outperform the standard drug clarithromycin in the mouse model.

There are major points that need to be clarified to further support and validate the conclusions raised.

Major points:

1. FF-NH2 resistant mutants were selected on plates (Figure S2), leading to large and rough colonies for *whiB7* colonies and small and smooth for *eis2* colonies. Both *whiB7* and *eis2* null mutants have been reported previously in the literature with no mention on changes in the morphology and/or growth rate. These differences in the morphology need to be clarified. The presence of the rough colonies can be simply explained by the presence of the antibiotic in the agar. What happens if these colonies are streaked on antibiotic-free plates, do they remain rough or become smooth? In addition, roughness is often linked to the absence of glycopeptidolipids (GPL). What are the GPL profiles of these *whiB7* and *eis2* mutants? Does complementation with *whiB7* restore a smooth background in the *whiB7* mutants?
2. Given the very small size of the *eis2* mutants (again not reported in previous studies) and growth impairment, how did the authors grow these strains to properly determine their drug susceptibility profiles (Table S2)?
3. The efficacy of FF-NH2 has been tested against a very small collection of clinical strains of the *M. abscessus* complex, including only two isolated of *M. abscessus* subsp. *abscessus* (Table S3). What does "CL I" mean. To support the usefulness of this new compound for the treatment of *M. abscessus* diseases, this table should be expanded by testing more clinical strains, including multidrug resistant strains of the *M. abscessus* complex.
4. An important finding is that FF-NH2 increases the expression of the *WhiB7* regulon, that includes *eis2* (Figure 3). However, this is only based on RNAseq and transcriptional data and needs to be proven at a protein level. I suggest the authors to confirm that *Eis2* expression is increased upon treatment of cultures with FF-NH2 by Western blotting.
5. I understand that FF-NH2 synergizes with different antibiotics, including macrolides, which are often used in clinical settings. However, the data in Fig. 4c also demonstrates that FF-NH2 antagonizes amikacin and cefoxitin, which are both used in the standard treatments. On the one hand, FF-NH2 increases the potency of macrolides but, on the other hand, it decreases the potency of aminoglycosides and β -lactams. Therefore, I suggest the authors to dampen this in the Results and Discussion sections. So, what would be the net result of FF-NH2 when combined with all these different drugs?
6. Related to the previous point, and in order to highlight the usefulness and applicability of FF-NH2 in treatment, drug combinations need to be evaluated *in vivo*. Fig. 4f, shows that FF-NH2 is not better than CLR or LNZ (given at lower concentrations than FF-NH2). Therefore, what would be the advantage of FF-NH2 from a therapeutic perspective, unless the authors can demonstrate that, in mice, the FF-NH2/CLR combination is better than the drugs administered alone? This data would make the study much more appealing.

Minor points:

-use the correct designation of gene throughout the whole manuscript and figures (for instance MAB_2989 rather than Mab2989).

-line 191: "indicate"

-line 717: "liver and spleen was determined in three mice". However, the CFU shown in Figure 4f suggest 5 mice/group. This needs to be better clarified (as well as the mode of drug administration) in the figure legend.

-Figure S4: I cannot distinguish the colors in the different graphs.

Reviewer #3 (Remarks to the Author):

Phelps and colleagues discovered that florfenicol amine (FF-NH2) acts as a narrow spectrum antibiotic against several non-tuberculous mycobacteria (NTM) including *M. abscessus*. Interestingly, FF-NH2 is a prodrug that requires activation by a modifying enzyme (*Eis2*) regulated by the transcription factor *WhiB7*. As is expected for prodrugs that require activation by a non-essential enzyme, the frequency of resistance is relatively high with loss of activation being the most frequent mechanism of resistance. Intriguingly, in the case of FF-NH2 resistance due to mutations inactivating *Eis2/WhiB7* leads to a collateral sensitivity to several other antibiotics, which could help develop resistance avoiding multi-drug regimens. Overall, this is a very convincing study that describes an attractive starting point to develop a new drug for the treatment of NTM infections (which are among the most difficult to cure infections).

My comments are all minor:

1. The abstract describes FF-NH2 as orally bioavailable, which is technically correct. But oral bioavailability is clearly limited. Otherwise subcutaneous administration would not have been necessary to demonstrate *in vivo* efficacy. Personally, I would tone down the claims regarding oral bioavailability.
2. Perhaps the authors could comment on why they used 6 day experiments for the synergy studies when 14 days seem to have been more informative (lines 263-265).

3. Consider rewording the sentence starting in line 322. Modification of the ribosomal target site likely occurs but with a frequency much lower than those affecting Eis2/WhiB7.

4. The authors describe leveraging drug resistance as a generalizable strategy for antibacterial drug development. I agree that this would be attractive, but I wonder how generalizable this strategy actually is. Perhaps the authors could add a few sentences to discuss at least one other resistance mechanism to which they think this strategy could be applied.

Version 1:

Reviewer comments:

Reviewer #2

(Remarks to the Author)

The authors have considerably improved the quality of the manuscript by conducting additional experimentations and have convincingly responded to all my previous concerns.

Reviewer #3

(Remarks to the Author)

This revised manuscript is much improved and addressed all concerns I expressed in a satisfactory manner.

Reviewer #4

(Remarks to the Author)

In the submitted manuscript "Hijacking intrinsic resistance in *Mycobacterium abscessus*", the authors report the discovery and characterization of an antibiotic pro-drug (FF-NH2) that has narrow spectrum activity against *Mycobacterium abscessus* (Mab) and low toxicity to mammalian cells, and which is insensitive to common resistance mechanisms. These properties make FF-NH2, or drugs that can be bioactivated by exploiting a similar mechanism, compelling leads for the development of novel therapeutics for non-tuberculous mycobacterial infections.

The authors commence their study by demonstrating that resistance to the existing SOC NMT drug chloramphenicol is mediated by the WhiB7-regulated enzyme Cat, and can be avoided by substitution of the C3 hydroxyl group with fluorine, as in florfenicol (FF). The authors subsequently discover that a FF metabolite, florfenicol amine (FF-NH2), demonstrates an inverse relationship to WhiB7 activity to chloramphenicol and requires the transcription factor for antibacterial activity. Through generating resistant mutants and biochemical studies, the authors validate the N-acetyltransferase Esi2 as the enzyme responsible for the bioactivation of FF-NH2 to generate FF-Ac, which inhibits the bacterial ribosome.

The most compelling aspects of this manuscript are that (a) FF-NH2/FF-Ac induce their own bioactivation by increasing expression Esi2 (supported by transcriptomic and proteomic data), and that (b) resistance developed to FF-NH2/FF-Ac increases bacterial sensitivity to other commonly used antibiotics, and vice versa. This provides a unique opportunity for FF-NH2 to improve the efficacy of antibiotic regimens. The authors demonstrate that FF-NH2 synergizes with some drugs used in existing SOC, (but appears to antagonize other compounds (Fig 5c)); and unlike chloramphenicol and FF, does not have the mammalian mitochondrial toxicity as a liability. Thus, while FF-NH2 is only modestly active, it could potentially be used at higher doses. Finally, the authors provide preliminary data regarding the pharmacokinetic properties of FF-NH2 and its efficacy in a mouse infection model, which is on par with comparator compounds.

Although this is not the first example of a pro-drug that is activated by an antibiotic resistance enzyme (e.g., beta-lactamase-activated prodrugs, *Antimicrob Agents Chemother.* 1976, 10, 245-8; *J. Med. Chem.* 2019, 62, 9, 4411–4425; aminomethyl oxazolidinones; *ACS Infect. Dis.* 2024, 10, 5, 1679–1695, etc.); it is the first example, to our knowledge, of a pro-drug that (a) induces its bioactivating enzyme and intracellular accumulation, and (b) where mutations that confer resistance to the pro-drug actually increase sensitivity to SOC, and vice versa. This unique mechanism should influence the field to think creatively about solutions to overcome antibiotic resistance and deepens our understanding of the complex interactions mediating drug efficacy. The science reported in this manuscript is robust and comprehensive. The authors have conducted detailed studies to elucidate the mechanisms of action, resistance, potential efficacy, and safety of FF-NH2. Experiments include a sufficient number of replicates, and data are represented clearly in a suitable way. Appropriate statistical analysis has been included where needed. The methods are sufficiently detailed to enable experiments to be repeated and source/raw data has been provided/uploaded into appropriate repositories for reuse.

There are several unanswered questions that remain from this study, including (a) why is there no intracellular accumulation of FF-NH2 in Mab Δ eis2?; (b) is the frequency of resistance decreased by combining FF-NH2 with standard-of-care (cycling or co-administration)?; (c) is the ribosome the main/only target of FF-Ac. This reviewer hopes these can be addressed in future studies, potentially through the use of chemical proteomics to profile the scope of protein targets that are covalently modified by FF-Ac.

I thoroughly enjoyed reviewing this manuscript and look forward to its publication. The potential for FF-amine to drive its own bioaccumulation and decrease the development of resistance when used in combination with existing antibiotics provides a compelling example of how exploiting intrinsic resistance mechanisms could be used to develop safer therapeutics or drugs with novel mechanisms. This concept can be applied across infectious diseases and potentially extended to other therapeutic contexts where resistance is problematic (e.g., cancer).

Minor comments:

- Line 460 – The authors do not mention why eis2 T258R is used as a control. Presumably this is a catalytically inactive enzyme or one of missense mutations identified in resistance mutants? It would be good to clarify in the text.
- Line 647 - A biosafety level statement should be included in the methods section as the manuscripts describes experiments using BSL-3 pathogens (*M. tuberculosis*, H37Rv). Currently, only the mouse experiments mention the use of BSL-3 facilities
- Line 856: “50 MM HEPES” > “50 mM HEPES”
- Line 860: “30 samples per day method”. This refers to proprietary method on the EvoSepOne instrument. It would be helpful for the authors to include, at minimum, the length and gradient of the LC elution profile to aid in reproducibility/in the case of a change in manufacturer settings.

Decision Letter:

Our ref: NMICROBIOL-24092832A

23rd July 2025

Dear Professor Lee,

Thank you for submitting your revised manuscript "Hijacking intrinsic resistance in *Mycobacterium abscessus*" (NMICROBIOL-24092832A). It has now been seen by two of the three original referees and their comments are below. Please note that we also added a proteomics referee given the revisions. The reviewers find that the paper has improved in revision, and therefore we'll be happy in principle to publish it in *Nature Microbiology*, pending minor revisions to satisfy the referees' final requests and to comply with our editorial and formatting guidelines. The comments by the additional proteomics referee can be addressed with text edits.

We are now performing detailed checks on your paper and will send you a checklist detailing our editorial and formatting requirements in about two weeks. Please do not upload the final materials and make any revisions until you receive this additional information from us.

Thank you again for your interest in *Nature Microbiology*. Please do not hesitate to contact me if you have any questions.

Sincerely,

Reviewer #2 (Remarks to the Author):

The authors have considerably improved the quality of the manuscript by conducting additional experimentations and have convincingly responded to all my previous concerns.

Reviewer #3 (Remarks to the Author):

This revised manuscript is much improved and addressed all concerns I expressed in a satisfactory manner.

Reviewer #4 (Remarks to the Author):

In the submitted manuscript "Hijacking intrinsic resistance in *Mycobacterium abscessus*", the authors report the discovery and characterization of an antibiotic pro-drug (FF-NH₂) that has narrow spectrum activity against *Mycobacterium abscessus* (Mab) and low toxicity to mammalian cells, and which is insensitive to common resistance mechanisms. These properties make FF-NH₂, or drugs that can be bioactivated by exploiting a similar mechanism, compelling leads for the development of novel therapeutics for non-tuberculous mycobacterial infections.

The authors commence their study by demonstrating that resistance to the existing SOC NMT drug chloramphenicol is mediated by the WhiB7-regulated enzyme Cat, and can be avoided by substitution of the C3 hydroxyl group with fluorine, as in florfenicol (FF). The authors subsequently discover that a FF metabolite, florfenicol amine (FF-NH₂), demonstrates an inverse relationship to WhiB7 activity to chloramphenicol and requires the transcription factor for antibacterial activity. Through generating resistant mutants and biochemical studies, the authors validate the N-acetyltransferase Esi2 as the enzyme responsible for the bioactivation of FF-NH₂ to generate FF-Ac, which inhibits the bacterial ribosome.

The most compelling aspects of this manuscript are that (a) FF-NH₂/FF-Ac induce their own bioactivation by increasing expression Esi2 (supported by transcriptomic and proteomic data), and that (b) resistance developed to FF-NH₂/FF-Ac increases bacterial sensitivity to other commonly used antibiotics, and vice versa. This provides a unique opportunity for FF-NH₂ to improve the efficacy of antibiotic regimens. The authors demonstrate that FF-NH₂ synergizes with some drugs used in existing SOC, (but appears to antagonize other compounds (Fig 5c)); and unlike chloramphenicol and FF, does not have the mammalian mitochondrial toxicity as a liability. Thus, while FF-NH₂ is only modestly active, it could potentially be used at higher doses. Finally, the authors provide preliminary data regarding the pharmacokinetic properties of FF-NH₂ and its efficacy in a mouse infection model, which is on par with comparator compounds.

Although this is not the first example of a pro-drug that is activated by an antibiotic resistance enzyme (e.g., beta-lactamase-activated prodrugs, *Antimicrob Agents Chemother.* 1976, 10, 245-8; *J. Med. Chem.* 2019, 62, 9, 4411–4425; aminomethyl oxazolidinones; *ACS Infect. Dis.* 2024, 10, 5, 1679–1695, etc.); it is the first example, to our knowledge, of a pro-drug that (a) induces its bioactivating enzyme and intracellular accumulation, and (b) where mutations that confer resistance to the pro-drug

actually increase sensitivity to SOC, and vice versa. This unique mechanism should influence the field to think creatively about solutions to overcome antibiotic resistance and deepens our understanding of the complex interactions mediating drug efficacy. The science reported in this manuscript is robust and comprehensive. The authors have conducted detailed studies to elucidate the mechanisms of action, resistance, potential efficacy, and safety of FF-NH2. Experiments include a sufficient number of replicates, and data are represented clearly in a suitable way. Appropriate statistical analysis has been included where needed. The methods are sufficiently detailed to enable experiments to be repeated and source/raw data has been provided/uploaded into appropriate repositories for reuse.

There are several unanswered questions that remain from this study, including (a) why is there no intracellular accumulation of FF-NH2 in Mab Δeis2?; (b) is the frequency of resistance decreased by combining FF-NH2 with standard-of-care (cycling or co-administration)?; (c) is the ribosome the main/only target of FF-Ac. This reviewer hopes these can be addressed in future studies, potentially through the use of chemical proteomics to profile the scope of protein targets that are covalently modified by FF-Ac.

I thoroughly enjoyed reviewing this manuscript and look forward to its publication. The potential for FF-amine to drive its own bioaccumulation and decrease the development of resistance when used in combination with existing antibiotics provides a compelling example of how exploiting intrinsic resistance mechanisms could be used to develop safer therapeutics or drugs with novel mechanisms. This concept can be applied across infectious diseases and potentially extended to other therapeutic contexts where resistance is problematic (e.g., cancer).

Minor comments:

- Line 460 – The authors do not mention why eis2 T258R is used as a control. Presumably this is a catalytically inactive enzyme or one of missense mutations identified in resistance mutants? It would be good to clarify in the text.
- Line 647 - A biosafety level statement should be included in the methods section as the manuscript describes experiments using BSL-3 pathogens (*M. tuberculosis*, H37Rv). Currently, only the mouse experiments mention the use of BSL-3 facilities
- Line 856: “50 MM HEPES” > “50 mM HEPES”
- Line 860: “30 samples per day method”. This refers to proprietary method on the EvoSepOne instrument. It would be helpful for the authors to include, at minimum, the length and gradient of the LC elution profile to aid in reproducibility/in the case of a change in manufacturer settings.

Version 2:

Decision Letter:

12th September 2025

Dear Richard,

I am pleased to accept your Article "Prodrug florfenicol amine is activated by intrinsic resistance to target *Mycobacterium abscessus*" for publication in *Nature Microbiology*. Thank you for having chosen to submit your work to us and many congratulations.

Over the next few weeks, your paper will be copyedited to ensure that it conforms to *Nature Microbiology* style. We look particularly carefully at the titles of all papers to ensure that they are relatively brief and understandable.

You may wish to make your media relations office aware of your accepted publication, in case they consider it appropriate to organize some internal or external publicity. Once your paper has been scheduled you will receive an email confirming the publication details. This is normally 3-4 working days in advance of publication. If you need additional notice of the date and time of publication, please let the production team know when you receive the proof of your article to ensure there is sufficient time to coordinate. Further information on our embargo policies can be found here:

<https://www.nature.com/authors/policies/embargo.html>

Authors may need to take specific actions to achieve compliance with funder and institutional open access mandates. If your research is supported by a funder that requires immediate open access (e.g. according to [a href="https://www.springernature.com/gp/open-science/plan-s-compliance">Plan S principles](https://www.springernature.com/gp/open-science/plan-s-compliance) or the [a href="https://www.springernature.com/gp/open-science/us-federal-agency-compliance">NIH public access policy](https://www.springernature.com/gp/open-science/us-federal-agency-compliance)) then you should select the gold OA route, and we will direct you to the compliant route where possible. Because authors warrant under

our subscription licensing terms that they haven't committed to licensing any version of their article under a licence inconsistent with the terms of our agreement – including the applicable embargo period – publication under the subscription model isn't suitable for authors whose funders require no embargo.

Congratulations once again and I look forward to seeing the article published.

With kind regards,

P.S. Click on the following link if you would like to recommend Nature Microbiology to your librarian
<http://www.nature.com/subscriptions/recommend.html#forms>

** Visit the Springer Nature Editorial and Publishing website at http://editorial-jobs.springernature.com?utm_source=ejp_NMicro_email&utm_medium=ejp_NMicro_email&utm_campaign=ejp_NMicro for more information about our career opportunities. If you have any questions please click [here](mailto:editorial.publishing.jobs@springernature.com).

use, sharing, adaptation, distribution and reproduction in any medium or format, as long as you give appropriate credit to the original author(s) and the source, provide a link to the Creative Commons license, and indicate if changes were made. In cases where reviewers are anonymous, credit should be given to 'Anonymous Referee' and the source. The images or other third party material in this Peer Review File are included in the article's Creative Commons license, unless indicated otherwise in a credit line to the material. If material is not included in the article's Creative Commons license and your intended use is not permitted by statutory regulation or exceeds the permitted use, you will need to obtain permission directly from the copyright holder. To view a copy of this license, visit <https://creativecommons.org/licenses/by/4.0/>

General response to reviewers:

We greatly appreciate the reviewers for their thoughtful and constructive feedback on our manuscript. It is encouraging to hear that experts in the field recognize the potential of this work. The reviewers' insights have greatly contributed to improving the rigor, clarity, and overall quality of the manuscript. In response, we have carefully addressed all recommendations by conducting additional experiments and revising the text, figures, and tables, accordingly, as outlined below.

We now provide evidence that chloramphenicol-NH₂ (CAM-NH₂) and thiamphenicol-NH₂ (TAM-NH₂) exhibit comparable activity to FF-NH₂ in the *MAB_2989* (*cat*) isogenic deletion strain (**Figure 1**). This finding highlights the involvement of Cat in resistance to phenicols containing the C3-hydroxyl group, underscoring the critical nature of substitution at this position (for example with fluorine) for the structure activity relationship. We have also validated both an *eis2* and *whiB7* mutant, in addition to the isogenic deletion strains, through *eis2* complementation and FF-NH₂ susceptibility testing (**Figures 2 and S3**). To supplement our transcriptomic findings at the protein level, we conducted a quantitative proteomics experiment, which demonstrates that WhiB7 and Eis2 are upregulated upon FF-NH₂ treatment (**Figure 3**). Furthermore, we have also expanded our rapidly-growing non-tuberculosis mycobacterium panel from 18 to 34 clinical isolates, showing FF-NH₂ displays robust activity against a number of *M. abscessus-chelonae* clinical isolates, including drug-resistant strains (**Table S5**). We have performed additional pharmacokinetic studies, metabolic stability, and metabolite identification studies of FF-NH₂ to supplement our previous preclinical characterization of FF-NH₂ (**Extended Data Figure S9, Table S8-10**).

We would like to emphasize that the primary objective of this manuscript is to introduce florfenicol-amine (FF-NH₂) and describe its mechanism of action/activation as a novel approach to tackling antimicrobial drug resistance. It is worth noting that we view FF-NH₂ as a chemical starting point, and ongoing research efforts are focused on designing and synthesizing novel derivatives with enhanced potency and improved pharmacokinetic profiles.

Note of RNA-sequencing analysis:

We previously used DESeq2 for differential gene expression analysis of transcriptomic profiles in our original submission. In a subsequent dataset prepared for the revision, FF-NH₂ treatment at 100 μM for 30 minutes and 3 hours identified 70 and 98 genes, respectively, with an FDR of 0 using the DESeq2 platform. To prevent the arbitrary assignment of FDR values in corresponding data representations, we performed statistical testing using *t*-statistics modified to incorporate gene strength into variance estimation, utilizing the Limma and Voom packages alongside edgeR. The FDR (q-values) displayed in the volcano plots in Figures 3b-c, S1a, and 4a-e and Extended Dataset are derived from Limma. These updates are reflected in the revised manuscript and do not impact the conclusions drawn from this study.

Point by point response:

We provide a detailed point-by-point response to each comment of the three reviewers in this document. We have included responses to each point by the reviewers in **red** below.

Referee 1: Pages 2-9

Referee 2: Pages 10-14

Referee 3: Pages 15-16

We truly hope that the revised manuscript with all the additional new data will meet the expectation of the reviewers for publication of this work in Nature Microbiology.

Reviewer 1

Comment 1.1. The authors report an intriguing discovery of collateral sensitivity by exploiting an N-acetyltransferase for bioactivation of an antibiotic precursor, converting it to an active drug, whereas that same N-acetyl transferase inactivates other classes of antibiotics. The opposing use of the same enzyme for bioactivation of one antibiotic and resistance to a different antibiotic provides a unique opportunity to exploit collateral sensitivity through antibiotic cycling. This approach also simultaneously eliminates the toxicity of the parent antibiotic, caused by inhibition of the corresponding host target. While antibiotics are generally regarded as highly selective and safe, several antibiotics targeting protein synthesis do inhibit the human mitochondrial ribosomes leading to host toxicity. Indeed, the antibiotic investigated, florfenicol (FF), is a member of phenicol class of antibiotics used worldwide for almost two decades before toxicity was broadly recognized that greatly curtailed the clinical use of this highly effective antibiotic. Antibiotics targeting the peptidyl-transferase center of the ribosome, like the phenicols, exhibit unavoidable mechanism-based toxicity since the binding site is highly conserved between humans and bacteria. Thus, the findings reported herein exploiting a bacterial N-acetyltransferase for selective bioactivation of des-acetyl FF (or FF-NH₂ in the manuscript) to FF mitigates the major liability of the phenicol class of antibiotics. Moreover, the described mechanism is selective for members of the *Mycobacterium abscessus-chelonae* complex, responsible for a growing number of incurable infection worldwide. The translation potential is significant since the resistance to des-acetyl FF would lead to collateral sensitivity to other protein synthesis inhibitors routinely used for therapy of these nontuberculous mycobacterial infections. The approach may be generalizable to other classes of antibiotics and indeed a recent paper by Boshoff and co-workers disclosed a related finding with the oxazolidinones (ACS Infect. Dis. 2024, 10, 5, 1679–1695, doi: 10.1021/acscinfecdis.4c00025). The study by Boshoff has many parallels and is certainly impressive in scope, but was focused on the medicinal chemistry and translational discovery, whereas the present manuscript is focused on mechanism, selectivity, antibiotic potentiation, and collateral sensitivity using different methodology. These studies are therefore highly complementary. Overall, the manuscript demonstrates a fascinating strategy that hijacks a pathogen's resistance mechanism against itself further exploiting positive regulation for amplification of the response. Antimicrobial resistance is inevitable, but the proposed findings offer a potential solution using antibiotic cycling to induce collateral sensitivity forcing bacteria to oscillate between two states, both of which are antibiotic susceptible. Moreover, the antibiotic described is effective in a murine *Mycobacterium abscessus* infection model showing translational potential. The work is deemed of high significance and impact and appropriate for Nature Microbiology once the following concerns have been addressed.

Response 1.1. We are delighted by the Reviewer's enthusiasm for this work and their insights into the future implications of this mechanism of action.

Comment 1.2. Can the authors cite some relevant clinical studies of Mab in addition to the cited 2020 review. What are the mortality rates of *M. abscessus* infections if the cure rates are so low?

Response 1.2. The following references have been added to the introduction: PMID 30880280 [ref. 14- *Mab*-PD cure rates], 31619468 [ref 15- mortality in NTM-PD], 38746951 [ref. 16- mortality of *Mab*-PD in transplant patients], and 39360834 [ref 18- recent review on *Mab*-PD and outlook]). Changes in text reflect the addition of these references.

Comment 1.3. Line 58. Change "Mycobacterium" to "mycobacteria"

Response 1.3. We have corrected this error as recommended by the Reviewer.

Comment 1.4. The authors may want to rephrase “However, closely related compounds within the phenicol class, such as florfenicol (FF)....” as “however” is used twice within three sentences. Chloramphenicol is on the WHO’ list of essential medicines and anecdotally appears to have high use in many developing countries, so the negative connotation of “however” may be unnecessary.

Response 1.4. We have revised the transitions in this section to improve context and readability.

Comment 1.5. Line 99. Why is IC₅₀ used instead of a more conventional minimum inhibitory concentration (MIC)? If it simply selected because the fold-difference at 50% inhibition between WT and whiB7 mutant strains for FF is maximal, then this the value is not justified. The growth inhibition curves (Figure 1, panel B, middle curve for FF-NH₂) show the MIC against the Mab WT is remarkably flat such that the actual MIC value is closer to 128 µg/mL. The use of GR₅₀ value or the concentration that inhibits 50% of the normalized growth rate during the early exponential phase is a superior metric for measuring drug responsiveness. (see: Hafner, M. et al. Nature Methods, 13 (6), 2016; Sasseti, et al. PNAS 2022, 19(15):e2201632119:521-7. doi: 10.1038/nmeth.3853).

Response 1.5. The Reviewer raises an excellent point regarding the use of IC₅₀ vs. MIC, a matter that the authors have carefully considered. FF-NH₂ displays significant trailing activity in standard MIC assays, similar to what is observed for tetracycline analogs like eravacycline against *Mab* (see: PMID: 35943269). Trailing MICs can pose challenges when reading and interpreting MIC values and might not capture the full activity of the compound, which is why we initially chose to present activity as fitted dose-response curves from the resazurin assay with the IC₅₀ values assigned rather than MIC values being called by visual inspection. We believe this representation of the data captures the trailing nature of FF-NH₂ activity while also highlighting the differences in activity between the strains tested.

In this resubmission, we have listed the MIC conservatively as 64 µg mL⁻¹ in the main text of this section and is shown in Table 1. Additionally, we have performed agar spotting assays (Extended Data Figure S2b), which demonstrate that FF-NH₂ prevents growth at 64 µg mL⁻¹ against Mab WT, compared to the FF-NH₂-resistant mutants.

Comment 1.6. The drug concentrations should also be reported in units of molarity (µM) at least somewhere.

Response 1.6. We thank the reviewer for this helpful suggestion to improve the readability of our manuscript. In response, we have added antimicrobial activity concentrations in units of molarity in “A WhiB7 balancing act exposes a vulnerability in *Mab*” section. Additionally, we have included exposure concentrations relative to relative to antimicrobial activity in the section discussing transcriptomics, proteomics, and accumulation studies to provide better context.

Comment 1.7. It would be nice to validate an *eis2* mutant genetically and/or biochemically. The validation provided was in isogenic deletion strain and complementation should ideally use a mutant background.

Response 1.7. Per the Reviewer’s suggestions, we have repeated the complementation experiments in two FF-NH₂ resistant mutant strains, *Mab* *Eis2* T258R and *Mab* *WhiB7* A51T, as well as the isogenic *Mab* Δ *whiB7* strain. The antimicrobial activity results for the mutant strains were similar to those of their

isogenic deletion counterparts. These data are now presented in Figure 2d,e and Extended Data Figure S3a,b in the revised manuscript.

Comment 1.8. Line 153. Please do not report IC50 values to 4 significant figures as the precision in this measurement is likely only to two significant figures. Use scientific notation. This applies to all fitted values throughout. In order to report to this level of precision, the drug must be accurately to 5 significant figures and all subsequent operations must be performed with such precision. Propagation of error in a two-fold series will lead to higher error at lower concentrations.

Response 1.8. We thank the Reviewer for noticing this error. We have corrected the values reported in the text and figures.

Comment 1.9. a) Line 161. Adjust the significant figures of Km. The symbols Km and Vmax should have the 'K' and 'V' italicized by standard convention. However, suggest reporting the kcat value rather than the Vmax by dividing by the enzyme concentration giving units of per second. The specificity constant (kcat/Km) is the best parameter for characterizing substrate specificity (expressed in units of M⁻¹s⁻¹) and this should be reported too. It would be useful to have one known substrate with reported kinetic parameters included in the Supporting Information Table (if one of the drugs has reported data already, then this can be included with a reference).

Response 1.9. We appreciate the Reviewer for their helpful comments regarding the enzymatic assay. In response to the Reviewer's suggestion, we have updated the kinetic parameters (Table S3 in revised manuscript) to include the *k_{cat}* and specificity constant (*k_{cat} K_M⁻¹*) values. These values for FF-NH₂ have also been reported in the corresponding section of the main text.

Additionally, we have incorporated our in-house results for the kinetics assay of two known substrates, amikacin and hygromycin B (see Table 2 in PMID 30880280 [ref. 45 in text]). We have provided a comparison of the *k_{cat}* and specificity constant (*k_{cat} K_M⁻¹*) values from both studies below for the Reviewer's convenience. Our study estimates these two controls as having approximately a two-fold lower catalytic efficiency, reflecting a more conservative assessment in our assay.

Compound	This study		Ung KL, et al. The FEBS Journal 286 (2019).	
	k_{cat} (s ⁻¹)	k_{cat} K_M⁻¹ (M ⁻¹ s ⁻¹)	k_{cat} (s ⁻¹)	k_{cat} K_M⁻¹ (M ⁻¹ s ⁻¹)
AMK	4.7 ± 0.5	2.0 × 10 ⁴ ± 3.2 × 10 ²	2.68 ± 0.25	4.51 × 10 ⁴ ± 4.27 × 10 ³
Hygromycin B	6.3 ± 0.4	3.1 × 10 ⁴ ± 4.5 × 10 ³	3.93 ± 0.57	6.29 × 10 ⁴ ± 14.2 × 10 ⁴

Comment 1.10. FF-NH₂ has a kcat/Km of 1.7x10e3 indicating it is a poor substrate and hence will be converted slowly. In the accumulation studies, the concentration of FF-NH₂ is far below Km, thus the Michaelis-Menten equation simplifies to kcat/Km x[E][S](where kcat/Km is effectively a second-order rate constant). This can be used to calculate the rate of bioactivation. Initially, Eis2 is uninduced and the constitutive expression is probably low. Using an estimate of 1 μM for Eis2, one can calculate the initial rate of bioactivation of FF-NH₂ (initial concentration ~ 10 nM) to FF-Ac, which equates to 1.7e103 M⁻¹s⁻¹ x [10x10⁻⁹ M] x [10⁻⁶ M] = 1.7e10⁻¹¹ M s⁻¹ = 1.7e10⁻⁵ μM s⁻¹. At 10 minutes (600 seconds), the total conversion would be: 1.7e10⁻⁵ μM s⁻¹ x 600 s = 10⁻² μM = 10 nM, which is within an order of magnitude of the measured concentration, but the bacteria remained viable until lysed, which by the protocol would have easily taken another 20 minutes giving an estimate of ≥30 nM within 2-fold of the observed value. At 3 hours, this predicts about 180 nM of FF-Ac, but as Eis2 is induced the rate of bioactivation will

increase. Indeed, the observed value of 1800 nM suggests significant induction. Thus, the kinetic parameters can be quite useful.

Response 1.10. We are grateful to the Reviewer for their thoughtful comments regarding the bioactivation of FF-NH₂ to FF-Ac and its dependency on Eis2 induction. At this time, we do not yet fully understand all the variables influencing bioactivation in a cellular system. Therefore, we have elected to not revise the main text currently. However, we have provided a detailed response below, which we believe strengthens the validity of the data contained in this revised version of the manuscript.

In response to Reviewer 2's Comment 2.5 below, we expanded our mechanism of action studies to include an unbiased proteomics experiment following FF-NH₂ (100 μM) exposure at two time points: 30-minutes and 3-hours, to align with the RNA sequencing dataset and accumulation studies. Notably, we repeated the RNA sequencing experiment using 100 μM FF-NH₂ in addition to the previously reported 25 μM treatment (this data has now been moved to Extended Data Figure S5). As a result, all data presented in Figure 3 (RNAseq, proteomics, and accumulation) were generated under consistent experimental conditions (100 μM FF-NH₂).

In the proteomics data set, we see that Eis2 is significantly induced at both 30-minutes (FC= 2.73; Log2FC = 1.45) and 3-hours (FC= 10.55; Log2FC = 3.4). For the sake of argument, if we assume a linear increase in Eis2 expression over the course of the 3-hour experiment, and using the kinetic parameters in Table S3 (kcat/km = 6.7 x 10³) and Michaelis-Menten equation outlined above, we see accumulation values within 2-fold of the observed value. These calculations are provided below for the Reviewer's reference.

Time (s)	Relative MabEis2 (μM) (Fig. 3d,e)	Average FF-Ac Accumulation (nM) Observed (Fig. 3g)	FF-Ac Accumulation (nM) Predicted	Relative difference (Predicted/Observed)
600	1.62*	83	65	0.77
1800	2.7	324	324	1.32
3600	4.32*	1261	1039	0.82
7200	7.56*	4781	3641	0.76
10800	10.8	4930	7806	1.58

The underlined values were observed from the proteomics experiment in Fig. 3d,e, whereas the values denoted with an asterisks were derived from linear extrapolation using the equation $*=0.0009(s)+1.0675$.

Comment 1.11. The rate of bioactivation should also be discussed briefly as this becomes an important parameter in the context of animal studies where the drug concentration is not constant, but decreasing with time. Thus, if the rate of drug elimination in vivo ($t_{1/2} = 0.8$ h, Table S.6) is faster than the rate of bioactivation ($t_{1/2} = 6.8$ h, estimated initial rate before induction of Eis2, $k_{obs} = k_{cat}/K_m \times [E] = (1.7 \times 10^3 \text{ M}^{-1}\text{s}^{-1}) \times [10^{-6} \text{ M}] = 1.7 \times 10^{-3} \text{ M}$; $t_{1/2} = \ln 2/k_{obs} = 6.8$ h), then efficacy in vivo cannot be obtained. This likely explains the need to dose subcutaneously, when the drug has 50% oral bioavailability.

Response 1.11. We thank the Reviewer for their insights regarding the bioactivation of FF-NH₂ from an in vivo perspective. Currently, we do not feel confident addressing this comment in the manuscript, as it would require extrapolation that may overinterpret data obtained from independent experiments. Nevertheless, we have provided a detailed response below to address the Reviewer's comment.

The Reviewer is concerned that the rate of bioactivation may limit FF-NH₂'s in vivo efficacy due its short half-life. Below we have extrapolated out the rate of bioactivation based on our proteomics dataset conducted at 100 μM (equivalent to 28.4 μg mL⁻¹). Without Eis2 induction, the calculated rate of bioactivation is approximately 1.72 hours, which we agree is above the drug elimination half-life and

would be problematic. However, at the 30-minute (1800 s) Eis2 induction rate (2.7-fold increase), this rate is reduced to less than 0.5-hours.

Time (s)	$k_{cat} \text{ } KM^{-1} \text{ (} M^{-1} \text{ s}^{-1} \text{)}$	Relative MabEis2 (μM) (Fig. 3d,e)	k_{obs}	Time to Bioactivation (h)
0	6.70E+03	1.00	0.00670	1.72
600	6.70E+03	1.61	0.01077	1.07
1800	6.70E+03	2.69	0.01801	0.64
3600	6.70E+03	4.31	0.02886	0.40
7200	6.70E+03	7.55	0.05057	0.23
10800	6.70E+03	10.79	0.07228	0.16

In reference to Comment 1.20, we have conducted the pharmacokinetic studies at the 400 mg kg⁻¹ SC dose. This dosing regimen achieved a C_{max} of 108 $\mu\text{g mL}^{-1}$ at the 0.83-hour time point, which is approximately 4 times higher than the concentration used in the proteomics study (100 μM or 28.4 $\mu\text{g mL}^{-1}$). Additionally, based on the terminal half-life of this dosing strategy ($t_{1/2} = 1.02$ h), the concentration would remain above the 28.4 $\mu\text{g mL}^{-1}$ threshold beyond the 0.5-hour time point. These results would suggest that concentrations sufficient for in vivo activation would likely be achieved.

We appreciate the Reviewer's attention to detail regarding the relationship between in vitro results and in vivo observations. These calculations emphasize the dynamic nature between the rate of bioactivation and elimination, providing a valuable framework for further investigation in future studies into dosing strategies and in vivo potential of FF-NH₂ and next generation amine phenicols'.

Comment 1.12. TAM-NH₂ and CAM-NH₂ are bioactivated with similar or better efficiencies (TAM-NH₂ is 3x better) yet TAM-NH₂ shows no enhancement in activity against the whiB7 mutant, despite being a significantly better substrate (three-fold better than FF-NH₂) while CAM-NH₂ shows a modest 2-fold increase in potency against the whiB7 mutant. The authors should provide a rationale for the observed differences.

Response 1.12. The Reviewer brings up a valid point. Based on the results from this study, we believe that the activity of amine phenicols against *Mab* is primarily influenced by three parameters: Eis2 affinity, Cat affinity, and ribosomal affinity of the N-acetylated product. The transcriptomic and updated proteomic analyses (see Response 2.5 below) indicate that the induction of the WhiB7 regulon by FF-NH₂ leads to increased expression of both Cat (MAB_2989) and Eis2 (MAB_4532c). This presents a conundrum, as Eis2 activity is necessary for activating CAM-NH₂ and TAM-NH₂, while Cat functions to inactivate them, resulting in a relative balance of activation and inactivation. The mechanism described in this report is dynamic, and further studies are required to fully elucidate the structure-activity relationship of these new series.

The observation mentioned by the Reviewer could be explained by TAM-NH₂ being a better substrate for Cat inactivation. Alternatively, TAM-NH₂ may be a less potent translational inhibitor. Given that both CAM and CAM-NH₂ have better activities than TAM and TAM-NH₂ in each *Mab* strain tested (see updated Figure 1, orange curves), we believe these differences could initially be attributed to variations in translation inhibition activity (CAM > TAM), which has been previously described (PMID: 2228823). Regardless, the presence of the C3-hydroxyl group is clearly detrimental to whole-cell activity, leading us to focus our efforts on FF and FF-NH₂ due to their ability to evade Cat.

Comment 1.13. Was the concentration of FF-Ac in the supernatant measured. The discussion invokes Fick's law to explain accumulation, but this would presume the FF-Ac is trapped within the bacteria.

Presumably, FF-Ac can equilibrate with the extracellular environment. The compound could be sequestered by the abundant ribosomes explaining their accumulation.

Response 1.13. We apologize for any confusion. The discussion regarding Fick's Law and Le Chatelier's principle pertains to FF-NH₂, rather than FF-Ac. Although FF-NH₂ is inactive, it is intriguing that its accumulation is also dependent on the expression of Eis2 and WhiB7, indicating that its conversion to FF-Ac is necessary for its own accumulation.

This portion of the discussion section was included because it suggests a strategy for rationally designing compounds with modifiable motifs (such as amines) that inducible microbial enzymes can act upon as a means of improving drug accumulation. However, several alternative scenarios could also explain this observation, such as changes in cell wall permeability, induction of transporter proteins, or expression of an unidentified deacetylase enzyme which can convert FF-Ac back to FF-NH₂. For these reasons, we have elected to remove this portion of the discussion section.

Comment 1.14. Does O-acetyltransferase Mab2989 (Cat) confer resistance to the amphenicols? Once bioactivated they may be susceptible to inactivation by Cat. What is the MIC when Cat is deleted? This would be valuable information and may explain the overall modest activity of the amphenicols.

Response 1.14. We have added the activity data of the series against *Mab Δ2989 (cat)* to Figure 1 (orange curves), which we hope will clarify these observed differences. MAB_2989 (Cat) does indeed confer resistance to phenicol antibiotics that possess a C3-hydroxy group, including CAM, TAM, CAM-NH₂, and TAM-NH₂. FF and FF-NH₂, which possess a fluorine at this position, avoid inactivation by Cat, which is critical for its ability to exploit native *Mab* resistance.

Comment 1.15. The time-course of the RNA-Seq experiments is very nice. The experimental states 25 uM of FF-NH₂ and FF were used that corresponds to about 1/8 of the MIC values (MIC is ~64 ug/mL = 200 uM). It would also useful to give the concentration relative to the MIC values. Expressing the concentration in units of molarity throughout would be preferred. Was the concentration optimized?

Response 1.15. We appreciate the Reviewer's positive feedback regarding the RNA-seq study. We apologize for the inconsistency in reporting units and have now included concentrations relative to MIC values in this mechanism of action section.

Initially, the concentration used for the transcriptomic studies was based on 1/2 FF-NH₂ IC₅₀ values, which corresponds to 1/8 MIC, as the Reviewer correctly noted. In the subsequent accumulation/conversion assay, treatment of cells with 25 μM was inadequate to reliably detect FF-NH₂ at 10-minutes, prompting us to increase the concentration to 100 μM for that assay. For consistency across our experimental platforms, we repeated the RNA-sequencing experiment at 100 μM. Figure 3 now includes this updated RNA-seq data (Fig. 3b,c), proteomics data (Fig. 3d,e; see Response 2.5 below) and accumulation data (Fig. 3f,g), all obtained upon treatment with 100 μM FF-NH₂. The transcriptomic profiles for the 25 μM treatment have been moved to Extended Data Figure S5.

Comment 1.16. The reported analytical method for compound accumulation does not report an internal standard that is required for quantitation. No information is provided on how the intracellular concentration was determined. No information is provided for the chromatography (column, gradient).

Response 1.16. We thank the Reviewer for this comment. We analyzed these samples using Echo MS system, which employs acoustic droplet ejection to deliver nanoliter-sized droplets directly to the mass spectrometer using a carrier liquid (methanol with 1 mM ammonium fluoride), thereby eliminating the need for traditional liquid chromatography separation. We have updated the corresponding method section to clarify these differences.

Additionally, we apologize for not including internal standards (IS) in our initial quantitation report. We have now repeated the analysis using warfarin as the IS for FF-Ac and FF-NH₂ measured in positive mode, and propylparaben as the IS for FF measured in negative mode. In our previous submission, accumulation concentrations of unknown compounds were determined by the linear fit of the peak area for corresponding standards in extraction buffer (50% acetonitrile). For the subsequent analysis, we used area ratio (compound/IS) using lysate from mock treated samples for our calibration curve. In the positive mode, we observed significant ion suppression due to the lysate matrix compared to extraction buffer alone, resulting in higher intracellular concentrations of FF-NH₂ and FF-Ac reported in this revision. All raw and processed data, including chromatograms for total ion and exact masses for FF, FF-NH₂, FF-Ac, warfarin, and propylparaben, are included in Extended Dataset for Accumulation.

Of note, we were able to reprocess all data points except for the 0.167 h time point for *Mab ΔwhiB7* Replicate C due to sample mishandling. Both of the other replicates had levels below the limit of quantification, so we did not prioritize repeating that sample and have noted this in the methods section.

Comment 1.17. The reported clearance and volume of distribution for the pharmacokinetic experiments in Table S6 are incorrect. These are based on data from one mouse with a reported half-life of 2.4 hours following intravenous administration, whereas the oral half-life is 0.8 hours. These values are incongruent.

Response 1.17. Table S6 from the previous submission was originally derived from a pharmacokinetic report provided by Sai Life Limited. To address the Reviewer's concerns regarding discrepancies, we have added a pharmacokinetics expert, Dr. Bernd Meibohm (University of Tennessee Health Science Center), and his team as coauthors to this study.

Under Dr. Meibohm's supervision, the pharmacokinetic studies were repeated to ensure accuracy and consistency. Additionally, as requested Comment 1.20 below, we have included the pharmacokinetic parameters for BALB/c mice dosed at 400 mg/kg subcutaneously (SC), as well as 300 mg/kg by oral route (PO). These updated data are now included in Table S10 of the revised manuscript and are provided below for the Reviewer's reference.

Dose	t _{max} [h]	C _{max} [mg L ⁻¹]	AUC _{inf} [mg h L ⁻¹]	t _{1/2} [h]	F [%]
10 mg/kg IV	0.083	6.28	1.92	0.28	100
30 mg/kg PO	0.25	5.96	2.55	0.74	44
300 mg/kg PO	0.25	57.7	38.0	0.74	66
400 mg/kg SC	0.083	108	65.9	1.02	86

Comment 1.18. Line 300. The data is reported to 6 significant figures. Based on the method, the data should not be reported beyond 2 significant figures.

Response 1.18. We apologize once again for this error. We have corrected all reported values throughout the manuscript to reflect the correct number of significant figures.

Comment 1.19. The reported Pharmacokinetic Studies in the Supplementary methods lack information on the bioanalysis (internal standard, LC-MS methods).

Response 1.19. We have revised the experimental section for the pharmacokinetic studies, which is detailed in the Supplementary Materials and Methods section of the Supplementary Information document accompanying this resubmission.

Comment 1.20. Since mice were dosed subcutaneously, the PK for this route of administration would be important, to demonstrate that sufficient exposure was obtained.

Response 1.20. As noted in Response 1.17 above, we have included the PK parameters for this route of administration and dose in Table S10.

Comment 1.21. Source data is provided for the RNA-Seq experiments, but is lacking for everything else. All data reported in the manuscript and the Supplementary methods should have all data for every data point provided to reproduce the values given in the Figures and Tables.

Response 1.21. We have provided the complete Source Data for all figures presented in this manuscript.

Comment 1.22. Information on metabolism of FF-NH₂ should be provided given Boshoff observed the linezolid amine metabolite exhibited pronounced mitochondrial cytotoxicity.

Response 1.22. We thank the Reviewer for this comment. In response, we have conducted metabolic stability studies, including metabolite identification in human, rat, and mouse hepatocytes.

Following hepatocyte incubation, the major metabolites identified were a reduction product and a glucuronide conjugate in mouse hepatocytes, and a glucuronide conjugate in rat hepatocytes. In contrast, no such metabolites were detected after incubation with human hepatocytes. Metabolites formed through oxidative deamination or N-acetylation — metabolites suspected of mitochondrial toxicity as described in the Boshoff manuscript — were not detected in any of the three species. These findings have now been incorporated into the revised manuscript (Extended Data Figure S9; Table S9).

Reviewer 2

Comment 2.1. *M. abscessus* infections are extremely difficult to treat, requiring lengthy treatments often associated with therapeutic failure and toxicity. New therapeutic approaches are, therefore, needed to bypass these toxicity issues. In this study, Phelps et al. introduced an amine derivative in florfenicol (FF-NH₂), a drug used in veterinary medicine. Through the use of MedChem, genetic and biochemical approaches, the authors demonstrated that FF-NH₂ avoids neutralization by O-acetylation (catalyzed by Cat) and, at the same time, exploits N-acetylation (catalyzed by Eis2) in *M. abscessus*. Both Cat and Eis2 are part of the WhiB7 regulon, associated to resistance to different antibiotic classes, such as aminoglycosides and macrolides. Herein, the authors show that FF-NH₂ acts as a prodrug that requires activation by Eis2, leading to the accumulation of the active acetyl metabolite, FF-Ac, which in turn inhibits ribosome translation. Importantly, this prodrug displays no toxicity linked to mammalian mitochondrial ribosome inhibition, as compared to the parental FF molecule. In addition, it is orally bioavailable and synergizes with other anti-*M. abscessus* drugs.

In general, the manuscript is very well-written and the results are supported by the data with appropriate controls. The mode of action/activation of FF-NH₂ described here is particularly original as it demonstrates the possibility to exploit the presence of an intrinsic drug resistance element known to be detrimental to other antibiotics used to treat the same pathogen. However, while the mechanisms described here sound very interesting and novel, the global impact of the work is somehow mitigated by the fact that FF-NH₂ is only active to a very restricted number of strains (essentially the *M. abscessus* complex). This is due to the absence or divergence of the amino acid sequence of Eis2 in other pathogens, which seriously limits the applicability of the concept to other (mycobacterial) pathogens. In addition, FF-NH₂ does not outperform the standard drug clarithromycin in the mouse model.

Response 2.1. We thank the Reviewer for their positive assessment of our work. We found the Reviewer's comments helpful and have addressed the criticisms noted above and below in the revised manuscript, which we hope will meet the expectations of the referee for publication of this work in *Nature Microbiology*. We believe there is a place for narrow-spectrum Mabs therapy due to the long duration of treatment of this severe chronic infection. This specificity may also enable much-needed diagnostic development.

Major points:

Comment 2.2. FF-NH₂ resistant mutants were selected on plates (Figure S2), leading to large and rough colonies for *whiB7* colonies and small and smooth for *eis2* colonies. Both *whiB7* and *eis2* null mutants have been reported previously in the literature with no mention on changes in the morphology and/or growth rate. These differences in the morphology need to be clarified. The presence of the rough colonies can be simply explained by the presence of the antibiotic in the agar. What happens if these colonies are streaked on antibiotic-free plates, do they remain rough or become smooth? In addition, roughness is often linked to the absence of glycopeptidolipids (GPL). What are the GPL profiles of these *whiB7* and *eis2* mutants? Does complementation with *whiB7* restore a smooth background in the *whiB7* mutants?

Response 2.2. We agree with the Reviewer that one of the most puzzling observations from this study is the phenotypic difference between the *whiB7* and *eis2 Mab* mutants selected on FF-NH₂ agar plates.

To address this concern, we further characterized the ten FF-NH₂-resistant mutants verified by whole-genome sequencing and used for the high-throughput collateral sensitivity screen (Figure 4a,b). Starting from single colonies, the strains were grown in antibiotic-free media to an OD₆₀₀ of 0.4-0.8, serially diluted ten-fold, and plated on agar with and without FF-NH₂ (Extended Data Figure S2b). Of note, in the absence of FF-NH₂, these colonies do not display a growth defect and reached plating density at the same time (roughly 48-hours). Although all the *whiB7* mutants were initially selected as “large and rough” colonies, they reverted to a smooth phenotype when FF-NH₂ pressure was removed. However, upon addition of FF-NH₂, the *whiB7* mutants began to form larger, rough colonies in the plating assay. Furthermore, we demonstrated that this phenotype is transient through passaging experiments utilizing the *Mab ΔwhiB7* strain and *Mab WhiB7 A51T* mutant (Extended Data Figure S2c).

Overexpression of WhiB7 has been reported to induce a switch from a smooth to a rough phenotype (PMID: 26195529). Temporarily overexpression of WhiB7 is induced upon exposure of wild-type *Mab* to florfenicol amine (Figure 3b-e), leading to a rough morphotype. Acquisition of loss-of-function mutations in *whiB7* results in FF-NH₂ resistance, but nascent colonies maintain the rough phenotype until FF-NH₂ pressure is removed or until functional WhiB7 is diluted through bacterial cell division. On the other hand, antibiotics such as amikacin can induce a smooth-to-rough transition in *M. abscessus* (PMID: 28752292). FF-NH₂ resistant mutants with *WhiB7* mutations have rough morphotype, whereas FF-NH₂ resistant mutants with *eis2* mutations maintain a smooth phenotype. This indicates that *whiB7* may upregulate genes associated with glycopeptidolipid (GPL) biosynthesis, and its disruption impairs GPL production under FF-NH₂ treatment. Once FF-NH₂ is removed, GPL biosynthesis can resume *whiB7* independently.

While this observation was helpful for phenotypic selection of various mutants in our study, a comprehensive investigation into the mechanism underlying the morphotype switch is beyond the scope of this work.

Comment 2.3. Given the very small size of the *eis2* mutants (again not reported in previous studies) and growth impairment, how did the authors grow these strains to properly determine their drug susceptibility profiles (Table S2)?

Response 2.3. The Reviewer brings up a valid concern regarding potential fitness cost. As mentioned in Comment 2.2, the *eis2* mutants only display a fitness cost in the presence of FF-NH₂. All the strains used for susceptibility profiles were grown in antibiotic-free media. For transparency, the drug-responsivity curves for each strain tested against the anti-*Mab* antibiotic panel (Figure 5a-b) is now included in Extended Data Figure S8, with the raw and processed experimental data provided as Source Data in the updated revision.

Comment 2.4. The efficacy of FF-NH₂ has been tested against a very small collection of clinical strains of the *M. abscessus* complex, including only two isolated of *M. abscessus* subsp. *abscessus* (Table S3). What does “CL I” mean. To support the usefulness of this new compound for the treatment of *M. abscessus* diseases, this table should be expanded by testing more clinical strains, including multidrug resistant strains of the *M. abscessus* complex.

Response 2.4. We thank the Reviewer for this suggestion, and we agree to validate translatability, we needed to test additional clinical isolates, including drug-resistant isolates. We also apologize for the lack of clarity regarding strain details.

We have expanded our rapidly-growing non-tuberculosis mycobacterium panel from 18 to 34 clinical isolates, 25 of which belong to the *M. abscessus-chelonae* clade. This expansion includes details regarding the strain designation, relevant mutations associated with drug-resistance, and susceptibility data against backbone therapeutics, amikacin and clarithromycin (Table S5). These data show that FF-NH₂ displays robust activity (MIC ≤ 64 µg mL⁻¹) against all the *M. abscessus-chelonae* clinical isolates, including drug-resistant strains.

Comment 2.5. An important finding is that FF-NH₂ increases the expression of the WhiB7 regulon, that includes *eis2* (Figure 3). However, this is only based on RNAseq and transcriptional data and needs to be proven at a protein level. I suggest the authors to confirm that *Eis2* expression is increased upon treatment of cultures with FF-NH₂ by Western blotting.

Response 2.5. The Reviewer raises an important consideration regarding the potential for disconnect between transcription and translation, especially given that the activated molecule inhibits translation. Since antibodies specific to MabEis2 are not commercially available, we employed an unbiased mass spectrometry based proteomic approach to demonstrate changes in *Eis2* protein level upon treatment with FF-NH₂. The results of this study show that the expression of *Eis2*, as well as *Cat* and *WhiB7*, is significantly increased at 30 minutes (Log₂FC = 1.45) and 3 hours (Log₂FC = 3.4) after treatment with 100 µM of FF-NH₂. These differential protein expression data are now displayed in Figure 3d-e and Source Data is provided in Extended Data Proteomics.

Raw mass spectra from proteomics experiment have also been deposited to proteomeXchange and MassIVE repositories with identifiers PXD059834 and MSV000096854. Reviewers can see the data using FTP.

ftp://MSV000096854@massive.ucsd.edu
Login with Filezilla (requires FTP and not SFTP, no encryption)
Host: massive.ucsd.edu
Username: MSV000096854
Password: Reviewer_login

Comment 2.6. I understand that FF-NH₂ synergizes with different antibiotics, including macrolides, which are often used in clinical settings. However, the data in Fig. 4c also demonstrates that FF-NH₂ antagonizes amikacin and cefoxitin, which are both used in the standard treatments. On the one hand, FF-NH₂ increase the potency of macrolides but, on the other hand, it decreases the potency of aminoglycosides and b-lactams. Therefore, I suggest the authors to dampen this in the Results and Discussion sections. So, what would be the net result of FF-NH₂ when combined with all these different drugs ?

Response 2.6. We agree that the dichotomy between synergy and antagonism among front-line antibiotics used to treat *Mab* infections could pose a challenge for standard regimens. However, it is important to highlight that standard regimens do not work for most patients (PMID: 30880280, 31619468).

The purpose of this section in the original submission was to highlight the in vitro results showing FF-NH₂ ability to work favorably with macrolides and linezolid, both of which are orally bioavailable. An all-oral regimen would be highly encouraging for patients initiating therapy. This regimen could select for resistance via *eis2* mutations, at which point bactericidal agents like aminoglycosides, which now demonstrate a much greater degree of efficacy, could be applied (see Lore NI, et al. Eur Respir J. 2022; PMID: 36265879).

Admittedly, this is speculative, and additional studies will be needed to validate the therapeutic potential of molecules like FF-NH₂. In response to the Reviewer's concerns, we have toned down this section considerably.

Comment. Related to the previous point, and in order to highlight the usefulness and applicability of FF-NH₂ in treatment, drug combinations need to be evaluated in vivo. Fig. 4f, shows that FF-NH₂ is not better than CLR or LNZ (given at lower concentrations than FF-NH₂). Therefore, what would be the advantage of FF-NH₂ from a therapeutic perspective, unless the authors can demonstrate that, in mice, the a FF-NH₂/CLR combination is better than the drugs administered alone? This data would make the study much more appealing.

Response 2.7. We agree that demonstrating superior efficacy compared to CLR or LZD in vivo would be ideal. However, at this stage, we believe that a compound with a novel mechanism of activation offers significant scientific value and merit for further study and publication in this prestigious journal, even if it does not yet outperform existing therapeutics.

Regarding the Reviewer's point on the therapeutic advantages of FF-NH₂, we identify two key points. First, the prodrug nature of FF-NH₂ is particularly appealing, as it may reduce the broad-spectrum effects and off-target toxicities associated with extended treatment durations, while precisely targeting the infection source. Second, resistance that could emerge through selection might lead to collateral sensitivity to other anti-*Mab* agents, such as standard-of-care treatments like amikacin. While a broader spectrum of activity against mycobacteria would undoubtedly be ideal, the novelty of this research is the concept that drug-resistance enzymes can be exploited for pro-drug activation – a concept we believe is not unique to *Mab*.

Our updated pharmacokinetic results (Table S10) indicate that FF-NH₂ is rapidly eliminated from the body, resulting in drug concentrations above the MIC being maintained for only a small portion of the dosing interval. This aligns with the natural metabolic pathway of FF, where conversion to FF-NH₂ facilitates elimination in many animal species [PMID: 37273507 (salmon), 17147454 (swine), 17570454 (dogs)]. Consequently, FF-NH₂ is not a viable clinical candidate under standard dosing regimens.

To address the Reviewers' comments regarding drug-combination testing, we tested FF-NH₂ in combination with CLR and LZD in the GM-CSF KO murine acute infection model (see results below). In this preliminary experiment, we did not see increased efficacy compared to each drug by themselves, except for slight reductions in the FF-NH₂ + LZD in the liver and spleen CFU. The combination drug dosing in this experiment was unoptimized. We speculate that the lack of combined efficacy may be attributed to FF-NH₂'s short pharmacokinetic profile, which limits the two drugs' overlap in exposure. As *M. abs* animal infection models are notoriously challenging, future studies will focus on optimizing dosing regimens for FF-NH₂ and analogs to achieve adequate exposures and expand in vivo efficacy in acute and chronic infection models that better recapitulate how these infections are experienced in humans.

Minor points:

Comment 2.8. use the correct designation of gene throughout the whole manuscript and figures (for instance MAB_2989 rather than Mab2989).

Response 2.8. We thank the Reviewer for noticing this error. We have corrected the gene/protein designations throughout the manuscript.

Comment 2.9. line 191: “inidcate”

Response 2.9. We appreciated the careful eye of the Reviewer. This specific sentence has been revised.

Comment 2.10. “liver and spleen was determined in three mice”. However, the CFU shown in Figure 4f suggest 5 mice/group. This needs to be better clarified (as well as the mode of drug administration) in the figure legend.

Response 2.10. We apologize for any confusion. The sentence referenced by the Reviewer pertains to the baseline bacterial burden in the different organs at day 0 and day 2 post-infection. These baseline counts are available upon request. The values presenting in Fig. 4f represent the final bacterial burden (CFU counts), which were enumerated on day 12 and include data from 5 mice per group. To clarify the experimental design, we have updated the figure legend to include details on the number of mice per group and the route of administration.

Comment 2.11. I cannot distinguish the colors in the different graphs.

Response 2.11. We apologize for this oversight. All figures throughout the manuscript are now displayed as different colors and symbols. For the volcano plots, genes of interest are specifically labeled on each graph.

Reviewer 3

Comment 3.1. Phelps and colleagues discovered that florfenicol amine (FF-NH₂) acts as a narrow spectrum antibiotic against several non-tuberculous mycobacteria (NTM) including *M. abscessus*. Interestingly, FF-NH₂ is a prodrug that requires activation by a modifying enzyme (Eis2) regulated by the transcription factor WhiB7. As is expected for prodrugs that require activation by a non-essential enzyme, the frequency of resistance is relatively high with loss of activation being the most frequent mechanism of resistance. Intriguingly, in the case of FF-NH₂ resistance due to mutations inactivating Eis2/WhiB7 leads to a collateral sensitivity to several other antibiotics, which could help develop resistance avoiding multi-drug regimens. Overall, this is a very convincing study that describes an attractive starting point to develop a new drug for the treatment of NTM infections (which are among the most difficult to cure infections).

Response 3.1. We thank the Reviewer for the positive assessment of our work.

Minor points:

Comment 3.2. The abstract describes FF-NH₂ as orally bioavailable, which is technically correct. But oral bioavailability is clearly limited. Otherwise subcutaneous administration would not have been necessary to demonstrate in vivo efficacy. Personally, I would tone down the claims regarding oral bioavailability.

Response 3.2. We agree with the Reviewer. Due to limited potency and a poor PK profile, we wanted to push the exposure to establish proof-of-principle in vivo efficacy. While oral bioavailability is key for the applicability of future analogs in this series, we have toned this aspect down in this manuscript.

Comment 3.3. Perhaps the authors could comment on why they used 6 day experiments for the synergy studies when 14 days seem to have been more informative (lines 263-265).

Response 3.3. We looked to take a high-throughput approach for these experiments in order to test many strains against a panel of anti-*Mab* antibiotics. In our experience, incubation of plates for longer durations (> 7 days) in a 384-well format results in significant edge effects and inconsistent measurements.

To address this concern, we conducted standard MIC testing for five strains, including *Mab* WT, *Mab* Δ *whiB7*, *Mab* Δ *eis2*, *Mab* Eis2 T258R, and *Mab* WhiB7 A51T, against macrolides and peptide antibiotics (aminoglycosides and capreomycin). All of the *eis2* mutants displayed increased sensitivity to the peptide antibiotics, similar to what was previously shown in Figure 4. In addition to peptide antibiotics, the *whiB7* mutants also displayed increased sensitivity to the macrolides and spectinomycin. These results align with previous reports by others in the field (PMID: 28874378, 28874379). These data are now displayed in Supplemental Table S6 and the text has been updated to reflect this.

Comment 3.4. Consider rewording the sentence starting in line 322. Modification of the ribosomal target site likely occurs but with a frequency much lower than those affecting Eis2/WhiB7.

Response 3.4. We thank the Reviewer for this observation and agree that resistance could emerge through mutations in the ribosomal binding site. We have revised this sentence in the revision to acknowledge the possibility of mutations at this site.

Comment 3.5. The authors describe leveraging drug resistance as a generalizable strategy for antibacterial drug development. I agree that this would be attractive, but I wonder how generalizable this strategy actually is. Perhaps the authors could add a few sentences to discuss at least one other resistance mechanism to which they think this strategy could be applied.

Response 3.5. We thank the Reviewer for this foresighted comment. Rather than focusing on a specific gene and organism where this mechanism could be applied, we provide a broader response to highlight the potential of exploiting enzymes involved in chemical modification as a general strategy to discover new antibiotics.

We have added the following passage to the end of the discussion section to touch on generalizability:

“Eis2 is one of over 272,000 GNAT proteins found in bacteria, several of which are directly implicated in antimicrobial resistance. While acetylation is just one form of chemical modification utilized by bacteria, other mechanisms— including phosphorylation, adenylation, and hydroxylation—also contribute to drug resistance, and we hypothesize that, in the right circumstances, some of these enzymes could also be leveraged for drug activation for antibiotic discovery.”

General response to reviewers:

We sincerely appreciate all of the Reviewers' thoughtful, thorough, and constructive feedback throughout this process, which has been instrumental in shaping the manuscript. It is gratifying to have the potential impact of our work recognized by experts in the field.

Point by point response:

We provide a detailed point-by-point response to each comment of the three reviewers in this document. We have included responses to each point by the reviewers in **red** below.

Reviewer 2

Comment 2.1. The authors have considerably improved the quality of the manuscript by conducting additional experimentations and have convincingly responded to all my previous concerns.

Response 2.1. We thank the Reviewer for their time and are pleased that the revisions have addressed your concerns.

Reviewer 3

Comment 3.1. This revised manuscript is much improved and addressed all concerns I expressed in a satisfactory manner.

Response 3.1. We appreciate the Reviewer's efforts and are glad that our revisions have satisfactorily addressed all concerns.

Reviewer 4

Comment 4.1. In the submitted manuscript "Hijacking intrinsic resistance in *Mycobacterium abscessus*", the authors report the discovery and characterization of an antibiotic pro-drug (FF-NH₂) that has narrow spectrum activity against *Mycobacterium abscessus* (Mab) and low toxicity to mammalian cells, and which is insensitive to common resistance mechanisms. These properties make FF-NH₂, or drugs that can be bioactivated by exploiting a similar mechanism, compelling leads for the development of novel therapeutics for non-tuberculous mycobacterial infections.

The authors commence their study by demonstrating that resistance to the existing SOC NMT drug chloramphenicol is mediated by the WhiB7-regulated enzyme Cat, and can be avoided by substitution of the C3 hydroxyl group with fluorine, as in florfenicol (FF). The authors subsequently discover that a FF metabolite, florfenicol amine (FF-NH₂), demonstrates an inverse relationship to WhiB7 activity to chloramphenicol and requires the transcription factor for antibacterial activity. Through generating resistant mutants and biochemical studies, the authors validate the N-acetyltransferase Esi2 as the enzyme responsible for the bioactivation of FF-NH₂ to generate FF-Ac, which inhibits the bacterial ribosome.

The most compelling aspects of this manuscript are that (a) FF-NH₂/FF-Ac induce their own bioactivation by increasing expression Esi2 (supported by transcriptomic and proteomic data), and that (b) resistance developed to FF-NH₂/FF-Ac increases bacterial sensitivity to other commonly used antibiotics, and vice versa. This provides a unique opportunity for FF-NH₂ to improve the efficacy of antibiotic regimens. The authors demonstrate that FF-NH₂ synergizes with some drugs used in existing SOC, (but appears to

antagonize other compounds (Fig 5c)); and unlike chloramphenicol and FF, does not have the mammalian mitochondrial toxicity as a liability. Thus, while FF-NH₂ is only modestly active, it could potentially be used at higher doses. Finally, the authors provide preliminary data regarding the pharmacokinetic properties of FF-NH₂ and its efficacy in a mouse infection model, which is on par with comparator compounds.

Although this is not the first example of a pro-drug that is activated by an antibiotic resistance enzyme (e.g., beta-lactamase-activated prodrugs, *Antimicrob Agents Chemother.* 1976, 10, 245-8; *J. Med. Chem.* 2019, 62, 9, 4411–4425; aminomethyl oxazolidinones; *ACS Infect. Dis.* 2024, 10, 5, 1679–1695, etc.); it is the first example, to our knowledge, of a pro-drug that (a) induces its bioactivating enzyme and intracellular accumulation, and (b) where mutations that confer resistance to the pro-drug actually increase sensitivity to SOC, and vice versa. This unique mechanism should influence the field to think creatively about solutions to overcome antibiotic resistance and deepens our understanding of the complex interactions mediating drug efficacy.

The science reported in this manuscript is robust and comprehensive. The authors have conducted detailed studies to elucidate the mechanisms of action, resistance, potential efficacy, and safety of FF-NH₂. Experiments include a sufficient number of replicates, and data are represented clearly in a suitable way. Appropriate statistical analysis has been included where needed. The methods are sufficiently detailed to enable experiments to be repeated and source/raw data has been provided/uploaded into appropriate repositories for reuse.

There are several unanswered questions that remain from this study, including (a) why is there no intracellular accumulation of FF-NH₂ in Mab Δ eis2?; (b) is the frequency of resistance decreased by combining FF-NH₂ with standard-of-care (cycling or co-administration)?; (c) is the ribosome the main/only target of FF-Ac. This reviewer hopes these can be addressed in future studies, potentially through the use of chemical proteomics to profile the scope of protein targets that are covalently modified by FF-Ac.

I thoroughly enjoyed reviewing this manuscript and look forward to its publication. The potential for FF-amine to drive its own bioaccumulation and decrease the development of resistance when used in combination with existing antibiotics provides a compelling example of how exploiting intrinsic resistance mechanisms could be used to develop safer therapeutics or drugs with novel mechanisms. This concept can be applied across infectious diseases and potentially extended to other therapeutic contexts where resistance is problematic (e.g., cancer).

Response 4.1. We sincerely thank Reviewer 4 for their thorough and insightful review of our manuscript. We appreciate the time devoted to evaluating our work. Your encouraging comments on the rigor, clarity, and comprehensiveness of our study are valued, and we share your enthusiasm for the potential of this approach to inspire new strategies against antibiotic resistance, as well as its broader implications for other resistance contexts. We look forward to exploring the unanswered questions you raised in future studies.

Comment 4.2. Line 460 – The authors do not mention why eis2 T258R is used as a control. Presumably this is a catalytically inactive enzyme or one of missense mutations identified in resistance mutants? It would be good to clarify in the text.

Response 4.2. We appreciate the reviewer's observation. In response to a previous request, we included complementation experiments with spontaneously generated FF-NH₂-resistant mutants alongside the isogenic deletion strains from our original submission. Eis2 T258R and WhiB7 A51T were chosen as representative missense mutations identified in resistant isolates, rather than nonsense mutations. Both WhiB7 A51T and Eis2 T258R were included in Extended Data Figure 2, Figure 5a,b and Extended Data Figure 8. These data demonstrate that their morphotypes and susceptibility patterns are consistent with those of other spontaneous mutants and isogenic deletions, indicating that these mutations result in catalytic inactivity. We have clarified this in the text as follows:

“To confirm Eis2 expression is responsible for amine phenicol activity, we conducted susceptibility testing using both isogenic deletion strains (Δ eis2, Δ whiB7) and spontaneously generated missense mutants (Eis2 T258R, WhiB7 A51T), each complemented with either the multicopy pOLYG-aac(3)IV-eis2 vector or the pOLYG-aac(3)IV control vector.”

Comment 4.3. Line 647 - A biosafety level statement should be included in the methods section as the manuscript describes experiments using BSL-3 pathogens (*M. tuberculosis*, H37Rv). Currently, only the mouse experiments mention the use of BSL-3 facilities

Response 4.3: Thank you for highlighting this oversight. We have updated the Methods section to include a biosafety level statement as follows:

“All work with *Mycobacterium tuberculosis* H37Rv was conducted in a certified BSL-3 facilities in accordance with institutional and national biosafety regulations.”

Comment 4.4. Line 856: “50 MM HEPES” > “50 mM HEPES”

Response 4.4. We thank the reviewer for noting this error. It has been corrected to “50 mM HEPES” in the revised manuscript.

Comment 4.5. Line 860: “30 samples per day method”. This refers to proprietary method on the EvoSepOne instrument. It would be helpful for the authors to include, at minimum, the length and gradient of the LC elution profile to aid in reproducibility/in the case of a change in manufacturer settings.

Response 4.5. We have updated the Methods section to include details on the elution profile for the EvoSepOne instrument, as recommended. Please note that specific parameters are preset and proprietary, and detailed settings are not openly accessible.